# SOTIP is a versatile method for microenvironment modeling with spatial omics data

Zhiyuan Yuan [1,2,3,6] ✉, Yisi Li[3,6], Minglei Shi[4], Fan Yang[2], Juntao Gao[3], Jianhua Yao [2] ✉ & Michael Q. Zhang [3,4,5] ✉

The rapidly developing spatial omics generated datasets with diverse scales and modalities. However, most existing methods focus on modeling dynamics of single cells while ignore microenvironments (MEs). Here we present SOTIP (Spatial Omics mulTIPle-task analysis), a versatile method incorporating MEs and their interrelationships into a unified graph. Based on this graph, spatial heterogeneity quantification, spatial domain identification, differential microenvironment analysis, and other downstream tasks can be performed. We validate each module's accuracy, robustness, scalability and interpretability on various spatial omics datasets. In two independent mouse cerebral cortex spatial transcriptomics datasets, we reveal a gradient spatial heterogeneity pattern strongly correlated with the cortical depth. In human triple-negative breast cancer spatial proteomics datasets, we identify molecular polarizations and MEs associated with different patient survivals. Overall, by modeling biologically explainable MEs, SOTIP outperforms state-of-art methods and provides some perspectives for spatial omics data exploration and interpretation.

Tissue and cell state are jointly modulated by the intracellular gene regulatory network and extracellular ecosystem (i.e., the microenvironment). Recent studies highlighted the importance of such microenvironment in tissue homeostasis[1–4], disease occurrence[5–7], and tumor progression[8–10]. For instance, in healthy vertebrate liver, the zonation patterns of hepatocytes perform mutual effect on the microenvironment within liver lobules[4]. While during liver fibrosis, there occurs metabolic alteration around the liver-fibrotic microenvironment[7]. In cancer, researchers also reported that different degrees of immune cell infiltration was associated with patient survival[11–13].

Recent years, various spatial omics technologies have been developed, including spatial transcriptomics (e.g., in situ sequencing (ISS)[14], single-molecule fluorescence in situ hybridization (smFISH)[15], multiplexed error-robust fluorescence in situ hybridization (MERFISH)[16], sequential fluorescence in situ hybridization (seqFISH)[17], spatial transcriptomics (ST)[18], slide-seq[19,20], High-definition spatial transcriptomics (HDST)[21], stereo-seq[22]), spatial proteomics (e.g., iterative indirect immunofluorescence imaging (4i)[23], co-detection by indexing (CODEX)[24], multiplexed ion beam imaging by time-of-flight (MIBI-TOF)[12,25], imaging mass cytometry (IMC)[13,26], as well as spatial metabolomics (e.g., airflow-assisted

[1]Institute of Science and Technology for Brain-Inspired Intelligence; MOE Key Laboratory of Computational Neuroscience and Brain-Inspired Intelligence; MOE Frontiers Center for Brain Science, Fudan University, Shanghai 200433, China. [2]Tencent AI Lab, Shenzhen, China. [3]MOE Key Laboratory of Bioinformatics; Bioinformatics Division and Center for Synthetic & Systems Biology, BNRist; Department of Automation, Tsinghua University, Beijing 100084, China. [4]MOE Key Laboratory of Bioinformatics; Bioinformatics Division and Center for Synthetic & Systems Biology, School of Medicine, Tsinghua University, Beijing 100084, China. [5]Department of Biological Sciences, Center for Systems Biology, The University of Texas, Richardson, TX 75080-3021, USA. [6]These authors contributed equally: Zhiyuan Yuan, Yisi Li. ✉e-mail: zhiyuan@fudan.edu.cn; jianhuayao@tencent.com; michael.zhang@utdallas.edu

desorption electrospray ionization mass spectrometry imaging (AFADESI-MSI)[27], SpaceM[28], spatial single nuclear metabolomics (SEAM)[7]. These advances provide fertile biomolecular profiles of each observation unit (viz., spots, pixel, or single cells) as well as the corresponding spatial coordination, thus facilitating deep and systematic investigation of tissue microenvironment.

Although above technologies have achieved unprecedented profiling coverage and/or spatial resolution, only a few studies focused on the quantitative description of microenvironment (ME)[29–31]. Existing studies typically represent an ME as a vector of cell type frequency within cellular neighborhood, and use Euclidean distance among ME representations to construct a graph (e.g., KNN graph[32]), which is further exploited to perform downstream analysis such as clustering (termed classical cell neighborhood representation, CCNR, Supplementary Table 1)[29–31]. While easily implemented, this practice fails to consider the mutual relationship of cells within the ME, which may undermine the reliability of the subsequently constructed ME graph. This defect would easily hinder the downstream analytical performance especially in cases containing continuous cell states, such as liver[2] and cortex[33].

The complexity of the cell composition within the spatial context is reported to be associated with disease progression[34] and tissue development[16,33]. Classical methods quantify the heterogeneity of an ME by counting the number of unique cell clusters within it (termed NUCC, Supplementary Table 1)[16,33]. Such practice assumes equal importance of different clusters, potentially leading to the inaccuracy of heterogeneity quantification. For example, they may fail to distinguish the heterogeneity difference between two MEs (one is composed of tumor cells and immune cells, and another is composed of two subsets of immune cells).

Partitioning tissues into spatial domains with the integration of both gene expression and spatial information is an important task in tissue biology. Existing methods can be divided into three categories based on their computational principles: Zhu et al.[35] and BayesSpace[36] (Supplementary Table 1) applied Hidden-Markov random field (HMRF) to model the spatial dependency of hidden variables associated with spots. SpaGCN[37] (Supplementary Table 1) utilized existing structures in the field of graph convolutional network (GCN) to aggregate gene expression features according to the spatial graph. StLearn[38] (Supplementary Table 1) normalized the spatial information, tissue histology, and gene expression features to perform clustering. While these methods and other very recent preprints[39–41] achieved good performance in their cases, they may lack proper interpretation for the parameters (e.g., weight trade-off between spatial prior and conditional distributions for HMRF-based methods) or features (e.g., features learned from deep learning-based methods) of biological entities. In addition, they may not be flexible enough to be compatible with different spatial omics data types.

Here, we present SOTIP (Fig. 1), a scalable method to jointly perform three main spatial omics tasks, namely, spatial heterogeneity (SHN) quantification (Fig. 1e), spatial domain (SDM) identification (Fig. 1f), differential microenvironment (DME) analysis (Fig. 1g), and other downstream spatial omics tasks (Fig. 1h) within a unified and biologically interpretable framework. With SOTIP, we propose to use MECN (molecular-expression-aware cellular neighborhood) as an agent to abstract and mathematically represent the microenvironment (Fig. 1b). SOTIP's algorithm modules perform these tasks based on an MECN graph (MEG, Fig. 1d), in which each node represents an MECN and the edge weight between two nodes was the function of cell states discrepancy (Fig. 1c) and cell state composition (Fig. 1b) within MEs. In

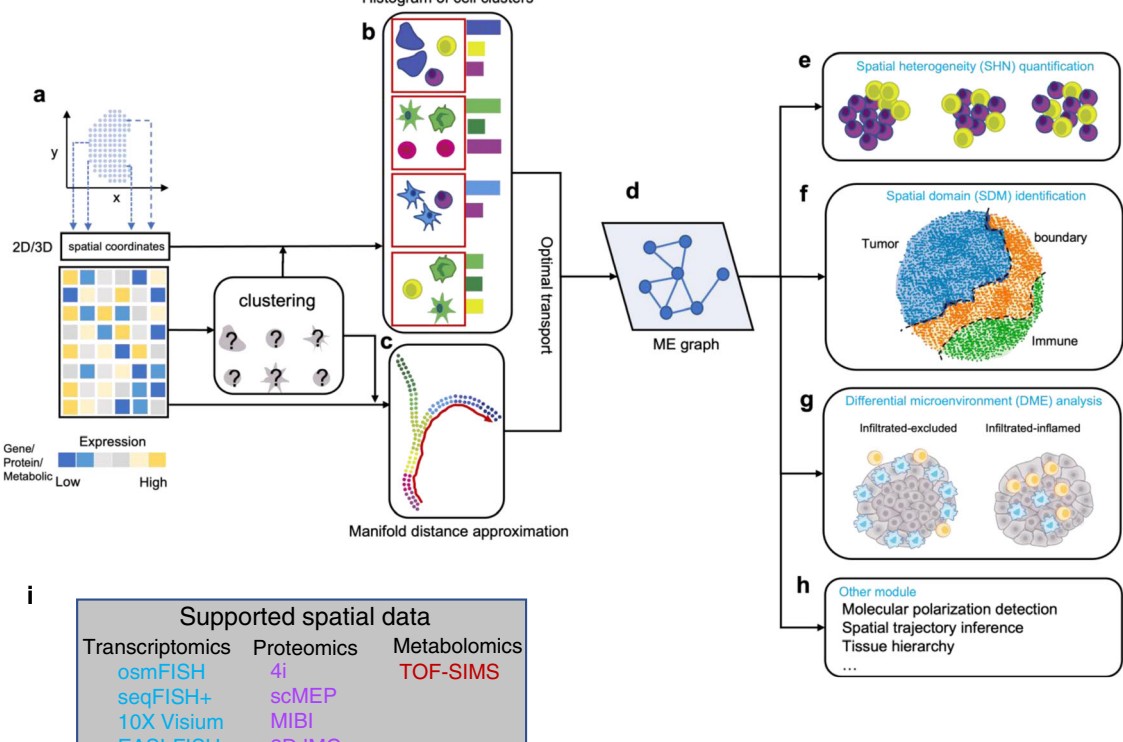

**Fig. 1 | Schematic overview. a–h** Overview of SOTIP. SOTIP takes 2D/3D spatial omics data (spatially resolved gene expression, protein or metabolic profiles) as input (**a**). Next, the cell (spot) type clustering is obtained solely by the expression matrix. Then the histogram of cell clusters is obtained combing the spatial information and clustering result (**b**). The manifold distance is approximated by combing clustering result and the expression matrix (**c**). The ME graph is built using optimal transport between histograms characterizing microenvironments (MEs) (**d**). Finally, multiple tasks can be performed based on the MECN graph, including spatial heterogeneity quantification (**e**), spatial domain identification (**f**), differential microenvironment analysis (**g**), and other downstream tasks (**h**). **i** The supported spatial data.

this manner, gene expressions, single cells, microenvironments, and tissue regions can be linked in a biologically meaningful way. Then SHN is calculated by measuring the gene expression variation within each node, SDM was identified by hierarchical merging nodes guided by the connectivity derived from edges, and DME analysis was performed by assessing nodes' relative densities between conditions on the MEG. We demonstrate all SOTIP modules with various types of spatial omics data[42–45] (Fig. 1i, Supplementary Table 3), including FISH-based[46,47] and sequencing-based[48] spatial transcriptomics, mass-spectrometry-based and fluorescence-based spatial proteomics[12,49], and secondary ion mass spectrometry (SIMS) based spatial metabolomics[7]. We summarize the datasets in Supplementary Table 8 and the demonstration logic in Supplementary Figs. 13–15.

## Results
### Model rational
To smooth the reading, we describe SOTIP's rational in this section (see "Methods" for more technical description). In the following, we use "cells" to denote the measurement units in spatial omics data for ease of explanation, which can be freely replaced to "spots" in technologies like 10X Visium or ST.

For any data with N cells and P features (e.g., gene expression vector with length P), traditional (non-spatial) clustering methods[32] directly construct a graph for the N cells, by computing distances on the P feature vectors. Then Leiden[50] algorithm is performed on this graph to get the clustering result. Such traditional clustering methods only take into account the gene expression profile of each cell, without considering the neighborhood information of each cell.

On the contrary, SOTIP considers both gene expression profiles of cells, and the neighborhood information of cells. The main idea of SOTIP is to represent the neighborhood of each cell (neighborhood indexed by cell-i is called MECN-i) as a histogram of cell types (following figure shows example of 3 MECNs characterized by histograms), then SOTIP uses the earth mover's distance (EMD, a distance metrics for histograms) based on an optimal transport principle to construct a graph (termed MEG) between all MECNs, and finally a clustering procedure is performed on MEG to get the clustering result (i.e., spatial domain identification result). Here, it is worth re-emphasizing that MEG is the graph of all MECNs, each node of MEG is a MECN, and edge between nodes is the distance between two MECNs, which is the EMD distance between the two histograms characterizing the two MECNs.

The key design of SOTIP is construction of the MECN graph (MEG) using EMD. In this way, when measuring the distance between two MECNs (e.g., MECN-i and MECN-j, indexed by cell-i and cell-j), SOTIP considers how to design an optimal transportation plan on the gene expression manifold to move cells (characterized by cell type) within MECN-i to match cells within MECN-j, so that the histogram characterizing MECN-i after the transportation is exactly the same as the histogram characterizing MECN-j. Another key design is that when designing the optimal transportation plan, the cells must be moved along the gene expression manifold. The optimal transportation cost is used as the distance between MECN-i and MECN-j. One can imagine that the transportation cost is determined by two factors: (1) the number of cells to be moved, and (2) the cost of moving cells between two positions along the gene expression manifold. The first factor has already been encoded in the histograms associated with MECNs, and the second factor (named ground distance in optimal transport theorem) is the geodesic distance between cell types along the gene expression manifold.

### SOTIP framework demonstration with in silico spatial transcriptomics data
To demonstrate SOTIP's utility and to conduct adequate comparisons with other methods, we generated three sets of in silico spatial

transcriptomics data by simulating scRNA-seq data, followed by arranging cell spatial coordinates (see "Methods" and Supplementary Fig. 1). It is worth noting that since each MECN is generated by searching the spatially nearest neighbors of each cell, in the following sections, every MECN (consisted with multiple cells within the neighborhood of a center cell) is associated with its center cell.

In the first simulation, we aimed to compare SOTIP with other methods in SHN quantification. Other methods include NUCC[16,33] and IGD (see Methods "NUCC and IGD"). To achieve this goal, we simulated three clusters of single cells with inter-cluster discrepancy (Fig. 2a left). Then we randomly positioned these three cell clusters as a sequential tissue band on a two dimensional plane (Fig. 2a right). With this simulation, we expected higher spatial heterogeneity around tissue boundaries than inner tissue, which corresponded to higher SHN values between C2 (middle band in Fig. 2a right) and C3 (right band in Fig. 2a right) than between C1 (left band in Fig. 2a right) and C2. We next separately performed SOTIP, IGD and NUCC[16,33] (Supplementary Table 1) to quantify the spatial heterogeneity across the tissue sample. The result showed that all three methods highlighted two tissue boundaries (Fig. 2b), while only SOTIP successfully quantified the difference between two boundaries as expected (Fig. 2b right). We also tested how cluster size influence the analysis, and the result also showed higher SHN peaks around highlighted SHNs at C1-C2 boundary, and lower SHN peaks at C2-C3 boundary, both peaks are significantly higher than background (Supplementary Fig. 12).

Since SOTIP relies on the MECN graph (MEG) to represent MECN relationships, the quality of MEG determines the performance of downstream tasks. In the second simulation, we aimed to evaluate the quality of the MECN graph constructed by SOTIP. To do this, we generated five single cell clusters along a continuous manifold (Fig. 2c left), and regularly mixed these five clusters in five sequential bands on a two-dimensional plane (Fig. 2c right). In this data, each cell type occupies a major component in a band, and other cell types occupy minor components in its band. The arrangement of major cell types along these tissue bands (R1 to R5) is in accordance with the order of cell types along the gene expression manifold (C1 to C5) (Fig. 2c middle). With this simulation, we expected that R1 to R5 microenvironments should display a continuous pattern when embedded into low-dimensional space. We separately performed SOTIP and CCNR[29–31] (Supplementary Table 1) on this simulation data, and embedded their generated graphs with three popular embedding algorithms (diffusion maps[51], UMAP[52] and PHATE[53], Fig. 2d). The results showed that both methods can separate these 5 MECNs in the embedded space (Fig. 2d), while only SOTIP well preserved the continuous order of R1 to R5 with different embedding methods (Fig. 2d bottom row).

In the third simulation, we want to ask whether SOTIP can perform differential microenvironment (DME) analysis to identify specific microenvironments between samples, even if these samples shared exactly the same cellular composition. Since there is currently no other method for this task, it was not possible to compare different methods on this type of data. We firstly generated three discrete single-cell clusters (Fig. 2e left), then positioned them in two different orders, thus in silico generating two different tissue samples with exactly the same cell composition but different spatial organizations (Fig. 2e middle and right). With this simulation, differential abundance analysis based on non-spatial scRNA-seq[54–56] failed to detect any difference between the two samples, but SOTIP successfully highlighted (Fig. 2f) the major differential MECNs (C1-C2 boundary in sample 1, and C1-C3 boundary in sample 2), by estimating the relative likelihood of observing each microenvironment in the two tissue samples[55] (see "MELD analysis" in "Methods"). We also tested the case where C1 and C2 are more similar than C3 in gene expression (Supplementary Fig. 11), and the analysis result also highlighted unique microenvironments of each sample.

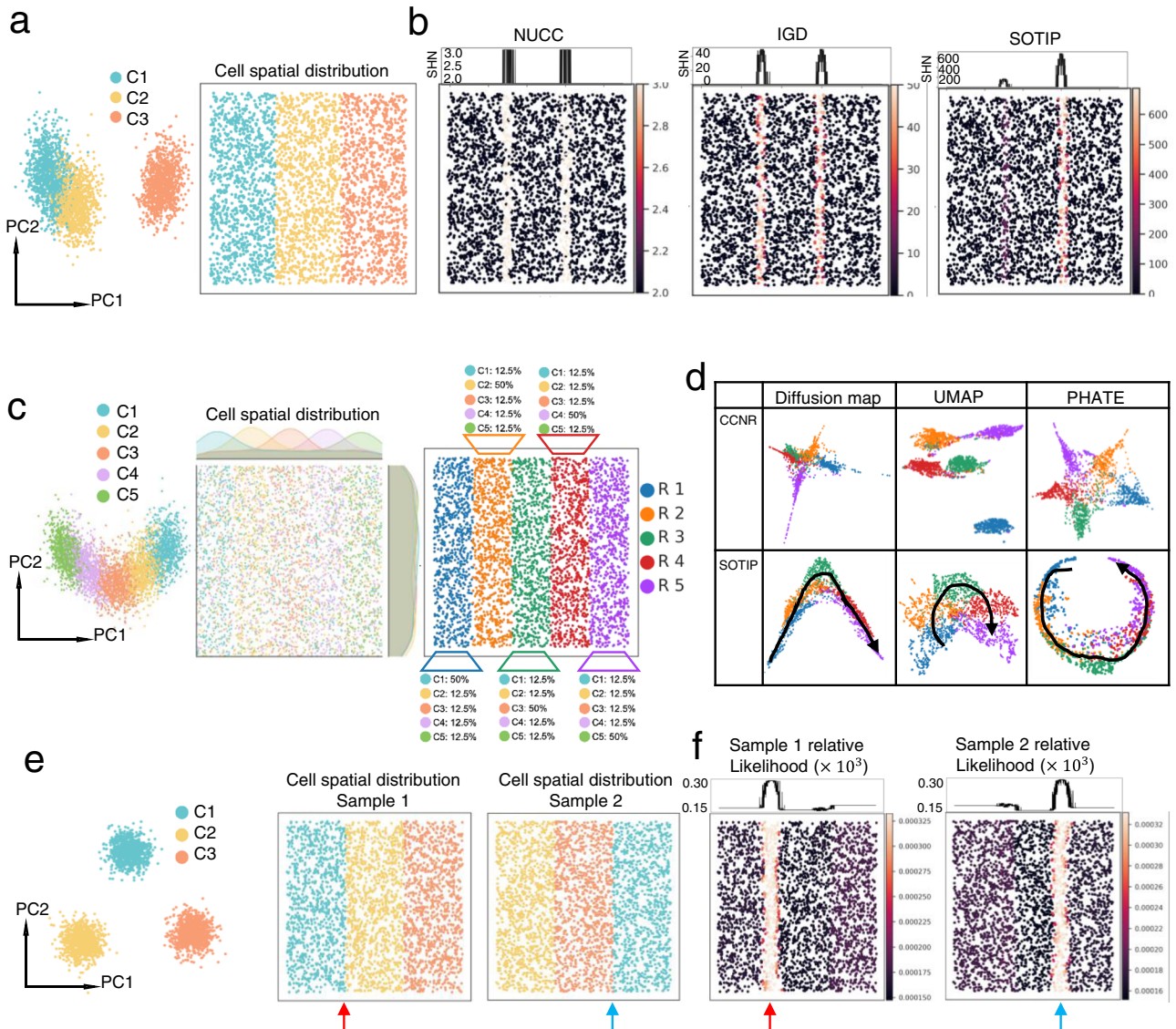

**Fig. 2 | Simulation study. a** PCA plot (left) and spatial distribution (right) of simulation data 1. In both plots, each point represents a cell. Both plots share the same color-coding scheme. **b** Comparison of NUCC (left), IGD (middle) and SOTIP (right) of SHN quantification performance on simulation data 1. Each panel consists of the two parts, the top part shows the SHN value of each MECN as a function of horizontal coordinate, and the bottom part shows the same set of cells with spatial coordinates as in Fig. 2a right, colored according to SHN value of associated MECN. **c** PCA plot (left), spatial distribution (middle) and tissue regions (right) of simulation data 2. In all plots, each point represents a cell. PCA plot (left) and spatial distribution (middle) share the same color-coding scheme, different with tissue region (right). **d** MECN graphs generated by CCNR (top row) and SOTIP (bottom row) are embedded with different algorithms, i.e. diffusion map (left column), UMAP (middle column) and PHATE (right column). In each embedding plot, each

point represents a MECN, colored according to the clustering label of its center cell in Fig. 2c right. **e** PCA plot and two version of spatial arrangement of simulation data 3. Each point represents a cell. All the plots share the same color-coding scheme. The red arrow points to C1-C2 boundary in sample 1, and blue arrow points to C3-C1 boundary in sample 2. **f** Sample-specific microenvironments (left panel for sample 1, right panel for sample 2) are highlighted by MEG constructed with SOTIP. Each panel consists of two parts. Take the left panel as an example, the bottom part shows the same set of cells as in Fig. 2e middle, colored according to the relative likelihood of observing each microenvironment in sample 1, and the top part shows the value of relative likelihood of each MECN as a function of horizontal coordinate. The right panel shares similar configuration. Same as (**e**), the red arrow points to C1-C2 boundary in sample 1, and blue arrow points to C3-C1 boundary in sample 2.

All these simulation results supported that SOTIP accurately described the microenvironments and corresponding properties, and detected finer biological differences, complementing with the existing methods.

**SOTIP quantifies accurate SHNs in accordance with subcellular and tissue structures**

In eukaryotic cells, the nuclear envelope compartmentalizes the DNA in nucleus and the protein synthesis machine in the cytoplasm, thus separating transcription from translation[57]. The spatial separation of two fundamentally different compositions makes the nuclear envelope

to be most heterogeneous region within a eukaryotic cell[58] (Fig. 3a). Iterative indirect immunofluorescence imaging (4i)[23] is a robust protocol that can achieve multiplexed protein staining at nanometer resolution, thus providing opportunities to characterize the covariance among molecular, spatial, and morphological properties of subcellular structures. Squidpy[59] provided a subset of original data, consisting of 270,876 pixels from 13 HeLa cells (~20,836 pixels per cell) with 40 protein assayed[23]. The data also provided pixel-level nucleus/cytoplasm annotations. Within each cell, since the nuclear envelope is expected to display the highest heterogeneity, if we take the scaled heterogeneity as the probability of being the nuclear envelope, the

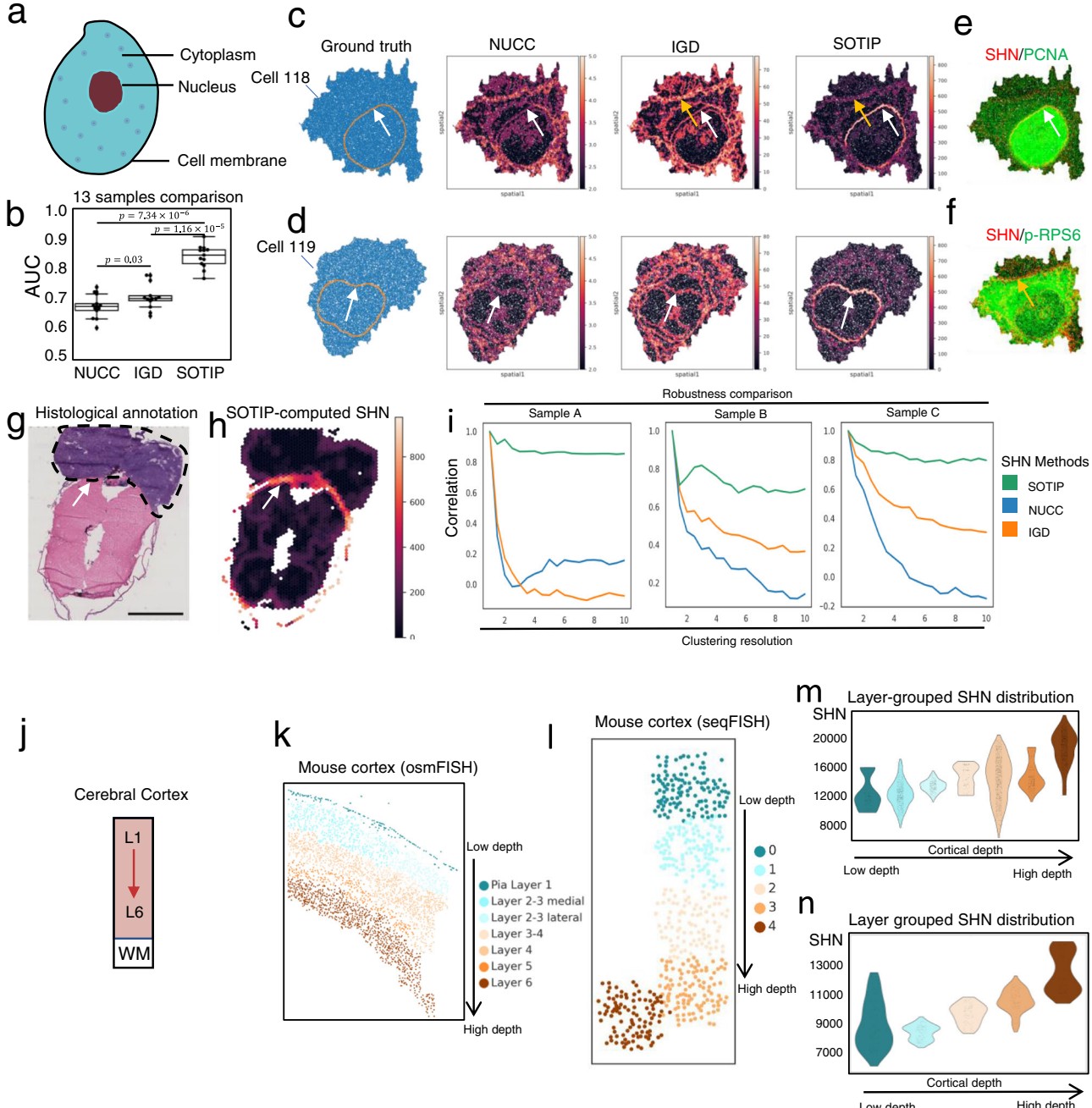

**Fig. 3 | SOTIP accurately quantifies spatial heterogeneity. a** Schematic diagram of a eukaryotic cell. **b** Summary of the SHN quantification performance on all 13 samples. The AUC is used to estimate the performance. Two-sided Wilcoxon rank-sum test. The detailed definition of box plot elements is described in Methods. The boxplots show *n* = 13 biologically independent cells. **c** Visualization of the ground truth (1st column) and SHN values computed by NUCC (2nd column), IGD (3rd column), and SOTIP (4th column) in cell 118. White arrows point out the same position of nuclear envelope. Orange arrow points out a computed potential intracellular membrane structure. **d** Similar with c but for cell 119. **e** Overlay of SOTIP computed SHN values with PCNA protein expression in cell 118. White arrow points out the same position with Fig. 3c. **f** Overlay of clipped SOTIP computed SHN values with p-RPS6 expression in cell 118. Orange arrow points to the same position as in Fig. 3c 4th column. **g** Histological image of the zebrafish data. Tumor region is annotated with black dashed line. Scale bar: 2 mm. **h** 10x Visium spots are paired with Fig. 3g, colored according to SOTIP computed SHN values. **i** The stability

(*y*-axis) of SHN computed using different methods, with different clustering resolutions (*x*-axis). The stability is evaluated by computing the correlations between SHN values under resolution 1 and SHN values under different resolutions. **j** Schematic diagram of the cerebral cortex. **k** Region annotation of the osmFISH mouse cortex dataset. Each point is a cell colored by its layer. **l** 5 FOVs of seqFISH+ cortex data. The layer depth (Layer 2–6) increased with the FOV label (0–4). **m** Distribution of SHN values (*y*-axis) computed using SOTIP in each cortical layer of osmFISH dataset (*x*-axis). Each violin represents a distribution of SHN values (*y*-axis) of MECNs in one layer (*x*-axis), and each point in one violin represents a MECN's SHN value. The violin-plot is colored by the layer label of the MECN. The *x*-axis is arranged in ascending order of the cortical depth. **n** Similar with (**n**) but with seqFISH+ dataset. The gradient pattern of SHN along the cortical layer depth. Violin-plots were colored according to (**m**). Source data are provided as a Source Data file. The experiment results in (**g**, **h**) were similar with three independent repeats.

obtained area under curve (AUC) given ground truth should be high for a good heterogeneity estimation method (see "Methods").

For all 13 cells, we compared SOTIP with NUCC[16,33] and IGD (Supplementary Table 1), the quantitative results (Fig. 3b) showed that SOTIP (median AUC 0.85) significantly outperformed NUCC (median AUC is 0.68, $p = 7.34 \times 10^{-6}$, two-sided Wilcoxon rank-sum test, $n = 13$ cells), and IGD (median AUC is 0.71, $p = 1.16 \times 10^{-5}$, two-sided Wilcoxon rank-sum test, $n = 13$ cells). To dive into details of above methods, we unbiasedly selected two cells as examples to qualitatively compare the computationally delineated nuclear envelopes with the ground truth (Fig. 3c, d). In cell 118, both IGD (AUC = 0.66, Fig. 3c third column) and SOTIP (AUC = 0.77, Fig. 3c fourth column) delineated clear curves with closed pattern, while NUCC (AUC = 0.64, Fig. 3c second column) tended to output blurred curves with irregular and unsmooth pattern. When further comparing IGD and SOTIP, we found that SOTIP exhibited the highest SHN value around the nuclear envelope (white arrows in Fig. 3c fourth column) with strong specificity, but IGD failed to differentiate the relative values of SHN around different regions (white and orange arrows in Fig. 3c third column). Similar results have also been found in cell 119 (Fig. 3d).

To further confirm these results, we inspected the relationship between the spatial localization of SOTIP-computed SHN values and protein enrichment. We implemented this comparison by encoding SHN values into red channel, and targeted protein intensities into green channel, then plotting the encoded colors for each cell. Again we used cell 118 as an example, and found that SOTIP-identified SHN values aligned well with proliferating cell nuclear antigen (PCNA), a known nuclear marker[60] (Fig. 3e). Although SOTIP successfully highlighted the region of the highest heterogeneity within a cell, we additionally asked whether SOTIP also delineated other cellular structures (e.g., orange arrow in Fig. 3c fourth column) with relatively lower heterogeneity. To answer this and to check whether SOTIP could reflect different levels of heterogeneity, we first clipped the highest SHN values, then encoded targeted protein intensities and the clipped SHN values in colors as before (Fig. 3f). It turned out that SOTIP-identified structure aligned well with ribosomal protein S6 (RPS6) (Fig. 3f), which is reported as a marker of endoplasmic reticulum (ER)[60].

To test whether SOTIP can quantify SHN values on tissue samples, we used a spatial transcriptomics dataset containing sections from $BRAF^{V600E}$ melanoma zebrafish model[61]. We firstly applied NUCC, IGD and SOTIP on the sample containing a clear tumor-muscle interface (Fig. 3g). The results showed that SOTIP-computed SHN values delineated the tumor-muscle interface, which was expected to be the most heterogeneous region[61], in accordance with histological annotation (Fig. 3g, h white arrow) with high specificity. On the contrary, both NUCC-computed and IGD-computed SHN values exhibited indistinguishable pattern within tissue bulk and the tumor-muscle interface (Supplementary Fig. 2e, f).

Since SOTIP involved a clustering step (i.e., Leiden[50]) before MEG construction (see "Methods"), we wanted to ask whether it could output stable results for different clustering resolutions. To evaluate the robustness of SOTIP regarding to the clustering resolution, we firstly quantified the SHN values of the zebrafish sample (Fig. 3g, h) by SOTIP with different Leiden clustering resolutions ranging from 1 to 10. Then we computed the Pearson correlation coefficient (Pearson's $r$) between SHN values on resolution 1 with SHN values on different resolutions. Finally we plotted the computed correlation as the function of Leiden resolution (Fig. 3i left). With this procedure, we compared the robustness of SOTIP, NUCC, and IGD, and showed that SOTIP have a substantially stronger robustness than the existing methods, and could output stable SHN values even with extreme over-clustering (Fig. 3i left). The other two samples of the zebrafish[61] study also supported this claim (Fig. 3i middle and right).

## SOTIP reveals gradient spatial heterogeneity along cerebral cortical layer axis

The laminar structure (Fig. 3j) of mammalian cerebral cortex is reported to exhibit layer-specific properties on gene expression, morphology and connectivity in single cell level[62–64]. One study[33] of the recent BRAIN Initiative Cell Census Network (BICCN) program[64] proposed an observation that the composition complexity of cell neighborhoods increased towards deeper layers. They counted the number of different cell clusters within cell neighborhood to define the spatial heterogeneity (like NUCC does). As our aforementioned analysis, the reliability of their statement could be compromised on different clustering resolutions (Fig. 3i). To confirm and refine their conclusion, we applied SOTIP, which was proved robust and accurate in SHN quantification, on other mouse cortex datasets of independent experiment protocols.

To this end, we firstly adopted a single-cell spatial transcriptomics dataset (cyclic-ouroboros single-molecule fluorescence in situ hybridization, osmFISH[46]) which covered the main layers of mouse somatosensory cortex with layer annotation[46]. This dataset contains 5328 single cells across cell-wise annotated cortical layers (Fig. 3k) ranging from pia layer 1 ($n = 159$), layer 2–3 medial ($n = 549$), layer 2–3 lateral ($n = 254$), layer 3–4 ($n = 131$), layer 4 ($n = 1002$), layer 5 ($n = 295$), layer 6 ($n = 1015$), and other regions. We applied SOTIP on this dataset to quantify each ME's SHN value and plotted the distribution of SHN values as violin-plots to display the trends along the cortical depth axis (Fig. 3m). Specifically, each violin-plot is a distribution of SOTIP-computed SHN values within a specific cortical layer (x-axis), and the x-axis conforms to the order of the cortex from shallow (viz. pia layer 1) to deep (viz. Layer 6) layer. The result visually showed a gradient pattern of SHN values from shallow layers to deep (Fig. 3m). To further quantify the ordinal agreement between the SHN values and layer depth, we used both ME-wise and layer-wise Spearman's rank correlation coefficients (Spearman's ρ) between them (see "Methods"). The results showed a strong correlation between the SHN value and the cortical depth, in both manner of ME-wise (Spearman's ρ = 0.674, $p < 1 \times 10^{-15}$) and layer-wise (Spearman's ρ = 0.847, $p = 0.016$).

To strengthen our conclusion, we additionally applied SOTIP on another independent single-cell spatial transcriptomics dataset (evolution of sequential fluorescence in situ hybridization, seqFISH+[17]) of mouse cortex covering different layers[17]. This dataset contains 913 single cells from five spatially continuous field of views (FOVs) ranging from layer 2 to layer 6 (Fig. 3l). We conducted the same SOTIP analysis as with osmFISH dataset, the result also confirmed a continuous gradient pattern of SHN values along the cortical axis (Fig. 3n), and a strong and positive correlation between the SHN value and the layer depth (ME-wise Spearman's ρ = 0.719, $p < 1 \times 10^{-15}$ and layer-wise Spearman's ρ = 1, $p < 1 \times 10^{-15}$). For osmFISH and seqFISH+, the performance of SOTIP were not disrupted by neither different over-clustering levels, nor different neighborhood sizes, being more robust than other methods, e.g., NUCC, IGD (Supplementary Fig. 2a–d).

## SOTIP stratifies known layers in brain tissues

To test the capability of SOTIP in the task of spatial domain identification, we took the SpatialLIBD dataset, which provided 10x Visium (Fig. 4a for illustration) spatial transcriptomics data on 12 tissue sections of human dorsolateral prefrontal cortex (DLPFC)[48]. For each tissue section, the original research[48] also provided spot-level annotations of cortical layers as well as white matter regions, which can be used as ground truth to numerally evaluate performance. Since other methods[36–38] have already demonstrated the improved performance of integrating spatial information by comparing with non-spatial clustering algorithms, e.g., k-means, Louvain[50], and SC3[65], on this dataset, we compared SOTIP only with the spatial methods, including stLearn[38], BayesSpace[36], and SpaGCN[37], Giotto[66], SEDR[39],

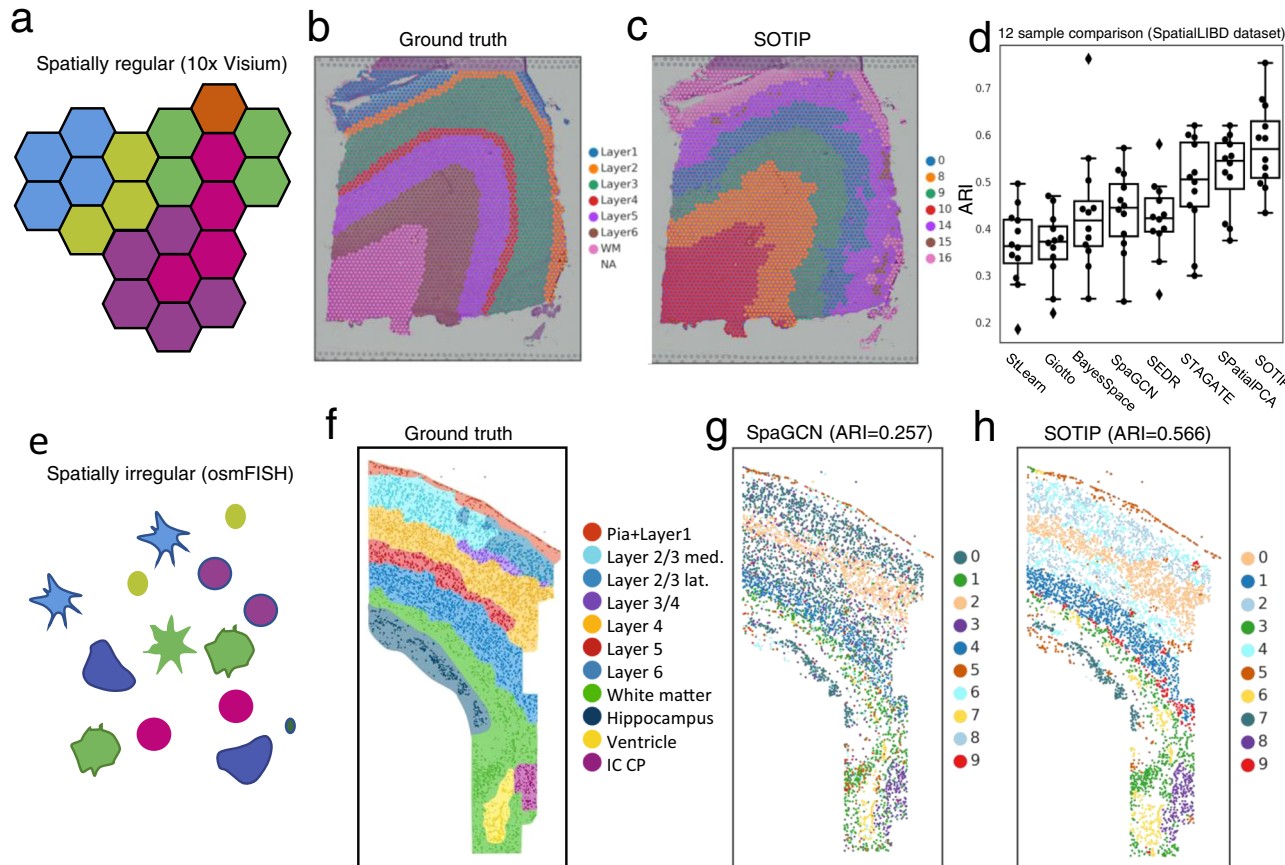

**Fig. 4 | SOTIP stratifies known layers in brain tissues. a–d** Human dorsolateral prefrontal cortex dataset with 10x Visium. **a** Spatial arrangement illustration for typical 10x Visium dataset. **b** Ground truth annotation for slice 151673. **c** Spatial domain detected by SOTIP. The labels are not consecutive because SOTIP follows a hierarchical merging scheme. **d** Boxplots summarize the detection accuracy (*y*-axis) of four methods (*x*-axis) in all 12 samples. The definition of each box can be found in Methods. **e–h** mouse somatosensory cortex dataset with osmFISH. **e** Spatial arrangement illustration for typical osmFISH dataset. **f** Ground truth of regional annotation for the osmFISH sample. **g, h** Spatial domains identified by SpaGCN (**g**) and SOTIP (**h**). Source data are provided as a Source Data file.

STAGATE[40], and SpatialPCA[41] (Supplementary Table 1). Following the comparison methodology of previous studies[36,37,48], we input to the algorithms the number of clusters, and used adjusted rand index (ARI) to quantify the similarity between the partition results and manual annotations.

We took one of the representative tissue sample, sample 151673 (*n* = 1639 spots, shown in Fig. 4b) for an example. Visually, SOTIP identified SDM exhibited a layered pattern consistent with the ground truth (Fig. 4c). Since it's difficult to reproduce the original ARI (also reported by other group[39]), especially for deep learning based methods[37–39], we directly used the ARI and clustering results as claimed in the original paper. When we applied these algorithms to all 12 tissue sections, we found that SOTIP outperformed all other methods (Fig. 4d, *n* = 12) in terms of mean and median. Specifically, SpaGCN (mean/median: 0.43/0.45) and BayesSpace (mean/median: 0.43/0.42) are two best peer-reviewed methods, have higher ARI than older methods, e.g., stLearn (mean/median: 0.36/0.36), and Giotto (mean/median: 0.36/0.37), while lower than STAGATE (mean/median: 0.49/0.51) and SpatialPCA (mean/median: 0.52/0.55), the two best pre-printed methods. SOTIP (mean/median: 0.58/0.57) performed best than all of them.

For the compatibility, BayesSpace was restricted to the lattice-shaped spatial data like 10x Visium (Fig. 4a), while SOTIP and SpaGCN does not rely on the spatial arrangement of spots/cells. To demonstrate whether SOTIP surpass other methods on non-lattice-shaped spatial transcriptomics data (Fig. 4e), we examined the two algorithms on a single-cell spatial transcriptomics dataset (osmFISH[46]) of the

mouse brain somatosensory cortex. We evaluated the performance with ARI taking the known cell-wise region annotations as the ground truth (Fig. 4f). The result showed that SOTIP (ARI = 0.566, Fig. 4h) substantially outperformed SpaGCN (ARI = 0.257, Fig. 4g).

In addition to all the above advantages, SOTIP is suitable for spatial proteomics, which is hardly manageable for any existing methods. We will further demonstrate SOTIP utility and performance with spatial proteomics technologies in the next section.

## SOTIP characterizes spatial and molecular tumor-immune organization

After demonstrating the superior performance of SOTIP on various spatial transcriptomics datasets, we next ask whether SOTIP is also compatible with spatial proteomics datasets. Since there are currently few methods that was explicitly applied to perform spatial domain identification on spatial proteomics data, we only compared SpaGCN with SOTIP. We found two spatial proteomics datasets, of which the first was produced by MIBI-TOF technique on 41 triple negative breast cancer (TNBC) patients[12], and the second was produced by scMEP technique on two colorectal carcinoma (CRC) patients and two healthy controls[49]. With these datasets, we attempted to identify the tumor-immune interplay (Fig. 5a for illustration), which has been widely regarded as a prognostic indicator in tumor progression[67–69].

As the original papers provided tumor/immune partition annotations for 2 samples in CRC (Fig. 5d), and two samples in TNBC (Fig. 5g), we used these four samples to evaluate the performance of different methods on stratifying tumor and immune regions. For

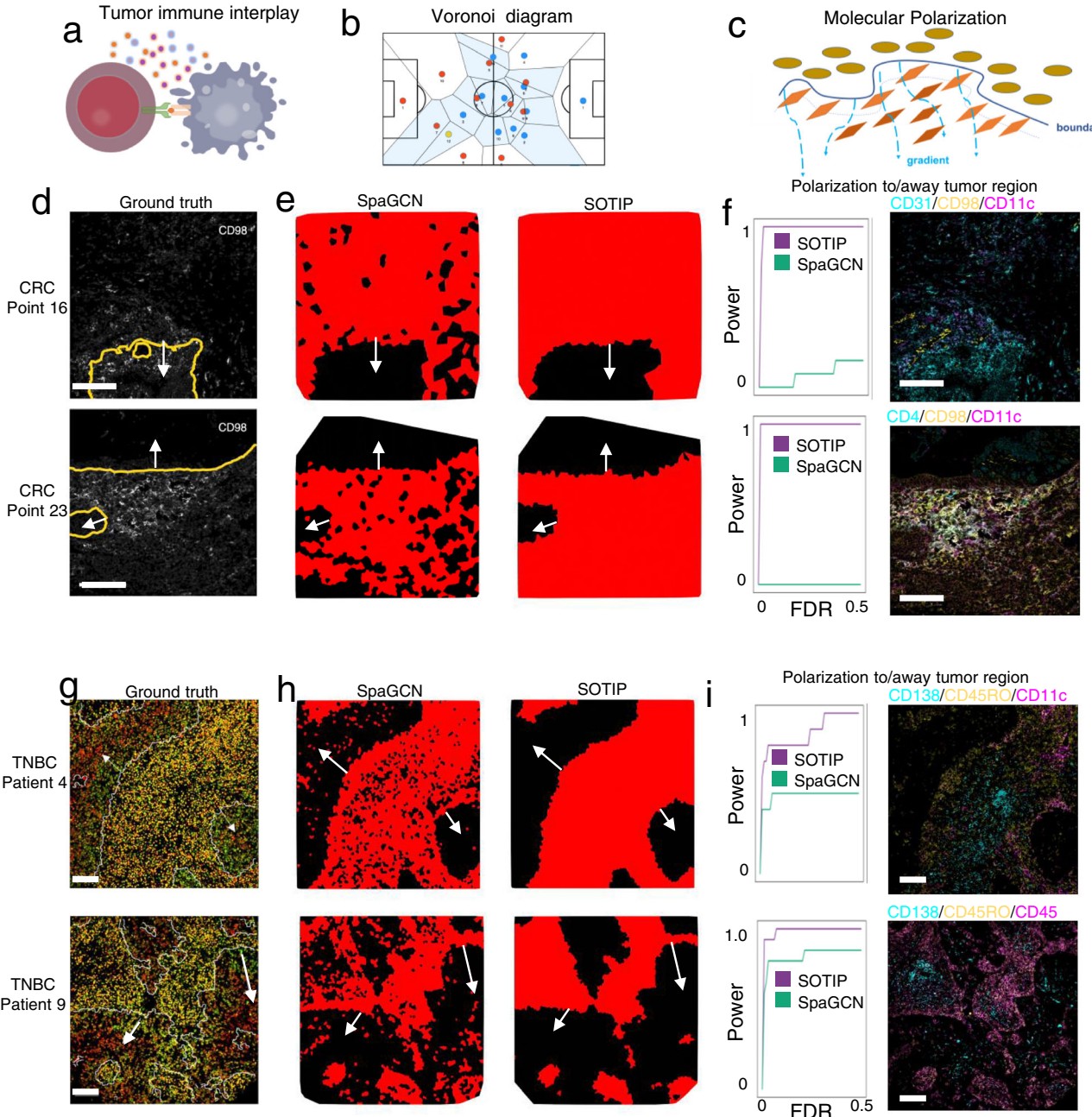

**Fig. 5 | SOTIP applications on two spatial proteomics datasets. a–c** Schematic diagram to show how SOTIP works spatial proteomics datasets. Illustration of tumor-immune interplay (**a**), Voronoi diagram used to visualize spatial domains (**b**), and molecular polarization to tissue boundary (**c**). **a** Tumor and immune cells have complex interplay around the tumor-immune boundary. **b** Voronoi diagram (as implemented in https://stackoverflow.com/questions/20515554/colorize-voronoi-diagram/20678647#20678647%203.18.2019) is a visualization method for spatial omics data[29]. **c** Molecular polarization is a phenomenon that some molecules exhibit gradient changes perpendicular to tissue boundaries. **d** Ground truth of tumor-immune boundary from the original paper of CRC samples. Top panel: point 16. Bottom panel: point 23. Edited from original paper. Scale bar: 100 μm. **e** Voronoi diagram of SpaGCN-detected (left column), and SOTIP-detected (right column) spatial domains for point 16 (top row) and point 23 (bottom row). **f** Comparison of SpaGCN-detected and SOTIP-detected spatial domains to find polarized proteins

to/away tumor region, in CRC point 16 (top row) and CRC point 23 (bottom row). Left column: The power plots show the proportion of true positives (y axis) detected by different methods at a range of FDRs (x axis) for point 16 (top row) and point 23 (bottom row). Right column: The overlay of three representative polarized proteins found by SOTIP-detected SDM in point 16 (top row) and point 23 (bottom row). Scale bar: 100 μm. **g** Ground truth of tumor-immune boundary from the original paper of TNBC samples. Top panel: patient 4. Bottom panel: patient 9. Edited from original paper. Scale bar: 100 μm. **h** Voronoi diagram of SpaGCN-detected (left column), and SOTIP-detected (right column) spatial domains for patient 4 (top row) and patient 9 (bottom row). **i** Similar with (**f**) but on TNBC patient 4 (top row) and patient 9 (bottom row). In **d**, **e**, **g**, **h**, all white arrows point to tumor regions. Scale bar: 100um. Source data are provided as a Source Data file. The experiment results in (**d**, **f**, **g**, **i**) were similar with three independent repeats.

better visualization and qualitative comparison[29], we plotted the detected spatial domains using Voronoi diagrams (Fig. 5b for illustration). The results (Fig. 5e, h) showed that in all datasets, SOTIP achieved better results in terms of visual coherence compared with

SpaGCN, and almost perfectly matched with the originally reported region annotations (Fig. 5d, g).

Tumor and immune cells are reported to perform mutual modulation by frequent cross-talk within the tumor-immune ecosystem[70].

Their intercellular communications and intracellular regulation networks generate patterned phenotypic cell identities and molecular distribution[71]. Molecular polarization (a form of spatial molecular distribution pattern that certain molecule tends to differentially enriched distal/proximal to a tissue landmark) was recently reported to appear nearby pathological tissue boundaries[7,12,61] (Fig. 5c for illustration). To show that SOTIP-detected tumor/immune regions better fit for downstream analysis, we next inspected the relationship between molecular polarization and tissue architecture. To be clear, in the following statements, when we say that a protein is polarized to/away a region, we mean that the protein tends to distributed proximally/distally to that region.

In the CRC datasets, we first classified immune cells as distal and proximal to tumor-immune boundary (see "Methods", Supplementary Fig. 4a, b) using SOTIP and SpaGCN results, respectively. Next, Wilcoxon rank sum test was adopted to estimate the protein enrichment/depletion with respect to the boundary, and polarized proteins with statistical significance were detected. We measured the detection power on the basis of false-discovery rate (FDR) in both datasets to conduct a fair comparison across methods. The results showed that SOTIP was more powerful than SpaGCN across a range of FDR cutoffs (Fig. 5f left). We also performed similar analysis on TNBC datasets (see "Methods", Supplementary Fig. 4c, d), to identify molecular polarization to/away tumor (Fig. 5i left) and immune regions (Supplementary Fig. 4e–g), confirming the superior detection power of SOTIP.

Last, we showed representative (see Methods) polarized proteins detected by SOTIP in CRC (Fig. 5f right), and in TNBC (Fig. 5i right, Supplementary Fig. 4e–g) samples. Apart from consistency with the original paper (e.g., CD98, CD11c polarization in CRC immune region[49], and CD45RO, CD11c polarization in TNBC immune region[12], and Keratin6, Keratin17, HLA-DR polarization in TNBC tumor region[12]), we also detected some unexpected polarizations. For example, in TNBC patient 9, we found that CD45 was significantly polarized to the tumor region ($p = 6.88 \times 10^{-44}$, two-sided Wilcoxon rank sum test, Fig. 5i bottom right), which was also uncovered in patient 4 ($p = 1.30 \times 10^{-9}$, two-sided Wilcoxon rank sum test). CD45 plays an significant role in T cell receptor signaling pathway[72], while CD45RO, an isoform of CD45, is of vital importance for the production of TNF-α and IFN-γ[73]. TNF-α and IFN-γ are powerful in monitoring tumor proliferation and trigger cell death[74]. TNF-α involves in complex immune response to induce tumor cell apoptosis and destroy the blood vessels which supports the tumor growth by transporting essential nutrients[75]. IFN-γ, a member of the type II interferon group, possibly contributes to the tumor cell apoptosis via JAK-STAT pathway in a microenvironment-dependent manner[76,77]. The enrichment of CD45 and CD45RO around the tumor-immune boundaries (Fig. 5i), which was also described in the original paper, indicated that they may function during the battle of immune tissue against the tumor tissue via multiple apoptosis-associated immune signaling pathways.

In both TNBC patient 4 and 9, CD138 has significantly polarized away the tumor-immune boundary ($p = 5.12 \times 10^{-27}$ in patient 4 and $p = 7.17 \times 10^{-52}$ in patient 9, two-sided Wilcoxon rank sum test, Fig. 5i top right and bottom right). CD138, also known as heparan sulfate (HS) proteoglycan Syndecan-1, is a key regulator responsible for inflammatory cytokines modulation[78,79]. Cytokines have been found to play anti-tumor effects by binding to the receptors to mediate cell-cell communications, supported by both animal studies and clinical researches[80]. Specifically, CD138 could upgrade the survival of mature plasma cells, which is critical for long-term humoral immune[81]. This function is in accordance with our observation that CD138 expression is highly enriched in the core of immune cell cluster (Fig. 5i). It indicates that activated B cells may play an active anti-cancer role in a persistent way with recruitment of other immune cells.

## SOTIP recovers known differential microenvironments in cirrhotic liver

The last task of SOTIP is differential microenvironment analysis (DMA), which aims to identify microenvironments between conditions (e.g. disease status, drug treatment, and experimental perturbation) with high specificity. This task is in analogy with the differential abundance analysis task[54–56] in single cell analysis, but unavailable in spatial omics analysis. To demonstrate the utility of SOTIP on this task, and as a sanity check, we firstly applied DMA in a spatial metabolomics dataset including two samples from healthy and fibrotic liver (Fig. 6a for illustration). They provided paired H&E staining, which could be taken as a positive control for our detection result (Fig. 6b). The healthy liver sample (Fig. 6b left) was a section from a standard liver lobule, which is a repeating hexagonal-shaped units of liver. The fibrotic sample (Fig. 6b right) was a liver section containing both regions of fibrotic niche (red arrow) and hepatic tissue bulk. The original paper provided cell type annotations (Fig. 6c). In this case, we expected that SOTIP could spot the fibrotic-specific microenvironments, e.g., the fibrotic region and tissue boundary.

To investigate on that, we firstly built an MEG of the joint MECNs from both samples, resulting in total 1608 nodes and 237,604 edges (Supplementary Fig. 5). The complete high-resolution graph was too complicated to read and interpreted, so we next detected communities in this graph and analyzed the abstracted topological structure (see "Methods"). The resulted MEG abstraction (a simplified version of MEG, Fig. 6d) consisted of 16 nodes, each of which is an MECN cluster (a group of similar MECNs). The edge width can be considered as similarity between nodes. To get a global view of the spatial organization of various microenvironments within the tissue field, we plotted the clustered MECN groups on physical space with Voronoi diagram (Fig. 6e), so that MECNs from the same cluster share the same color according to the MEG abstraction (Fig. 6d). From the MECN cluster map (Fig. 6e), we observed that certain microenvironments showed consistency with the histological images (Fig. 6b right).

To select the MECN clusters with high sample-specificity and cell type complexity, we plotted each MECN cluster on an EH plot (abbreviation for "entropy-of-ME-cluster (EMC)" versus "spatial heterogeneity (SHN)" plot, see "Methods"). On the EH plot, MECN clusters with higher sample-specificity corresponded to lower EMC (y-axis), and MECN clusters with higher compositional complexity corresponded to higher SHN value (x-axis), so that MECN clusters at the bottom right corner should be of special interest (Fig. 6f). Based on this criterion, we selected 7 MECN clusters, and all of them were specifically occurred in the fibrotic liver sample. We first looked at MECNs with highest cell composition complexity, i.e., the MECN cluster 15 (Fig. 6f, i). The hepatocytes account for more than half of the cell counts, followed by all other major cell types, fibroblast, Kupffer cell, immune cell, and endothelial cell. This high level of SHN has been expected, since the metabolomic profile of hepatocytes is reported to exhibit substantial difference with other non-hepatic parenchymal cells (NPCs)[2,3,7]. Given that MECN cluster 0, 9, 11, 12, 14 were densely connected in the MEG abstraction (Fig. 6d), we jointly plotted them in a single MECN cluster map (Fig. 6g left). The result indicated that these MECNs displayed fibrotic-niche phenotype in cell type composition (Fig. 6g right) and also exhibited high consistency with the histological information (right arrows in Fig. 6b, g right) in spatial localization. We also investigate MECN cluster 3 (Fig. 6h), whose cell type composition (Fig. 6h bottom) showed a balanced mixture of the fibrotic niche and the hepatic tissue bulk. Its spatial distribution confirmed this composition by approximately delineating the hepatic-fibrotic tissue boundary (green dashed line in Fig. 6b right and Fig. 6h). In summary, these analyses demonstrate the capability of SOTIP to identify samples associated microenvironments with high specificity, and interrogate the relationship between the phenotypic and spatial features of them.

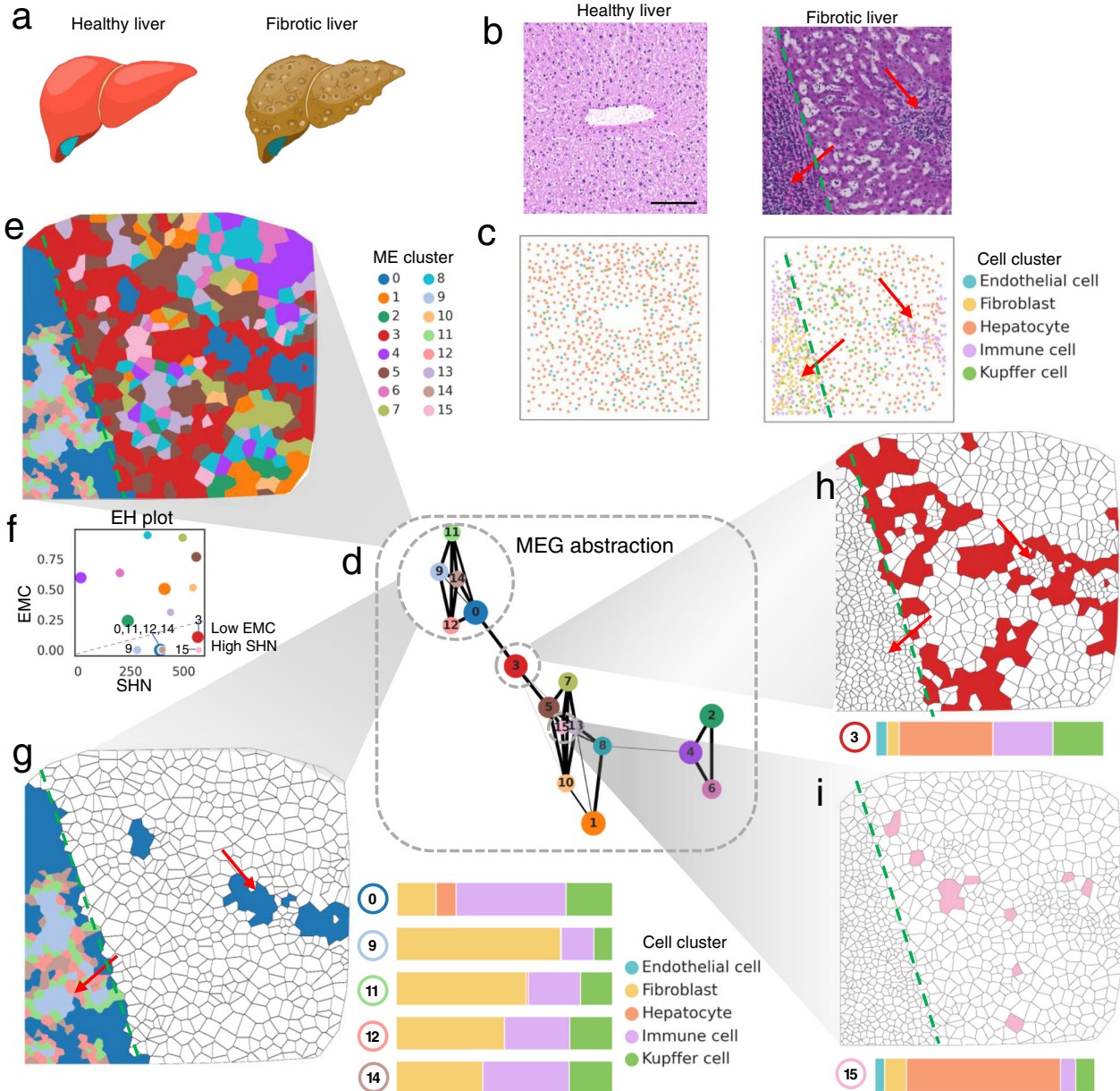

**Fig. 6 | SOTIP recovers differential microenvironments between liver samples.**
**a** Illustration of healthy and fibrotic livers. **b** Histological images of the healthy (left) and fibrotic (right) liver samples. The red arrows indicate fibrotic niche. The green dashed line indicates the fibrotic boundary. Scale bar: 100 μm.
**c** Spatial single cell map in healthy (left) and fibrotic (right) liver sample paired with Fig. 6b. The red arrows and green dashed line are consistent with those in Fig. 6b. **d** MEG abstraction. Each node is a cluster of densely connected MECNs. The width of edges can be considered as similarity between nodes (see "Methods"). Each node (i.e., MECN cluster) is assigned an unique color. The size of each node is proportional to the number of MECNs belonging to that MECN cluster. **e** MECN clusters of the fibrotic liver sample showed in Voronoi diagram. The colors of MECN clusters are consistent with those in Fig. 6d. The green dashed line is consistent with that in Fig. 6b. **f** Entropy-SHN (EH) plot

(see "Methods"). Each node is a MECN cluster with the same color scheme as Fig. 6d, e. The y-axis is the entropy ($H_C$) of the MECN cluster $C$. The x-axis is the average SHN values of the MECN cluster. MECN cluster 0, 3, 9, 11, 12, 14, 15 with low entropy and high SHN values lie at the right bottom corner of EH plot. **g** Left: The same MECN clusters as Fig. 6e, but only MECN cluster 0, 9, 11, 12, 14 in highlight. The red arrows and green dashed line are consistent with those in Fig. 6b. Right: Cell composition of MECN cluster 0, 9, 11, 12, 14, with the same color scheme as Fig. 6c. **h** Top: The same MECN cluster map as Fig. 6e, but only MECN cluster 3 in highlight. The red arrows and green dashed line are consistent with those in Fig. 6b. Bottom: Cell composition of MECN cluster 3, with the same color scheme as Fig. 6c. **i** Similar with Fig. 6h but using MECN cluster 15. Source data are provided as a Source Data file. The experiment results in (**b**, **c**, **d**) were similar with three independent repeats.

## SOTIP identifies highly specific MECNs associated with prognosis in subtypes of TNBC

Triple-negative breast cancer (TNBC), defined by lack of therapeutic targets (estrogen receptor, progesterone receptor, and Her2), is considered to be a form of breast cancer with more aggression and poorer prognosis[82]. Keren et al.[12] applied multiplexed ion beam imaging by time-of-flight (MIBI-TOF) to generate a 36-plex spatial single cell

proteomics dataset from 41 TNBC patients. In that paper, the researchers classified TNBC patients into three archetypical subtypes (viz., *cold*, *mixed*, and *compartmentalized*), by quantifying the mixture degree of tumor and immune cells[12]. The dataset hasn't been fully mined since the original paper mainly concentrated their analysis on protein co-expression and basic spatial statistics[12], we here focused on the microenvironment to draw a spatial organization of the tumor-

immune landscape and to inspect interesting microenvironment with strong specificity.

Since the "cold" samples, lack of immune infiltration, can be easily differentiated from other two subtypes. To demonstrate SOTIP's capability, we choose the more challenging case to identify differential MECNs between *mixed* (immune cells mixed with tumor cells)[12] and *compartmentalized* (immune cells spatially separated from tumor cells)[12] samples. The challenge to distinguish the *mixed* from the *compartmentalized* came from the fact that these two subtypes of TNBC samples typically contained similar cell type compositions[12], such that they could not be easily identified by current single-cell based analysis[54,56], or protein co-expression analysis[67]. To this end, we chose the most representative samples, suggested by the original research[12], from the two subtypes of TNBC, i.e., patient 4 for *compartmentalized* and patient 12 for *mixed* (Fig. 7a, b). As with previous section, we firstly built an MEG of the joint MECNs from both samples (Supplementary Fig. 6a), then generated an MEG abstraction (Fig. 7c), consisting of 29 MECN clusters as nodes and the corresponding edges as similarities between MECN clusters. We further plotted the MECN cluster map for the two samples (Fig. 7d for *compartmentalized* and Fig. 7e for *mixed*), and selected the MECN clusters with high SHN and low EMC on EH plot (Fig. 7f). Interestingly, we identified several pronounced MECN clusters, of which MECN cluster 9 and MECN cluster 24 were strictly enriched in specific samples with disappeared EMC.

MECN cluster 9 is specifically restricted in the *compartmentalized* sample. To inspect the cellular complexity of MECN cluster 9, we found that its SHN value was fairly low, since it only consisted of two subtypes of cells, tumor cell (Tumor cluster) and its Keratin (pan-Keratin, Keratin6, Keratin17, Supplementary Fig. 6b) positive counterpart (Keratin + tumor cluster). Since there lacks cells of Tumor cluster in *mixed* sample (Supplementary Fig. 6c, Fig. 7g, h), it's not surprising to detect this MECN differential between samples. More interestingly, MECN cluster 24 is specifically restricted in the *mixed* sample (Fig. 7i), and is composed of a high percentage of keratin+ tumor cells, as well as a mixture of CD8 T cells and macrophages (Fig. 7i bottom). We noticed that all the three cell types occupied a substantial proportion within both *mixed* and *compartmentalized* samples (Supplementary Fig. 6c), so we reasoned that these three cell types may distribute exclusively in the *compartmentalized* sample but closely in the *mixed* sample, which was confirmed by the spatial relationships of them (Fig. 7j).

We also looked at other differential MECNs (with non-zero EMCs, but still differentiated between samples by occurrence (with EMC cutoff of 0.5)), and found that they were mostly derivative from MECN cluster 9, and 24 (Supplementary Fig. 6d–g). Specifically, for the MECN clusters mainly occurred in the *compartmentalized* sample (Supplementary Fig. 6d, e), they were dominated by the same two cell types (i.e. Tumor and Keratin+ tumor) as MECN cluster 9. Correspondingly, for those MECN clusters mainly occurring in the *mixed* samples (Supplementary Fig. 6f, g), they were dominated by the same three cell types (i.e., CD8 T, macrophage, and Keratin+ tumor) as MECN cluster 24, except for a rather minor proportion of CD4 T cells. This indicated that MECN cluster 9 (Tumor and Keratin+ tumor) and/or 24 (CD8 T, macrophage, and Keratin+ tumor) may be the main driver for the differentiation of *mixed* and *compartmentalized* forms of TNBC.

To examine this statement, we evaluated the occurrence of MECN cluster 9 and 24 in all *compartmentalized* ($n = 15$) and *mixed* ($n = 19$) patients, and compared the occurrence score of these MECNs between two patient groups (see "Methods"). The results showed that the occurrence score of MECN cluster 9 (Tumor and Keratin+ tumor) displayed more consistency in *mixed* samples than in *compartmentalized* samples, with lower variance (Fig. 7k), but there is no significant difference of the occurrence of MECN cluster 9 between the *mixed* and *compartmentalized* groups ($p = 0.986$, two-sided Wilcoxon rank-sum test, Fig. 7k). On the contrary, MECN cluster 24 (macrophage, and Keratin+ tumor, CD8 T) significantly showed more occurrence in

*mixed* samples than in *compartmentalized* samples ($p = 9.20 \times 10^{-6}$, one-sided Wilcoxon rank-sum test, Fig. 7l). This indicated that MECN cluster 24 (macrophage, and Keratin+ tumor, CD8 T) might be used as a clinical indicator, so we termed MECN cluster 24 as MKT.

To investigate the relevance between the occurrence of MKT and prognosis, we partitioned the patients according to the occurrence score of MKT. Survival analysis showed that patients with lower occurrence of MKT was associated with better survival outcomes (Fig. 7m), regardless of the specific threshold used to separate two groups of patients (Supplementary Fig. 7). Note that the MKT is a kind of microenvironment consisting of triple-cell-type-interactions, it could not be identified by existing spatial interaction analysis, which focused on double-cell-type interactions.

## Discussion

In this study, we presented SOTIP, a unified framework to perform multiple important tasks with various spatial omics technologies. The core of SOTIP is the construction of MECN graph (MEG), of which each node represents an MECN and each edge encodes the relationship between MEs. Based on MEG, spatial heterogeneity (SHN) quantification, spatial domain (SDM) identification, and differential microenvironment (DME) analysis can be performed. Note that the second task is essentially a clustering task, we named it with "SDM identification" to keep consensus with other related methods (for example SpaGCN[37] and STAGATE[83]), so that readers may not be confused when reading and comparing them.

We conducted simulation experiments to demonstrate the utility of SOTIP, and compared the performance with different methods. We also performed exhaustive comparisons between SOTIP and state-of-art methods on different spatially resolved datasets. In the task of SHN quantification, we benchmarked SOTIP against NUCC and IGD by delineating the nuclear envelope in HeLa cell line (spatial proteomics, 4i[23]) and recovering the gradient heterogeneity pattern in mouse brain cortex (spatial transcriptomics, osmFISH[46]). In the task of SDM identification, we firstly showed SOTIP's superior performance compared with other spatial clustering algorithms on the commonly used SpatialLIBD dataset of human brain cortex[48] (spatial transcriptomics, 10x Visium). On another spatial transcriptomics dataset of mouse brain cortex[46], SOTIP also outperformed others in terms of ARI. We validated the compatibility of SOTIP on other spatial omics data by collecting samples from CRC (spatial proteomics, scMEP[49]) and TNBC (spatial proteomics, MIBI[12]). Although SpaGCN is also extendable to spatial proteomics dataset, SOTIP showed better performance when comparing with the ground truth in multiple cases (Fig. 4). Based on SOTIP detected tumor-immune boundary, we also detected proteins with spatial polarization. In DME analysis, we firstly compared a fibrotic liver sample with healthy liver sample[7], then identified MECNs specifically enriched in the fibrotic sample. The spatial localization and cellular composition of the detected MECN is consistent with the histology image. We finally applied SOTIP to compare the MECN differences between *compartmentalized* and *mixed* subtypes of triple negative breast cancer (TNBC), consistent with previously reported tumor-immune interactions and revealing possible theory to explain the driven factor for the differentiation of these two subtypes.

Apart from the aforementioned performance advantages and discoveries, there are other several points that make SOTIP a distinct method from others. As for interpretability, deep learning-based algorithms are typically considered as black boxes and hard to interpret the result and parameters. On the contrary, each component of SOTIP can be easily mapped back to a biological entity, which is beneficial for model diagnosis and biological interpretation. As for clustering procedure, current methods[36–38] need multiple runs to search for the optimal result when the true number of clusters is not accessible, while SOTIP maintains the intermediate result for every level during the one-time hierarchical merging process. As for scalability,

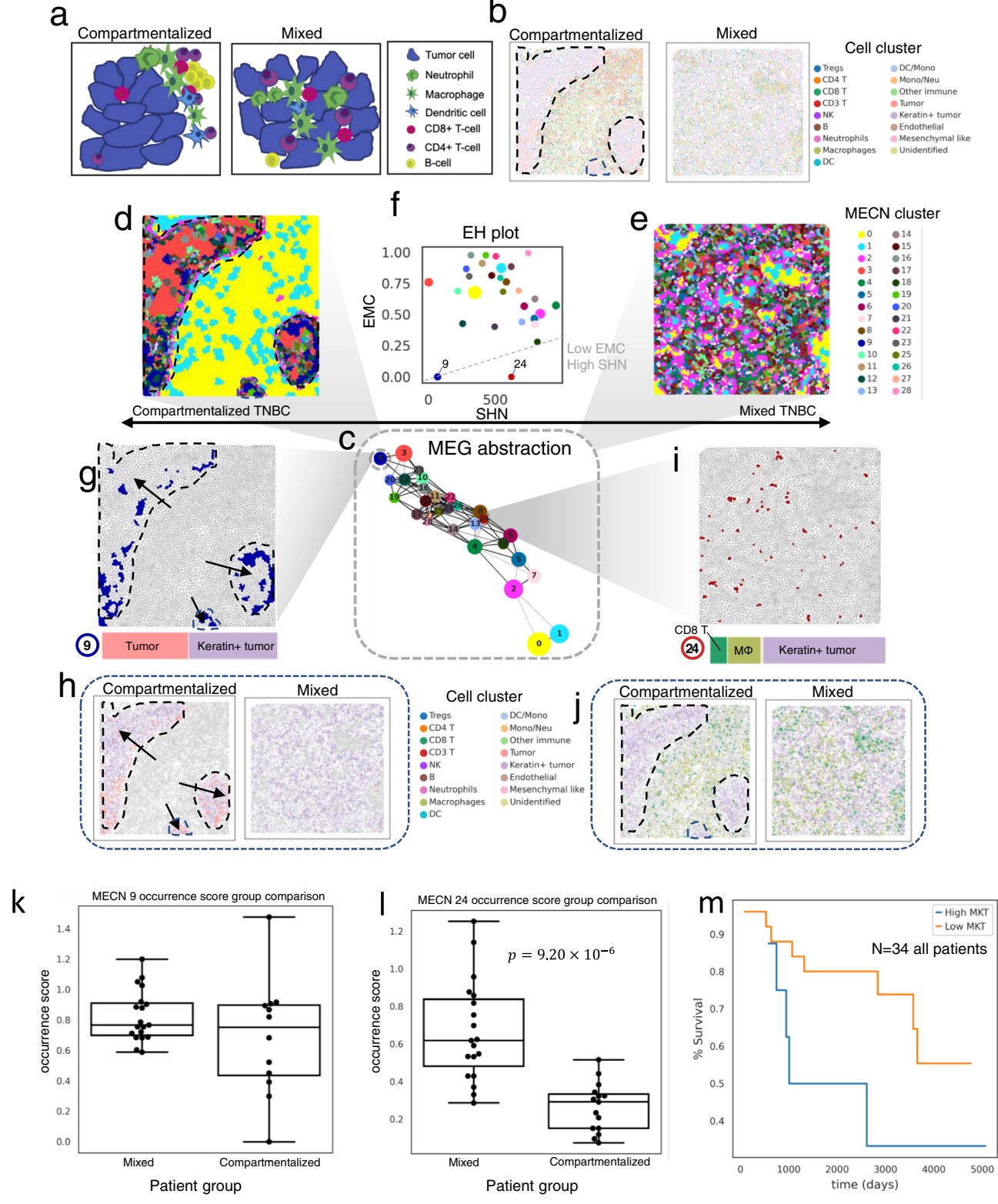

unlike methods whose applications may be limited to certain range of modalities, SOTIP can be freely generalized to different spatial techniques.

The core of SOTIP, MEG, was designed to only require distance/similarity matrix, without full accessibility of the coordination of the two spaces (i.e., physical space and molecular expression space). Several variants of SOTIP can be quickly implemented with different configurations of the two spaces. For example, SOTIP can be used to

characterize microenvironment of other spatial coordination, for example 3D (given 3D spatially resolved techniques), and virtually inferred spatial affinity[84,85]. For another example, SOTIP can use the physical space of the high resolution histological images and the molecule expression space of the high throughput scRNA-seq, to integrate their benefits. With the most advanced multi-modal biotechnologies[86,87], which can obtain multi-omics measurements within single cells, the physical space of SOTIP can be even replaced

**Fig. 7 | SOTIP identifies differential microenvironments between subtypes of TNBC. a** Illustration of *compartmentalized* and *mixed* subtypes of TNBC. **b** Spatial single cell map in *compartmentalized* (left) and *mixed* (right) TNBC samples. The black dashed lines in Fig. 7b, d, g, h, j annotate tumor regions according to the original paper. **c** MEG abstraction. Each node is a cluster of densely connected MECNs. The width of edges can be considered as similarity between nodes (see Methods). The size of each node is proportional to the number of MECNs. **d, e** MECN clusters of the *compartmentalized* (**d**) and *mixed* (**e**) sample shown in Voronoi diagram. **f** Entropy-SHN (EH) plot (see "Methods"). Each node is a MECN cluster with the same color scheme as Fig. 7d, e. The *y*-axis is the entropy ($H_C$) of the MECN cluster *C*. The *x*-axis is the average SHN values of the MECN cluster. **g** Top: The same MECN cluster as Fig. 7d, but only MECN cluster 9 in highlight. The black arrows point to the tumor region as in Fig. 5g top. Bottom: Cell composition of MECN cluster 9, with the same color scheme as Fig. 7d, e. **h** The same single cell

cluster map with Fig. 7b, but only highlights Tumor and Keratin+ tumor clusters. **i** Top: The same MECN cluster map with Fig. 7d, but only highlights MECN cluster 24. Bottom: Cell composition of MECN cluster 24, with the same color scheme with Fig. 7d, e. **j** The same single cell cluster map with Fig. 7b, but only highlights CD8 T, Macrophage and Keratin+ tumor clusters. All three clusters occupy a substantial abundance in both samples. **k** Occurrence score (*y*-axis) of MECN 9 comparison between Mixed and Compartmentalized patients (*x*-axis). Two-sided Wilcoxon rank-sum test, no significance. Boxplots are defined in "Methods". *N* = 34 independent patients. **l** Occurrence score (*y*-axis) of MECN 24 comparison between Mixed and Compartmentalized patients (*x*-axis). One-sided Wilcoxon rank-sum test. Boxplots are defined in "Methods". *N* = 34 independent patients. **m** Survival analysis with Kaplan–Meier curves shows survival as a function of time (days) for patients between two groups (high MKT vs low MKT). Source data are provided as a Source Data file.

with protein expression, chromatin accessibility or metabolites profiles.

One limitation is that SOTIP defines MECNs with predefined shape and size, which might lead to false negative discoveries. The high complexity and diversity of microenvironments in different tissue states and disease progressions make it intricate to fully understand the mechanistic properties, for example, the shapes and sizes of MEs, while SOTIP's rigid manner of MECN definition reduces the searching space of MEs, since those MEs falling out of the beforehand definition (e.g., MEs with non-spherical shapes) could be overlooked. This problem could especially stand out when the real applications need investigations of finer resolutions, for example when comparing the MECN differences between an early stage of disease (with fairly subtle alterations) and the healthy control. To alleviate this problem, two lines of methods can be considered. One is to formalize the geometric and connectomic properties as priors to be incorporated into MECN modeling. This approach might be hindered by the divergence of MEs. Another one is to expend the searching space by considering as many shapes and sizes and combinations of MEs. This approach might be troubled by computational efficiency, and supervisions need to be properly used to guide the searching process. We believe that methods belonging to this niche is essential for future research.

## Methods
### MEG construction
SOTIP takes spatial omics data (gene expression/protein/metabolomic profiles of spots/cells/pixels) as input. For ease of explanation, we will use spatial single cell transcriptomics data (formed as gene expression matrix and 2D spatial coordinates for each cell) to illustrate the method, which can be extended to other spatial omics modalities without loss of generality.

The core of SOTIP is the definition of the MECN graph (MEG). With the MEG, the relationships between every pair of MECNs are encoded by considering both the occurrence, and the gene expression dissimilarities of constituted single cell identities. In this way, gene expressions, single cells, microenvironments, and tissue regions could be linked in a biological meaningful way.

To describe MEG, its nodes and edges should be defined. Specifically, each node is defined as an MECN, which is a bag of cells encoded by cell clusters within the spatial neighborhood. Then each MECN is represented by counting the frequency of each cell cluster (generated by user-defined clustering algorithm, e.g., Leiden) within the MECN, resulting a histogram. In this way, for a spatial omics dataset of a tissue sample, the number of MECNs is the same as the number of cells, and every MECN are represented by a vector with the length of number of cell clusters.

To define the edge between two nodes in MEG, SOTIP computes earth mover's distance (EMD)[88] between them. Considering the cell discrepancy within high-dimensional gene expression space, we derive a connectivity guided minimum graph distance (CGMGD) as the

ground distance which approximates the distance in the transcriptome state space. We elaborate CGMGD 's better performance compared with other choices in "Connectivity guided minimum graph distance (CGMGD)" of the "Methods" section. With simulation data, we also demonstrate the better quality of MEG constructed based on CGMGD than based on an isotropic ground distance (IGD), which assumes equal distances among cell groups (Fig. 2c). After computing the pairwise distance, SOTIP estimates the connectivity between nodes using an efficient neighbor search similar with UMAP[32,89], and the MEG is then constructed.

We next formally define MEG and associated terms. Different with general application of earth mover's distance, in the same biological sample, SOTIP represents every MECNs using single cells with the same configuration of clusters. In a spatial omics sample with *n* cells and *t* cell clusters, we can formulize the definition of *MECN* as a histogram: $MECN_p = \{(T_i, w^p_{T_i})\}, T_i \in [1, t], p \in [1, n]$. $T_i$ enumerates all possible cell cluster labels in the tissue sample, and $w^p_{T_i}$ is the total-normalized count of cluster $T_i$ in $MECN_p$. The distance between two MECNs, e.g. $MECN_p$ and $MECN_q$ is defined as (1):

$$dist\left(MECN_p, MECN_q\right) = \frac{\sum_{i=1}^{t}\sum_{j=1}^{t} f_{ij} CGMGD(i,j)}{\sum_{i=1}^{t}\sum_{j=1}^{t} f_{ij}} \quad (1)$$

where *CGMGD(i, j)* is precomputed between all pairs of cell clusters according to Methods section "Connectivity guided minimum graph distance (CGMGD)". And $\sum_{i=1}^{t}\sum_{j=1}^{t} f_{ij} CGMGD(i,j)$ is minimized subject to (2–4).

$$f_{ij} \geq 0, i, j \in [1, t], \quad (2)$$

$$\sum_{i=1}^{t} f_{ij} \leq w_j^{MECN_q}, i, j \in [1, t], q \in [1, n], \quad (3)$$

$$\sum_{j=1}^{t} f_{ij} \leq w_i^{MECN_p}, i, j \in [1, t], p \in [1, n]. \quad (4)$$

### Spatial heterogeneity (SHN) quantification
We define the spatial heterogeneity (SHN) as a numerical property of a node in MEG, to assess the total gene expression variation within the represented MECN. For a *MECN*, suppose it consists of *k* single cells, that is (5).

$$CONSIST(MECN) = (c_1 \ldots c_i \ldots c_k), i \in [1, k] \quad (5)$$

$c_i$ is vectorized by a high-dimensional gene expression feature. Each single cell $c_i$ can be encoded with a categorical variable $T_i$, representing the belonging cell group, which is precomputed by

clustering with the gene expression feature vector (6).

$$GROUP(c_i) = T_i, i \in [1,k], T_i \in [1,t] \qquad (6)$$

Since we have also precomputed the pairwise CGMGD among cells, and the CGMGD between two single cells is a function of their belonging cell groups. We denote the pairwise gene expression variation (PGEV) as (7).

$$PGEV(c_i, c_j) = CGMGD\big(GROUP(c_i), GROUP(c_j)\big), i,j \in [1,k] \qquad (7)$$

The SHN of a *MECN* is computed as the total gene expression variation by summing over $i$ and $j$ (8).

$$SHN(MECN) = \sum_{c_i, c_j \in CONSIST(MECN)} PGEV(c_i, c_j) \qquad (8)$$

When estimating SHN with an isotropic ground distance (IGD), the pairwise gene expression variation is simply replaced with (9).

$$PGEV_{IGD}(c_i, c_j) = IDENTITY\big(GROUP(c_i), GROUP(c_j)\big), i,j \in [1,k] \qquad (9)$$

Where IDENTITY is an indicator function defined as (10).

$$IDENTITY(p,q) = \mathbf{1}_{p! = q}(p,q) \qquad (10)$$

## Spatial domain (SDM) identification

The relationship among MECNs encoded by MEG considers both the cellular composition by representing each MECN as a histogram, and the discrepancy among cells described by gene expression profiles. To connect the local microenvironment with the tissue domain at a relatively broader scale, SOTIP performs the task of spatial domain identification based on the pre-constructed MEG.

Leiden community detection[50] is firstly performed on the MEG to partition MECNs into clusters, to this end the clustered result already reflect meaningful tissue organization. To reach the pre-defined number of clusters, SOTIP next performs cluster merging with a hierarchical scheme based on the connectivity between MECN clusters. Specifically, similar to modularity[90,91], SOTIP measures the degree of connectivity of two MECN clusters by considering the ratio between the number of inter-edges and the expected number of inter-edges under random assignment. SOTIP then iterates between two sub-steps: assess the pairwise connectivity between MECN clusters, and update MECN cluster assignment by merging MECN clusters with largest connectivity, until the number clusters reached the predefined.

Since the process is guided by the finely designed MEG, in each step the algorithm merges two most similar sub regions of tissue. After the hierarchical merging, users can get a multi-level tissue partitioning hierarchy, from which tissue regions with different resolutions can be investigated. This is particularly an advantage over other SDM identification algorithms, e.g., stLearn[38], bayesSpace[36], and SpaGCN[37], especially in real cases that the ground truth of number of tissue regions is not known in advance.

## Differential microenvironment (DME) analysis

The aim of DME analysis is to identify specifically enriched microenvironments between samples (or conditions, disease status, drug treatment, etc.). This task is also based on the MEG construction, but from joining MECNs of two samples. After that, Leiden[50] is used for community detection based on MEG, resulting a cluster assignment for MEs. Then an abstract version of MEG is constructed with partition-based graph abstraction (PAGA)[91]. Each node of the resulted MEG

abstraction is an MECN cluster, and the connectivity between nodes is computed as the ratio between the number of inter-edges and the expected number of inter-edges under random assignment, similar to modularity[90].

To identify interesting MECN clusters with specific sample associated enrichment and more complex cellular composition, an EH plot (abbreviation for "entropy-of-MECN-cluster (EMC)" versus "spatial heterogeneity (SHN)" plot) is drawn for each DME analysis application. EH plot is a scatter plot, in which each point is an MECN cluster, the $x$-axis is defined as average spatial heterogeneity (SHN) across MECNs of the MECN cluster, and the $y$-axis is defined as the entropy of the MECN cluster (EMC) between the compared two samples. Suppose we want to compare the MECN differences of two samples: sample 0 and sample 1, the EMC ($EMC_C$) of an MECN cluster $C$ ($C$ is an MECN cluster, viz., a set of MECNs) is defined on the Bernoulli distribution (parameterized by $p$) modeling the probability of observing an MECN from either samples. That is:

$$EMC_C(p) = -p_C log_2 p_C - (1 - p_C)\log_2(1 - p_C), where\ C\ is\ a\ MECN\ set$$

The parameter of the distribution is estimated as the proportion of normalized count of MECNs (*NCME*) from sample 0, that is (11).

$$p_C = \frac{NCME(C, sample0)}{NCME(C, sample0) + NCME(C, sample1)}, \qquad (11)$$

where NCME of sample k is defined as (12).

$$NCME(C, sample\ k) = \frac{|\{MECN|MECN \in C\ and\ SAMPLE(MECN) = sample\ k\}|}{|\{MECN|SAMPLE(MECN) = sample\ k\}|}, \qquad (12)$$

where *SAMPLE*(*MECN*) is defined as which sample the MECN comes from. Interesting MECNs (strong specificity and high SHN) lies in the bottom right corner of the EH plot. Above analysis can be extended to multiple samples/conditions with multinomial distribution.

To summarize, the input of SOTIP-DME is two tissue samples (spatial omics data), and the output is the EH plot. Using the EH plot, each point is a MECN cluster, one can visually assess those MECN clusters with high sample-specificity and high spatial heterogeneity.

## Connectivity guided minimum graph distance (CGMGD)

Computing the earth mover's distance between microenvironments needs to define the ground distance between cell clusters. Directly computing Euclidean distance in the high-dimensional gene expression space would be cursed by the dimensionality. Early single cell analysis[92] attempted to compute distance in the embedded space, e.g. t-SNE[93], or UMAP[89], to perform downstream analysis such as clustering, but was proved problematic since the principle of these algorithms made it poorly preserving global topological data structure[94–96]. Diffusion pseudo-time (DPT)[97] computes the geodesic distance between each data point and a given root point in a diffusion map embedded space, which seems consistent with our objective. However, since we need to compute the ground distance between cell clusters, we need to run DPT (e.g., SCANPY implementation[32]) for multiple times by setting the root point to the center cell of every cell cluster. So the computational efficiency is a problem, not to mention the choice of the cluster centers. More recently, PAGA[91] is reported to preserve the topological data structure of the high-dimensional scRNA-seq data, and has been widely used as an official module of SCANPY[32]. It is also not suitable for us since it assesses the connectivity between clusters, instead of the distance we need. PHATE[53] is reported to preserve better distance than t-SNE and UMAP, there are also researches applying Euclidean distance within PHATE embedding to assess the distance

between single cells, we found it produced errors with our simulated data.

To more accurately approximate the distance in the transcriptome state space, and also accounting for computational efficiency, we propose connectivity guided minimum graph distance (CGMGD). Specifically, since we only need the cluster level distance instead of single cell level, CGMGD firstly assesses the connectivity between cell clusters with PAGA to produce a binary connectivity matrix (BCM) justifying whether there are edges (1) or not (0) between two cell clusters. In the second step, CGMGD embeds the single cells into UMAP space, and calculates the pairwise distance of cluster centers within the embedding space to produce a UMAP distance matrix (UDM). In the third step, a connectivity-guided cluster graph (CGG) is constructed, of which each node is defined as a cell cluster. The edge of the graph is defined with the adjacent matrix computed by element-wise multiplication between BCM and UDM. Finally, the CGMGD is computed by searching pairwise minimum distance of the CGG. With this procedure, CGMGD approximates the distance in the transcriptome state space by integrating the global topological preservation of PAGA, and the local data structure preservation of UMAP. We used simulation data 4 (see "Methods") to demonstrate the advantage of CGMGD over six manifold learning algorithms, PCA, UMAP[89], DPT[97], ForceAtlas[98], PHATE[53], and PAGA[91] (Supplementary Fig. 8).

### Correlation analysis between SHN and cortical depth

To quantify the ordinal agreement between the SHN values and the cortical layer, we used both ME-wise and layer-wise Spearman's rank correlation coefficients (Spearman's $\rho$) between them. For ME-wise manner, the SHN was computed for each MECN, and the layer depth of each MECN was assigned by setting a number for each layer, Pia Layer 1 (1), Layer 2–3 medial (2.5), Layer 2–3 lateral (2.5), Layer 3–4 (3.5), Layer 4 (4), Layer 5 (5), Layer 6 (6). The Spearman's $\rho$ was computed between the SHNs and layer depths of MEs. For layer-wise manner, the SHN was computed as median SHN of MECNs for each layer, and the layer depth was assigned as with ME-wise manner. The Spearman's $\rho$ was computed between the SHNs and layer depths of layers.

### Validation of identified DMEs in all TNBC patients

To validate that our identified differential MEs, i.e., MECN cluster 9 (Tumor and Keratin+ tumor) and 24 (CD8 T, macrophage, and Keratin+ tumor) consistently differentiated between compartmentalized and mixed samples from other patients, we performed MECN occurrence analysis as follows. For ease of explanation, we used MECN cluster 9 as an example.

For each patient sample, we firstly counted the number of target MECNs (MECNs which contained cells from both Tumor and Keratin+ tumor clusters) as $k_{obs}$. In order to account for difference in number of cells, the expected number of target MECNs ($k_{exp}$) under background distribution was computed by random permutation of cell cluster labels, followed by averaging across their number of target MEs. Formally, $\bar{k}_{exp} = \frac{1}{N} \times \sum_{i=1}^{N} \bar{k}_i$, where $N$ is the number of permutations, and $\bar{k}_i$ is the count of target MECNs in each permutation. The occurrence score of MECN cluster 9 is computed as $k_{obs}/\bar{k}_{exp}$. Following this procedure, we computed the occurrence score for every patient, and compare the occurrence scores between compartmentalized ($N=15$) and mixed ($N=19$) patients. The significance is estimated using one-sided Wilcoxon rank-sum test.

### Simulation datasets

There are four simulation datasets used in this study. Simulation 1-3 were firstly generated with Splatter[99], a R package for scRNA-seq data simulation, then arrange the spatial coordination for the generated cells. Simulation 4 is generated by Splatter as single-cell data, without spatial coordination. The full information for these datasets is summarized in Supplementary Table 2.

In the analysis of simulation 3 (Fig. 2e, f), the relative likelihood of observing each microenvironment in specific sample is performed by MELD[55], with the precomputed MECN distance matrix as input.

### MELD analysis

MELD[55] is an algorithm for estimating the relative likelihood of observing each cell state between different conditions (e.g., different disease state, before/after drug treatment or other experimental perturbations). In original MELD publication, the input of MELD is (1) the single cell graph on gene expression space, and (2) the condition label for each single cell. The output of MELD is the relative likelihood of observing each cell in each condition.

In our application (e.g., Fig. 2f), we instead input the microenvironment graph, and the condition label for each microenvironment. In this way, by combining SOTIP's mathematical definition of microenvironment and MELD, we can estimate the relative likelihood of observing each microenvironment between different conditions.

### PAGA analysis

PAGA[91] is an algorithm for trajectory inference through a topology-preserving map of single cells. In the original PAGA publication, the input of PAGA is (1) the single-cell graph on gene expression space, (2) a clustering assignment for the single cell data. The output of PAGA is graph, in which each node is a cluster, and the edge between two nodes is the strength of connectivity (the connectivity could also be interpreted as similarity) between two clusters. In our application (e.g., Supplementary Fig. 3), we instead input the microenvironment graph, and the clustering label. So that we can get the topological map of the targeted tissue.

### Public datasets

This study involved multiple datasets from seven different spatial omics technologies. All datasets are publicly available. Human HeLa cell line 4i dataset: https://squidpy.readthedocs.io/en/stable/[23]. Mouse brain cortex osmFISH dataset: http://linnarssonlab.org/osmFISH[46]. Mouse brain cortex seqFISH+ dataset: https://github.com/CaiGroup/seqFISH-PLUS[17]. Zebrafish melanoma Visium dataset: GSE159709[61]. Human brain cortex Visium dataset: http://research.libd.or g/spatialLIBD/[48]. Human colorectal carcinoma scMEP dataset: https://zenodo.org/record/3951613[49]. Human triple negative breast cancer MIBI dataset: https://mibi-share.ionpath.com/[12]. TOF-SIMS liver dataset: https://github.com/yuanzhiyuan/SEAM/tree/master/SEAM/data/raw_tar[7]. The full information of public datasets used in this study can be found in Supplementary Table 3.

### NUCC and IGD

NUCC[16,33] and IGD are both control methods for spatial heterogeneity (SHN) quantification. NUCC is a simple and widely used method. The main idea is to quantify the spatial heterogeneity around a cell by counting the number of unique cell clusters within the neighborhood. Its rationale is straightforward, if a cellular neighborhood contains many unique cell types, then the SHN of that neighborhood is high. The "cell clusters" could be defined by either single-cell clustering algorithm, or by cell type annotation. The"cellular neighborhood" could be defined by either k-nearest-neighbor like SOTIP-SHN, or by a user-defined radius.

IGD is a variant of SOTIP-SHN, and is also proposed as a control method for SHN quantification by this paper. We propose IGD to prove the advantage of using CGMGD in SOTIP-SHN. The only difference between IGD and SOTIP-SHN is that, SOTIP-SHN used the sum of CGMGD distance matrix within the cellular neighborhood, while IGD replaced the CGMGD distance matrix with IDENTITY distance matrix (please refer to "Spatial heterogeneity (SHN) quantification" section).

More explanations are in "Further explanations on SHN" in Supplementary Notes.

## Method comparison

To evaluate the performance in spatial heterogeneity (SHN) quantification, NUCC and IGD was compared with SOTIP. As with SOTIP, NUCC, IGD both need to perform clustering and define spatial neighborhood. For a fair comparison, NUCC, IGD, and SOTIP utilized exactly the same parameters (MECN size and cluster resolution). We also compared the relative performance under different combinations of these parameters, the parameters were also consistent in every comparison. With 4i dataset (Fig. 3a–f), since the original data (Supplementary Table 3) provided ground truth of nucleus and cytoplasm, we defined their boundary as width of 2 pixels. We used area under curve (AUC) to evaluate and compare the performance. With osmFISH dataset (Fig. 3k), the original data (Supplementary Table 3) provided cell-wise annotations of different cortical layers. We used Spearman's rank correlation coefficient to evaluate the ordinal agreement between the layer order and the computed SHN. To digitalize the layer order, we set a number for each layer, Pia Layer 1 (1), Layer 2–3 medial (2.5), Layer 2–3 lateral (2.5), Layer 3–4 (3.5), Layer 4 (4), Layer 5 (5), Layer 6 (6). To test the impact of over-clustering on SHN quantification (Supplementary Fig. 2), on osmFISH data and seqFISH data, we set the clustering resolution to *moderate* resolution, over-clustering (3 × *moderate*), and extreme over-clustering (5 × *moderate*). The moderate resolution was determined by resolution researching given true number of clusters (provided by original paper[17,46]).

To evaluate the performance in spatial domain (SDM) identification, for both brain cortex datasets (Fig. 4), we set the true number of clusters to run compared algorithms, and used adjusted Rand index (ARI) to compare the similarity between clustering result and ground truth. Note that for human sample, since original ground truth labels contain "N/A" (Fig. 4b), we filtered out these spots before the comparison. The ARI of the human sample was directly adopted from the ARI as they reported in the original paper[36,37]. We also applied the same gene filtering and data normalization procedures as with code provided by SpaGCN. For the mouse cortex osmFISH dataset, the original ground truth labels contain "excluded" (Fig. 4e), we filtered out these single cells before the comparison, and the parameter of SpaGCN was set as default. As for the spatial proteomics datasets (Fig. 5), the original paper provided ground truth for tumor-immune boundaries, based on which we compared SOTIP and SpaGCN qualitatively. For better visualization[29], we plotted the results by Voronoi diagrams, adopted from https://stackoverflow.com/questions/20515554/colorize-voronoi-diagram/20678647#20678647%203.18.2019.

## Parameter settings

There are three parameters for user to choose according to their applications. These parameters are set with a consistent manner except when evaluating the robustness to different combinations of parameters.

Across the manuscript, the resolution of Leiden clustering is set by searching according to the true number of cell clusters if available, otherwise 2. The parameter of MECN size (k-NN parameter) is set to 10. The number of spatial domains is set according to the ground truth. More discussions of parameter settings can be found in Supplementary Notes.

## Molecular polarization with spatial proteomics data

In the CRC datasets, following the consistent manner with Hartmann et al.[49], we first classified immune cells within a 51 pixels (20 μm) radius to tumor region as proximal, and other immune cells as distal (Supplementary Fig. 4a, b). In the TNBC datasets, and also consistent with Keren et al.[12], the definition of proximity of both "tumor cells to immune region" and "immune cells to tumor region" is defined with radius of 100 pixels (39 μm) (Supplementary Fig. 4c, d). The criterion for selection of representative polarized proteins (Fig. 4f, l,

Supplementary Fig. 4g) is by ascendingly sorting proteins by FDR, and the representative proteins are overlay and plotted by the interactive plotting webserver Ionpath (https://www.ionpath.com/).

## Steps of Fig. 5f–i

We identified the polarized proteins following the standard procedures of previous studies[12,49], with the following steps:
- Define spatial domains using either SpaGCN or SOTIP's SDM module.
- Classify immune cells into two groups. Group1: Proximal (those immune cells whose smallest distance from any tumor cells are smaller than a radius), and Group2: Distal (those immune cells whose smallest distance from any tumor cells are larger than a radius).
- For each protein, we compared the protein abundance between Group1 and Group2, and assessed the significance using two-sample rank-sum test, which is further corrected by Benjamini–Hochberg (BH) procedure for multiple hypotheses testing.
- We set different FDR cutoffs (the *x*-axis in Fig. 5f, i) to filter those statistical significant polarized proteins and obtained the identification power by comparing with the true positives.

The true positives are based on the polarized protein list identified in original papers[12,49].

## Boxplot

All boxplots in the manuscript share the same settings: the lower and upper hinges show the first and third quartiles (the 25th and 75th percentiles); the center lines correspond to the median; the upper whisker extends from the upper hinge to the largest value, which should be <1.5× the interquartile range (or distance between the first and third quartiles) and the lower whisker extends from the lower hinge to the smallest value, which is at most the 1.5× interquartile range. Data beyond the end of the whiskers are 'outlying' points and are plotted individually.

## Quantitative and statistical analysis

All statistical tests used in this study are described in detail in the corresponding figure legends. Spearman's rank correlation coefficients, Pearson correlation coefficient, and Wilcoxon rank-sum test are performed using scipy[100]. To account for multiple hypotheses testing, we applied the Benjamini–Hochberg (BH) procedure to report the associated FDR[101], which is performed using pingouin[102]. ARI and AUC are performed using scikit-learn[103]. Survival analysis was performed using scikit-survival[104]. Numpy, Pandas, and Scikit-learn were used to perform scientific computing. Matplotlib, seaborn, palettable were used to generate figures. SCANPY and Squidpy were used to analyze spatial data. Networkx and shapely was used to deal with graph representation. Pyemd was used to compute EMD distance.

## Statistics and reproducibility

For Figs. 3g, h, 5d, f, g, I, 6b, c, d, and Supplementary Fig. 4e, g, the experiment results were similar with at least three independent repeats.

# Data availability

All raw data are freely available at following links: Human HeLa cell line (4i): [https://squidpy.readthedocs.io/en/stable/] Mouse Brain cortex (osmFISH): [http://linnarssonlab.org/osmFISH] Mouse Brain cortex (seqFISH+): [https://github.com/CaiGroup/seqFISH-PLUS] Zebrafish melanoma (10X Visium): GSE159709 Human Brain cortex (10X Visium): [http://research.libd.org/spatialLIBD] Human colorectal carcinoma (scMEP): [https://zenodo.org/record/3951613] Human triple-negative breast cancer (MIBI): [https://mibishare.ionpath.com/] Liver (SIMS):

[https://github.com/yuanzhiyuan/SEAM/] Mouse brain (EASI-FISH): [https://janelia.figshare.com/articles/dataset/EASI-FISH_enabled_spatial_analysis_of_molecular_cell_types_in_the_lateral_hypothalamus/13749154] Human breast cancer (3D IMC): [https://doi.org/10.5281/zenodo.4752030] The processed data in this manuscript can be downloaded at figshare [https://doi.org/10.6084/m9.figshare.18516128]. Source data are provided with this paper.

## Code availability

An open-source Python implementation of SOTIP and reproduction code are available at https://github.com/TencentAILabHealthcare/SOTIP.

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

## Acknowledgements

This research was supported by the National Key Research and Development Program of China (2018YFA0801402), National Natural Science Foundation of China (81890994) to M.L.S., National Natural Science Foundation of China (81890991), Beijing Municipal Natural Science Foundation (Z200021), CAS Interdisciplinary Innovation Team (JCTD-2020-04), and the State Key Research Development Program of China (2021YFE0201100) to J.T.G. M.Q.Z. acknowledges the support by the Cecil H. and Ida Green Distinguished Chair. Z.Y.Y. acknowledges the support by Shanghai Municipal Science and Technology Major Project (No.2018SHZDZX01), ZJ Lab, Shanghai Center for Brain Science and Brain-Inspired Technology, and 111 Project (No.B18015).

## Author contributions

M.Q.Z. and Y.L. conceived and designed the project. Z.Y. and Y.L. developed and implemented the algorithms under the guidance of M.Q.Z. and J.Y. Z.Y. and Y.L. collected and processed public datasets. Z.Y. and Y.L. conducted the data analysis and methods comparisons. Y.L. and M.S. did the biological interpretation. M.S. and J.G. gave suggestions on the applications of the method. Z.Y. and Y.L. completed the figures and manuscript with the guidance of J.Y., M.S., and M.Q.Z. F.Y. helped with the figure generation. All authors approved the manuscript.

## Competing interests

F.Y. and J.Y. are employees of Tencent. The remaining authors declare no conflicts of interest.
