## [Transparent Peer Review File · Nature Communications]

SOTIP is a Versatile Method for Microenvironment Modelling with Spatial Omics Data

Redactions – published data

Parts of this Peer Review File have been redacted as indicated to remove third-party material.REVIEWER COMMENTS

Reviewer #1 (Remarks to the Author: Overall significance):

An integrated package to process spatial organization in tissues from multiple spatial omics datasets.

Reviewer #1 (Remarks to the Author: Impact):

Most methods have previously highlighted in previous literature. I would suggest Communications Biology for this paper.

Reviewer #1 (Remarks to the Author: Strength of the claims):

Quantitative metrics comparing previous methods seem solid. I suggest this user friendly comparison to allow other researchers in spatial omics.

Reviewer #1 (Remarks to the Author: Reproducibility):

The results agree with previously demonstrated spatial organization in previous methods and data analysis techniques. Thus, it sounds reproducible.

Reviewer #2 (Remarks to the Author: Overall significance):

Yuan et al. present a new method, SOTIP (Spatial Omics multiPle-task analysis), and show its versatility in analyzing a variety of spatial omics datasets including spatially resolved transcriptomics data and proteomics data. In particular, the algorithm performs 3 main analytical functions including spatial heterogeneity quantification (SHN), spatial domain identification (SDM), and differential microenvironment analysis (DME). The authors benchmark against several recently published/pre-printed algorithms (using different algorithms for different datasets and tasks as applicable). The authors also demonstrate some novel biological findings using their algorithm to mine existing data.

While the method performs well, it is difficult to interpret some of the benchmark studies due to the lack of clarity surrounding the 1) the large number of datasets/modalities/biological systems used in the manuscript and 2) differences between SOTIP and existing algorithms. The manuscript could be more accessible by building into Figure 1 the various spatial omic datasets being used and how they are being used (i.e. for SHN, SDM, or DME). While some of this information is in supplementary tables, it is very difficult to interpret while reading the manuscript. The datasets are often only superficially explained, which also makes it challenging to evaluate the data or how SOTIP performs on different types of data within a modality (for example, single cell spatially resolved transcriptomics vs. Visium). And only some tasks are run on certain datasets while others are not, which again, makes it difficult to interpret the benchmarking and claims of superior performance. Furthermore, SDM is a highly desirable task for spatial omics, but it was difficult to clearly understand the utility of SHN or DME—when one would want to use these tasks and why these tasks might be better employed than other methods—this needs to be better fleshed out.

In general, some novel features of SOTIP that are desirable compared to other algorithms are that it doesn't require pre-determined k for finding spatial domains. It is compatible with proteomics data and 3D data. It also identifies spatial domains more closely resembling ground truth compared to existing algorithms. The tutorials are also extremely useful, although there are some accessibility and clarity issues detailed below.

Reviewer #2 (Remarks to the Author: Impact):

The development of algorithms for unsupervised analysis of spatial omics data is rapidly growing as more spatial omics data sets come online. There is a need and high interest in improving algorithms for spatial data and SOTIP offers a unique and versatile approach. However, this work may be better suited for a more methods-focused journal. As the manuscript is currently written, it is for a highly specialized spatial biology computational audience (for example, different types of spatial data are not explained well, current algorithms are not described well, and computational terms/metrics are not explained—it is assumed that the reader knows these spatial techniques, algorithms, metrics).

Reviewer #2 (Remarks to the Author: Strength of the claims):

1. As already mentioned, the authors should provide better descriptions of the tasks SOTIP is performing and why and how it compares to other algorithms. For example, there is no explanation of NUCC and IGD, which the authors are benchmarking against. How is SOTIP different from these algorithms?

2. The authors should include a comparison between squidpy and SOTIP as squidpy uses a similar concept of nodes and distances between nodes to find similarities and dissimilarities.

3. How does SOTIP compare with RESEPT (<https://www.biorxiv.org/content/10.1101/2021.07.08.451210v1>) for spatial pattern inference?

4. A major benefit of BayesSpace is that it can be run across all samples for SDM such that the same clusters are identified across a large sample set. Can SOTIP be implemented across multiple samples or can it only be run tissue section by tissue section?

5. A major focus in the field has been the integration of single cell and spatial omics data sets. In terms of DME, when relevant single cell data is also available, how does SOTIP outperform algorithms such as Cell2Location (<https://www.nature.com/articles/s41587-021-01139-4>) that are also able to employ microenvironment analyses using non-negative matrix factorization approaches (see Fig. 4).

Reviewer #2 (Remarks to the Author: Reproducibility):

Major comments:

1. As noted, the manuscript lacks clarity and is difficult to read. In general, the authors could improve the clarity of the manuscript in the following ways:

- a. Shorten the text in all sections and make statements more concise.
- b. Define all acronyms and avoid use of acronyms where possible
- c. Remove jargon and better explain in text technical terms that may be unfamiliar to readers such as specific algorithms, spatial-omics data sets, and computational terms.
- d. There are several spelling/grammar issues that should be corrected throughout the manuscript in all sections.
- e. Provide more details in figure legends and explain all arrows.
- f. Parts of results that are better suited for discussion (for example, lines 433-459, 745-773)

2. There are several issues with code reproducibility:

- a. Provide code for installation instructions
- b. Unable to render the code on github. For example, these 3 files don't render:
https://github.com/TencentAILabHealthcare/SOTIP/blob/master/SOTIP_analysis/osmFISH_Cortex/SHN_osmFISH_MEwise.ipynb,
https://github.com/TencentAILabHealthcare/SOTIP/blob/master/SOTIP_analysis/MIBI_TNBC/DMA_TNBC.ipynb,
https://github.com/TencentAILabHealthcare/SOTIP/blob/master/SOTIP_analysis/Visium_Zebrafish/SHN_Visium_zebr oFISH_A_diff_res.ipynb
- c. Visium DLPFC figure 4c does not match the reproducible example on github (https://github.com/TencentAILabHealthcare/SOTIP/blob/master/SOTIP_analysis/Visium_Cortex/SDM_Visium_cortex_best.ipynb)
- d. Have the authors provided the exact code to reproduce all the figures in the manuscript?

Minor comments:

1. MKT is not defined in the abstract line 46.
2. Provide a clearer and simpler explanation of the term ME (microenvironment).
3. Intro line 68-69 is not clear
4. What do the blue dots along the x-axis in figure 1e represent?
5. Figure 2d legend and figure don't match
6. The result and interpretation of Figure 2f is unclear.
7. Figure 3, subplot 'L' missing?
8. The authors should show spatial trajectory graph of sample 151673 in figure 4i
9. Line 492 PAGA has not yet been defined?
10. Figure 5a-c needs more explanation on figure itself and in legend
11. Figure 5d,e which is tumor and which is immune, red or black? Bounded regions are not labels. Arrows are not described.
12. All figures in figure 5 and corresponding supplementary figure should be shown on same axis orientation (e.g. rotate 90 degrees as needed)
13. Line 310- what is meant by clipped?
14. Figure 6c is not referred to in text
15. Fig 7, "L" is missing?

Reviewer #3 (Remarks to the Author: Overall significance):

The manuscript described SOTIP, a suite of software tools for analyzing spatial omics data, where the molecular profiles of individual functional units in the tissue are linked to the spatial location of that unit. For technologies such as Visium and Slide-seq, the functional units are spots or beads used to capture the local RNA molecules, thus the RNAseq data for individual spots or beads are accompanied by their x-y coordinates. There is a strong need for computational tools that can analyze such data. SOTIP is designed to accomplish three main tasks: spatial heterogeneity (SHN) quantification, spatial domain (SDM) identification, and differential microenvironment (DME) analysis. The algorithms were described in Methods; while Results were devoted to reporting performance in various tasks, along with some comparisons with previously published software tools.

A brief note of the method. For data with N spots and P features, one can use the P -feature vectors to build spot-spot "distance" measures as one of the starting points to explore spatial heterogeneity. In SOTIP, every spot is decomposed as a mixture of t clusters, where the clusters have been predefined, and sometimes referred to as cell types. Thus a second way to build spot-spot distance measures is to use the t -element composition for each spot. SOTIP uses a third measure. For each spot, its neighboring k spots form a "graph" of k edges. The distance between Spot- i and Spot- j is by comparing graph- i and graph- j , where each graph has M edges, from the index spot to its M neighbors, and each of the $M+1$ nodes is characterized by the K -cluster composition. The rationale is that such a graph represents each spot's microenvironment. This is a good concept and, when used well, can complement each spot's innate properties as measured by the first two options (P features and K -cluster composition). The use of network-based distance measures is part of an original and significant area of research.

A minor confusion: The graph-to-graph connectivity-based distance is defined in line.933. What is confusing is that the same distance seems to also used to calculate edge-distance within each graph (line.954).

Reviewer #3 (Remarks to the Author: Impact):

The paper has important elements in the development of the algorithms. However, they are not always presented in an easy-to-understand fashion. The exploration of the algorithms may influence the thinking of the field. The claim that the package has better performance is difficult to evaluate, and not likely to have a strong impact.

Reviewer #3 (Remarks to the Author: Strength of the claims):

1. The descriptions of the algorithms in Methods were not easy to follow, especially with regard to the rationale and the innovation when compared to past efforts. It should be considered to explain the methods first.
2. Spatial heterogeneity quantification (the first task) is based on the total edge distance among the k neighboring cells. The better performance of the connectivity-based distance is shown in a simulation (Figure 2B). I am not convince because the use of the original P -feature vector would also perform well in the situation in Figure 2B.
3. The second task, spatial domain identification, has a major conceptual issue. When each of the N spots has been recoded as a graph with k edges, and each of the k nodes has been recoded as a histogram of t components, the "domain" is found by iteratively clustering the N spots by their connectivity-based distance, resulting in a hierarchical clustering tree. As described in lines.958-985, the spatial relationship of the spots are not used, therefore the "neighboring" spots on the tree may or may not be near each other, to form a domain. It is the reader's guess that the number of clusters will come from a tree-cutting method, say have T clusters by cutting the tree at the level of K branches. But they will not form K domains as the spots within a cluster are likely non-contiguous in the tissue. If spatial proximity is used in the algorithm, it is not described. And if it is indeed used, the relative weight of spatial distance and the connectivity -based distance is the key part of the method. Many existing tools have confronted this issue.
Besides, clustering of the N spots can be done by using the original P features or the dimension-reduced alternatives, such as the composition of the t clusters. The relative merits of these alternatives need to be better explored, first in simulation, before moving to real data.
4. The third task, differential microenvironment analysis, requires significant clarification. As described in lines.986-1016, sometimes the comparison seems to be between two different tissue samples, while some other times it seems to be between two different "clusters" within a sample, here the "clusters" seem to be the "domains" found in task-2, not the t spot clusters used in the early steps. The analysis is essentially a visualization in EH plot. If there is a test statistics also proposed, it was not easy to find in the text.
5. The choice of the distance measures discussed in "Connectivity guided minimum graph distance (CGMGD)" is the foundation of SOTIP. The advantages and disadvantages of the distance measures need to be extensively examined.
6. The Results section focused on examples of application. I did not go through them in detail. There are several difficulties. One is the degree of difficulty of knowing the true signal. Some examples are "easy", such as finding different cortical layers have different spatial properties (Figure 3j-o). What constitutes a relevant improvement of performance is not easy to define.

Reviewer #3 (Remarks to the Author: Reproducibility):

I did not evaluate the quality of the codes but appreciated that the link to GitHub was provided.

The simulations will be more helpful if they are described in greater detail: what type of statistical properties was embedded to probe which type of performance.

The terminology can be much improved. "Single cell" was used to refer to spots. The difference between MECN and MEG took repeated reading to understand. Differential analysis of microenvironment can happen on multiple scales. As written it is unclear which scale does the comparison take place.

Reviewer #4 (Remarks to the Author: Overall significance):

The authors propose a method called SOTIP to perform three main tasks: 1) spatial heterogeneity quantification, 2) spatial domain identification, and 3) differential microenvironment analysis.

Those tasks are tackled by constructing a MECN graph that uses the gene expression and cell-type information of the neighbourhood. The authors demonstrated their method via both simulation and real data. In general, this is a well-written paper.

Reviewer #4 (Remarks to the Author: Strength of the claims):

This method is convincing, although I have a few comments.

General remarks:

1) In simulation, the tissue structures are fairly simple.

Please explain/interpret the SHN values in figure 2b. Are they estimated using the same algorithm? If so, how to interpret the large range of SHN estimated by SOTIP? If not, how to compare across methods?

Based on simulation 1 (used in Figure 2a), c1 and c2 are indeed two clusters and there does exist spatial heterogeneity, although it is a weak one. NUCC and IGD can clearly identify it. However, the SHN of this boundary estimated from SOTIP is ~ 200 , much smaller than the one for C2 and C3. Is it possible that the statistical power of SOTIP is smaller than another two methods? Similar to Figure 2D, the UMAP of CCNR can confidently split the C1- C5 as five clusters, however, the UMAP of SOTIP cannot, if we don't have prior cluster information.

Authors argue this is an advantage that SOTIP preserves the continuous trend in the PC plot, which I'm afraid I have to disagree. In my opinion, this is a lack of power to cluster confidently.

For simulation 3 (Figure 2e and 2f), samples 1 and 2 are a clean contrast. C1|C2 in sample 1 is unique compared to sample 2, and C3|C1 does not exist in sample 1. In reality, there could exist C3|C1 in sample 1 as well. For example, there could be a subtle difference in cell spatial distribution in the different stages of disease progression. But this subtle difference could affect the treatment outcome. In this case, the contrast between samples is not as straightforward as in figure 2e, increasing the noise in identifying spatial heterogeneity. What is the performance of SOTIP in such a case?

Still, for simulation 3, there are equal distances among C1-C3. What is the effect if two clusters are close in PC space (e.g. C1 and C2 Figure 2a), compared to the third cluster (C3 in Figure 2a), but those clusters have different spatial distributions in different samples (eg the distribution of figure 2e).

In all simulations, the number of cells per cluster is almost identical. How the size of a cluster affects the detection of SHN?

In my opinion, comprehensive simulation can provide a good sense on the power and type I error rate of the method.

2) Figure 1a, please state what are those red squared boxes standing for?

Figure 1e, from left to right, is the degree of spatial heterogeneity increasing or decreasing?

In general, please add a more detailed description to the figure captions.

3) Line 989-990: "but from joining MECNs of two samples": do you perform PCA, neighbors, umap, Leiden within each sample to get MECN and then combine those two MECN? Or do you combine expression data and spatial data of two samples, then perform PCA, neighbors, umap, Leiden to get a MECN for the combined data? If it is the former one, the same cluster (e.g. Cluster 1) in two samples may stand for different cell types. Do you need to perform cell-type annotation beforehand? If it is the latter, how does the batch effect affect the MEG construction? It was not clear to me how you combined two samples.

4) Figure 6e-i demonstrate the DM in the fibrotic liver, compared to the healthy liver. Is there any DM in the healthy liver? Are they make biological sense?

Minor comments:

- 5) Too many acronyms and some are quite similar. I suggest having a table/ box listing all the acronyms. This will make the manuscript easier to read.
- 6) On Lines 145-174 (figure 1a): the MECN is a vector of cell type frequency within a cellular neighbourhood. Thus, the definition of the microenvironment is the same as before.
- 7) Figure 5: what is the meaning of white errors?
- 8) In Figure 7, the choice of interesting EMCN clusters is "bottom-right" of the EH plot. Is there a way to quantify this 'bottom-right'?
- 9) Line 352-353: how many cells within a radius of 100 um? Does "the same manner as NUCC" mean that NUCC uses cells within 100um to define a ME, but SOTIP uses 10 nearest neighbouring cells?
- 10) Typo on line 426: the mean of SpaGCN is 0.433, instead of 4.33
- 11) Supplementary figure 4: a-d, the dots in the scatter plots have different sizes. Please add a legend to explain the meaning of it.
- 12) Supplementary figure 9: please explain 'WB'.
- 13) Provide a guide for the time, number of CPUs, number of threads etc for each of the analyses. Thus, users can easily set up the required computational resource in their analyses.
- 14) Line 712: Keratin+ tumor means keratinocyte and tumour?
- 15) Lines 1101, please give sufficient detail about MELD, such that I don't have read the original paper. The same to PAGA.
- 16) Lines 1122-1123: I'm afraid I have to disagree that using exactly the same parameter for different models is a fair comparison.
- 17) Line 903-910 is not clear to me how to define a MECN. I understand that a MECN is measured per cell, the cell type frequency estimated based on its KNN (10). Or in MEG, each node is a cell, the feature of this node is MECN. The MECN is the cell type frequency estimated based on the cell's KNN.
- 18) Table Supp 4, 5 where is a comparison with RESEPT, FICT, ect?

Reviewer #4 (Remarks to the Author: Reproducibility):

The program on GitHub can be run successfully, and demo data can be downloaded too.

Responses to the reviewers' comments

Response to all reviewers:

We are very grateful to the reviewers for their positive and constructive comments on our manuscript. We believe that their valuable suggestions have helped us greatly improve the manuscript. We have updated the manuscript accordingly and provide below point-by-point responses to the reviewers' comments. In this letter, original comments of reviewers are highlighted in yellow, author's responses are in black, referred contents from original manuscript are in blue, changes in revised manuscript are in red.

Reviewer #1:

Reviewer #1 (Remarks to the Author: Overall significance):

An integrated package to process spatial organization in tissues from multiple spatial omics datasets.

Reviewer #1 (Remarks to the Author: Impact):

Most methods have previously highlighted in previous literature. I would suggest Communications Biology for this paper.

Reviewer #1 (Remarks to the Author: Strength of the claims):

Quantitative metrics comparing previous methods seem solid. I suggest this user friendly comparison to allow other researchers in spatial omics.

Reviewer #1 (Remarks to the Author: Reproducibility):

The results agree with previously demonstrated spatial organization in previous methods and data analysis techniques. Thus, it sounds reproducible.

Author Response: We thank the reviewer for his/her comments.

Reviewer #2:

Comment 1: Yuan et al. present a new method, SOTIP (Spatial Omics multiPle-task analysis), and show its versatility in analyzing a variety of spatial omics datasets including spatially resolved transcriptomics data and proteomics data. In particular, the algorithm performs 3 main analytical functions including spatial heterogeneity quantification (SHN), spatial domain identification (SDM), and differential microenvironment analysis (DME). The authors benchmark against several recently published/pre-printed algorithms (using different algorithms for different datasets and tasks as applicable). The authors also demonstrate some novel biological findings using their algorithm to mine existing data.

Author Response: After reading all your comments, we think your suggestions are very important in improving our work. You have also raised some important and critical issues, including some criticisms, and we are very grateful for your efforts. We have carefully taken your concerns and responded to your comments one by one below. Thank you again for your professional and constructive comments!

Comment 2: While the method performs well, it is difficult to interpret some of the benchmark studies due to the lack of clarity surrounding the 1) the large number of datasets/modalities/biological systems used in the manuscript and 2) differences between SOTIP and existing algorithms. The manuscript could be more accessible by building into Figure 1 the various spatial omic datasets being used and how they are being used (i.e. for SHN, SDM, or DME). While some of this information is in supplementary tables, it is very difficult to interpret while reading the manuscript. The datasets are often only superficially explained, which also makes it challenging to evaluate the data or how SOTIP performs on different types of data within a modality (for example, single cell spatially resolved transcriptomics vs. Visium). And only some tasks are run on certain datasets while others are not, which again, makes it difficult to interpret the benchmarking and claims of superior performance.

Author Response: We thank the review for the constructive advice! We interpreted this comment with 4 points as follows.

1. The large number of datasets/modalities/biological systems is lack of clarity.
2. The differences between SOTIP and existing algorithms is unclear.
3. A lot of information is in supplementary files, making it difficult when reading the manuscript. Important information (e.g., various spatial omics datasets) should be added in Figure 1 to make it more accessible.
4. Some tasks are run on certain datasets while others are not.

We response point by point:

1. To demonstrate SOTIP's applicability and versatility, the amount and diversity of different datasets and biological systems is necessary. For better clarity, we summarized the datasets and biological systems information in Supplementary Table 8. In that table, every dataset used in the manuscript is described in terms of 6 aspects, i.e., Module (which algorithm module of SOTIP was applied on this dataset), Design of demonstration (what biological feature was contained in this dataset), Species, Tissue, Protocol, Detected molecule, and positive controls.

To explain the demonstration logic, we added three figures, explaining the biological systems in SOTIP's three modules, respectively. Supplementary Figure 13 for SHN module, Supplementary Figure 14 for SDM module, and Supplementary Figure 15 for DME module. All these appended supplementary figures and tables are referred in the revised manuscript at the end of introduction section, so that readers can freely access this information before reading the results. These new figures and tables are referred as follows:

SOTIP is freely applicable to various experimental protocols (Fig. 1i, Supplementary Table 3).

We summarized the datasets and corresponding positive controls in Supplementary Table 8, the demonstration logic in Supplementary Fig. 13-15.

Besides, in our original manuscript, we described every dataset when they are firstly used in the manuscript, we also provided schematic image for the main datasets we used (Figure 3a, 3j, 4a, 4e, 5a, 6a, 7a).

Based on these efforts, we hope the datasets could be better explained to readers.

2. This is a very good and important advice to pay more attention to the difference between SOTIP and other related methods in the manuscript. We greatly appreciated this. In the following, we listed the added texts in the revised manuscript, to explain more for the potential control methods of SOTIP's three modules.

In the revised manuscript, we added following texts in Methods, to explain IGD and NUCC, the control methods in SHN quantification.

NUCC and IGD

NUCC¹⁻³ and IGD are both control methods for spatial heterogeneity (SHN) quantification. NUCC is a simple and widely used method. The main idea is to quantify the spatial heterogeneity around a cell by counting the number of unique cell clusters within the neighborhood. Its rationale is straightforward, if a cellular neighborhood contains many unique cell types, then the SHN of that neighborhood is high. The “cell clusters” could be defined by either single cell clustering algorithm, or by cell type annotation. The “cellular neighborhood” could be defined by either k-nearest-neighbor like SOTIP-SHN, or by a user-defined radius.

IGD is a variant of SOTIP-SHN, and is also proposed as a control method for SHN quantification by this paper. We propose IGD to prove the advantage of using CGMGD in SOTIP-SHN. The only difference between IGD and SOTIP-SHN is that, SOTIP-SHN used the sum of CGMGD distance matrix within the cellular neighborhood, while IGD replaced the CGMGD distance matrix with IDENTITY distance matrix (please refer to “Spatial heterogeneity (SHN) quantification” section).

More explanations are in “Further explanations on SHN” in Supplementary Notes.

We also added following texts in Supplementary Notes, to explain more detailed in SHN and DME and control methods.

Further explanations on SHN

SOTIP’s SHN (Spatial heterogeneity quantification) module is used to quantify the gene expression variance of cells within a spatial neighborhood. Since each cell has a spatial neighborhood, containing a number of cells, the output of SHN is a scalar value per cell. If a cell’s associated spatial neighborhood contains cells with very different gene expression profiles, the SHN value of this cell is high, meaning spatially more heterogeneous. The only control method of SHN is NUCC (IGD mentioned in the original manuscript is not a previous method, but a variant of SOTIP), which simply regards the number of unique cell clusters within each neighborhood as spatial heterogeneity. SOTIP-SHN considers the gene expression variation within neighborhood, and NUCC considers the cell cluster variation within neighborhood. That being said, SOTIP puts the cell relationships into a gene expression space so that distance between cells could be computed by distance metrics between gene expression profiles, but NUCC puts the cell relationships into a one-hot cell type representation space so that the relative distances between cells on the gene expression manifold are missed. So, SOTIP-SHN has higher resolution than NUCC. Because SOTIP-SHN considers gene expression relationships between cells, it is not restricted by the accuracy of cell clustering, but the reliability of NUCC could be easily influenced by the accuracy of cell clustering. SOTIP-SHN is especially useful when continuity pronounced in gene expression space (as demonstrated in Figure 2a,b), or when it is difficult to choose a cell clustering resolution (as demonstrated in Figure 3i and Supplementary Figure 2a-d).

Further explanations on DME

SOTIP’s DME (differential microenvironment analysis) module is used to identify those microenvironments which differentiate between two spatial omics tissue sections. Since there are currently no other methods for spatial omics data that can do the same thing as SOTIP-DME, we did not compare SOTIP-DME with other spatial methods. However, there

exist non-spatial methods that can identify cell states that can differentiate between two samples, for example, MELD⁴, Milo⁵, and CNA⁶. The main difference between SOTIP-DME and these methods are that SOTIP-DME additionally utilizes the spatial information. Regarding this, SOTIP-DME would perform better than these single cell non-spatial methods when two samples share similar cell state composition but have different cell spatial organization. The TNBC case in Figure 7 has demonstrated this strength.

As to SDM, since there exist many methods which can also perform SDM task like SOTIP. We compared them in Supplementary Notes:

Comparison of SOTIP-SDM with other related methods.

For clustering procedure, SOTIP followed a hierarchical merging procedure, and the intermediate clustering result could be maintained and evaluated in just one-time running. While other three methods do not enjoy this benefit due to their different principles. Specifically, SpaGCN needs to initialize the cluster labels with predefined number of clusters, then applying the iterative clustering procedure to stochastically optimize parameters and cluster labels. As a probabilistic graphical model, BayesSpace also needs predefined number of clusters to formalize the model. It regarded the labels as hidden variable, then building a fully Bayesian model with Gaussian as conditional distribution and Markov random field as smooth prior. As to stLearn, it adopts either k-means or Louvain to perform clustering after generating the normalized graph adjacency matrix. Because the true number of clusters is not accessible, which is the common situation in real applications, all these three methods need to constantly re-run entire program to test different number clusters or resolution parameters.

For the extra information usage, as with BayesSpace, SOTIP does not require incorporate histological information (e.g., H&E) to obtain better performance than SpaGCN and stLearn. In the common situation, producing spatial omics data together with paired histological image needs additional efforts on experimental design and finer operations, so that many mainstream spatial omics researches would not actually provide paired or vertically adjacent histological images^{2,7-12}. What's more, the discontinuity of adjacent sections and the uncontrollable noise further complicate the utility of histological information.

We also compared SOTIP with Squidpy, one of the most widely used integrated framework in spatial omics:

Comparison between SOTIP and Squidpy

Squidpy integrates neighborhood enrichment analysis, co-occurrence analysis, interaction analysis, autocorrelation analysis, and spatially variable gene analysis, as well as image processing tools for histological image analysis. We compared SOTIP with Squidpy in Supplementary Table 10. Specifically, SOTIP's key principle is optimal transport, while Squidpy integrated a large number of existing algorithms into a unified framework. SOTIP provides functions such as Spatial heterogeneity quantification (SHN), Spatial domain identification (SDM), and Differential microenvironment analysis (DME). While Squidpy provides functions such as neighborhood enrichment analysis, co-occurrence analysis, interaction analysis, autocorrelation analysis, spatially variable gene analysis, and image processing.

And another potentially related method, RESEPT, for supervised spatial domain segmentation.

Comparison between SOTIP and RESEPT

This section is added in 1st revision.

RESEPT is a supervised deep learning method for spatial domain identification. According to RESEPT's manuscript (<https://doi.org/10.1101/2021.07.08.451210>) line 33~38, RESEPT firstly learns a three-dimensional embedding using a graph autoencoder from the spatial transcriptomics data. The embedding is then visualized by mapping as color channels in an RGB image and segmented with a supervised convolutional neural network model. The comparison between RESEPT and SOTIP is in Supplementary Table 10. The common part between RESEPT and SOTIP is the spatial domain identification module. The different part between them is two folds:

- (1) For spatial domain identification (SDM), SOTIP is an unsupervised method which does not require any external datasets for training, while RESEPT requires external datasets for training before spatial domain inference.
- (2) SOTIP is a versatile method, which can perform multiple tasks, in which SDM is one of them.

In our original manuscript, we have also explained the underlying algorithmic principles of existing methods in each task briefly. For example, we explained the existing methods in SHN as “Classical methods quantify the heterogeneity of an ME by counting the number of unique cell clusters within it (termed NUCC, Supplementary Table 1)16,33,35. Such practice assumes equal importance of different clusters, potentially leading to the inaccuracy of heterogeneity quantification. For example, they may fail to distinguish the heterogeneity difference between two MEs (one is composed of tumor cells and immune cells, and another is composed of two subsets of immune cells).” in original manuscript line 102~108, we explained the existing methods in SDM as “Existing methods can be divided into three categories based on their computational principles: Zhu et al.36 and BayesSpace37 (Supplementary Table 1) applied Hidden-Markov random field (HMRF) to model the spatial dependency of hidden variables associated with spots. SpaGCN38 (Supplementary Table 1) utilized existing structure in the field of graph convolutional network (GCN) to aggregate gene expression features according to the spatial graph. StLearn39 (Supplementary Table 1) normalized the spatial information, tissue histology and gene expression features to perform clustering. While these methods and other very recent preprints40-42 achieved good performance in their cases, they may lack proper interpretation for the parameters (e.g. weight trade-off between spatial prior and conditional distributions for HMRF-based methods) or features (e.g. features learned from deep learning based methods) of biological entities. In addition, they may not be flexible enough to be compatible with different spatial omics data type.” in original manuscript line 113~126. For DME, since there are no such methods with spatial omics data, so we explained the parallel methods with single cell omics data as “Algorithms for identification of changes in response to biological insult (e.g. disease, ontogeny, or experimental perturbations) have been developed based on single-cell RNA sequencing (scRNA-seq) data43-46. The advances of spatial omics technologies have revealed the phenomenon that tissues at different disease state exhibit similar cell type composition but distinct spatial organizations1,29,34, thus urging for developing algorithms to identify differential microenvironment among different conditions.” in original manuscript line 132~139.

Furthermore, we also conducted functionality comparison in Supplementary Table 4, algorithm principle in Supplementary Table 5, and applicability on different data types in Supplementary Table 7.

3. Based on the reviewer's advice, we have added some spatial omics datasets information to Figure 1. We have added supported spatial data in Figure 1. Details about how these data are used are shown in Supplementary Figure 13~15, and Supplementary Table 8.
4. We thank the reviewer to point this out. As a method article, the very first and most important thing is to use datasets with the ground truth label to validate method's accuracy and to benchmark against those existing methods. And the availability of the ground truth label is the exact principle how we choose which dataset to be used in which algorithm module.
 - As to 4i dataset (Figure 3a-f), we used it to benchmark SHN task since the dataset provided positive control of SHN (the nuclear envelope) as well as protein marker expression, similar for zebrafish dataset (Figure 3g-i).
 - As to seqFISH+ dataset (Figure 3m), it doesn't provide spatial domain labels, so we did not use it to benchmark SDM. seqFISH+ contains the cortex depth information and has single cell resolution, and we have previous knowledge from the BICCN MERFISH paper² about the relationship between SHN and cortical depth, so we used this dataset to perform SHN.
 - As to osmFISH dataset (Figure 3k and Figure 4e-h), it provides both the spatial domain label and the cortex depth information, so we use it to perform both SDM and SHN.
 - As to SpatialLIBD dataset (Figure 4a-d), which provided spatial domain label, so we used it to benchmark SDM. The spatial resolution of SpatialLIBD dataset is not at the single cell level, so we did not perform SHN on it even if it provided cortical depth information.
 - As to MIBI proteomics dataset (Figure 5g-h), we used it to benchmark SDM since it provides spatial domain information.
 - As to scMEP proteomics dataset (Figure 5d-e), we used it to benchmark SDM since it provided spatial domain information.

Above discussions wrapped up our principles in the choice of datasets and applications.

Supplementary Table 8. Datasets and biological systems (This table is added in 1st revision)

Module	Design of demonstration	Species	Tissue	Protocol	Detected molecule	Figure	Positive control
SHN	subcellular resolution	Human	HeLa cell line	4i	protein	Fig. 3a-f	The nuclear envelope should have highest spatial heterogeneity.
	cellular resolution; physiological case	Mouse	Brain cortex	osmFISH	mRNA	Fig. 3j-o	The spatial heterogeneity should have a gradient pattern towards deeper cortical layer.
		Mouse	Brain cortex	seqFISH+	mRNA		
	spot resolution; pathological case	Zebrafish	melanoma	10X Visium	mRNA	Fig. 3g-i	The tumor boundary should have highest spatial heterogeneity.
SDM	Physiological case; Brain	Human	Brain cortex	10X Visium	mRNA	Fig. 4a-d	Labeled region
		Mouse	Brain	osmFISH	mRNA	Fig. 4e-h	Labeled region

	pathological case; CRC	Human	colorectal carcinoma (p16)	scMEP	protein	Fig. 5d-e	Labeled region
		Human	colorectal carcinoma (p23)	scMEP	protein		
	pathological case; TNBC	Human	triple negative breast cancer (compartmentalized, p4)	MIBI	protein	Fig. 5g-h	Labeled region
		Human	triple negative breast cancer (compartmentalized , p9)	MIBI	protein		
DME	Normal tissue vs two subtypes of TNBC	Human	triple negative breast cancer (compartmentalized)	MIBI	protein	Fig. 7	The two subtypes should have different cell type spatial organization, and the difference should be consistent on other 30+ patients.
		Human	triple negative breast cancer (mixed)	MIBI	protein		
	Healthy liver vs Fibrotic liver	Mouse	Healthy liver	TOF-SIMS	metabolite	Fig. 6	There should be microenvironment difference between healthy and fibrotic liver. The H&E could be act as positive control.
		Human	Fibrotic liver	TOF-SIMS	metabolite		

DME module of
SOTIP

Disease: In Fig. 6a-i, we apply the DME module of SOTIP on spatial metabolomics data of fibrotic and healthy liver. We show that SOTIP-DME identifies microenvironments that tend to specifically occur in the fibrotic liver, which could be verified by H&E images.

Cancer: In Fig. 7a-j, we apply the DME module of SOTIP on spatial proteomics data of mixed and compartmentalized subtypes of TNBC. We show that SOTIP-DME identifies microenvironments that tend to specifically occur in the mixed subtype.

Validation: In Fig. 7k-n, in a larger patient cohort, we find a more occurrence of differential microenvironments in the mixed than in the compartmentalized.

Comment 3: Furthermore, SDM is a highly desirable task for spatial omics, but it was difficult to clearly understand the utility of SHN or DME—when one would want to use these tasks and why these tasks might be better employed than other methods—this needs to be better fleshed out.

Author Response: We totally agree that SHN and DME, which are rarely explored tasks in spatial omics field, should be better explained to more general readers. In our original manuscript, SHN and DME were described in Methods section, and in a technical manner, which might be difficult to follow. We added following descriptions in Methods and Supplementary Notes to make it easier to follow.

NUCC and IGD

NUCC¹⁻³ and IGD are both control methods for spatial heterogeneity (SHN) quantification. NUCC is a simple and widely used method. The main idea is to quantify the spatial heterogeneity around a cell by counting the number of unique cell clusters within the neighborhood. Its rationale is straightforward, if a cellular neighborhood contains many unique cell types, then the SHN of that neighborhood is high. The “cell clusters” could be defined by either single cell clustering algorithm, or by cell type annotation. The “cellular neighborhood” could be defined by either k-nearest-neighbor like SOTIP-SHN, or by a user-defined radius.

IGD is a variant of SOTIP-SHN, and is also proposed as a control method for SHN quantification by this paper. We propose IGD to prove the advantage of using CGMGD in SOTIP-SHN. The only difference between IGD and SOTIP-SHN is that, SOTIP-SHN used the sum of CGMGD distance matrix within the cellular neighborhood, while IGD replaced the CGMGD distance matrix with IDENTITY distance matrix (please refer to “Spatial heterogeneity (SHN) quantification” section).

More explanations are in “Further explanations on SHN” in Supplementary Notes.

Further explanations on the utility of SHN

SOTIP’s SHN (Spatial heterogeneity quantification) module is used to quantify the gene expression variance of cells within a spatial neighborhood. Since each cell has a spatial neighborhood, containing a number of cells, the output of SHN is a scalar value per cell. If a cell’s associated spatial neighborhood contains cells with very different gene expression profiles, the SHN value of this cell is high, meaning spatially more heterogeneous. The only control method of SHN is NUCC (IGD mentioned in the original manuscript is not a previous method, but a variant of SOTIP), which simply regards the number of unique cell clusters within each neighborhood as spatial heterogeneity. SOTIP-SHN considers the gene expression variation within neighborhood, and NUCC considers the cell cluster variation within neighborhood. That being said, SOTIP puts the cell relationships into a gene expression space so that distance between cells could be computed by distance metrics between gene expression profiles, but NUCC puts the cell relationships into a one-hot cell type representation space so that the relative distances between cells on the gene expression manifold are missed. So, SOTIP-SHN has higher resolution than NUCC. Because SOTIP-SHN considers gene expression relationships between cells, it is not restricted by the accuracy of cell clustering, but the reliability of NUCC could be easily influenced by the accuracy of cell clustering. SOTIP-SHN is especially useful when continuity pronounced in gene expression space (as demonstrated in Figure 2a,b), or when it is difficult to choose a cell clustering resolution (as demonstrated in Figure 3i and Supplementary Figure 2a-d).

Further explanations on the utility of DME

SOTIP’s DME (differential microenvironment analysis) module is used to identify those microenvironments which differentiate between two spatial omics tissue sections. Since there

are currently no other methods for spatial omics data that can do the same thing as SOTIP-DME, we did not compare SOTIP-DME with other spatial methods. However, there exist non-spatial methods that can identify cell states that can differentiate between two samples, for example, MELD⁴, Milo⁵, and CNA⁶. The main difference between SOTIP-DME and these methods are that SOTIP-DME additionally utilizes the spatial information. Regarding this, SOTIP-DME would perform better than these single cell non-spatial methods when two samples share similar cell state composition but have different cell spatial organization. The TNBC case in Figure 7 has demonstrated this strength.

Comment 4: In general, some novel features of SOTIP that are desirable compared to other algorithms are that it doesn't require pre-determined k for finding spatial domains. It is compatible with proteomics data and 3D data. It also identifies spatial domains more closely resembling ground truth compared to existing algorithms. The tutorials are also extremely useful, although there are some accessibility and clarity issues detailed below.

Author Response: We thank the reviewer for his/her positive comments regarding to SOTIP algorithm and the tutorials. Regarding the accessibility and clarity issues, we respond in detail below.

Comment 5: The development of algorithms for unsupervised analysis of spatial omics data is rapidly growing as more spatial omics data sets come online. There is a need and high interest in improving algorithms for spatial data and SOTIP offers a unique and versatile approach. However, this work may be better suited for a more methods-focused journal. As the manuscript is currently written, it is for a highly specialized spatial biology computational audience (for example, different types of spatial data are not explained well, current algorithms are not described well, and computational terms/metrics are not explained—it is assumed that the reader knows these spatial techniques, algorithms, metrics).

Author Response: We thank the reviewer for pointing this out. Since spatial omics technologies and computational methods are moving very fast, existing data types, algorithms, terms, and metrics are highly complex and heterogeneous. We are sorry to cause the readability issues. We next describe the necessary information regarding data types, current algorithms, terms/metrics, respectively.

1. **Data types.** We made a new table (Supplementary Table 8) which summarizes the datasets/modalities information, including module (which algorithm module of SOTIP was applied on this dataset), design of demonstration (what biological feature was contained in this dataset), species, Tissue, protocol, and detected molecule.

This new table provides some complimentary information for Supplementary Table 2 and 3.

2. **Current algorithms.** Please refer to our response to Comment #2.
3. **Terms/metrics.** We totally agree with the reviewer that explaining every term/abbreviation is very important for general readers. So we summarize all the terms/abbreviation/metrics used in this manuscript as Supplementary Table 9. So that readers can quickly search for the term when they are reading the main manuscript.

Supplementary Table 9. terms

Term	Explanation
ISS	In Situ Sequencing
smFISH	single-molecule Fluorescence In Situ Hybridization
MERFISH	Multiplexed Error-Robust Fluorescence In Situ Hybridization
seqFISH	Sequential Fluorescence In Situ Hybridization
ST	Spatial Transcriptomics
HDST	High-Definition Spatial Transcriptomics
4i	iterative indirect immunofluorescence imaging
CODEX	CO-DEtection by indeXing
MIBI-TOF	Multiplexed Ion Beam Imaging by Time-Of-Flight
IMC	Imaging Mass Cytometry
AFADESI-MSI	AirFlow-Assisted Desorption ElectroSpray Ionization Mass Spectrometry Imaging
SEAM	Spatial single nucleAR Metabolomics
ME	MicroEnvironment
KNN graph	We referred to reference #32
CCNR	Classical Cell Neighborhood Representation, and we referred to Supplementary Table 1 and reference #29~31
NUCC	Number of Unique Cell Clusters, and we referred to Supplementary Table 1 and reference #16,33,35
BayesSpace	It is the name of an existing algorithm, and we referred to Supplementary Table 1 and reference #37
HMMRF	Hidden-Markov Random Field
SpaGCN	It is the name of an existing algorithm, and we referred to Supplementary Table 1 and reference #38
GCN	Graph Convolutional Network
StLearn	It is the name of an existing algorithm, and we referred to Supplementary Table 1 and reference #39
scRNA-seq	single-cell RNA sequencing
SHN	Spatial Heterogeneity
SDM	Spatial DoMain
DME	Differential MicroEnvironment
MECN	Molecular-Expression-aware Cellular Neighborhood
MEG	MECN Graph
TNBC	Triple Negative Breast Cancer
MKT	Highly specific MECNs identified by DME
SIMS	Secondary Ion Mass Spectrometry
IGD	An algorithm for quantifying SHN, we referred to Supplementary Table 1, where it is further linked to Methods section for detailed explanation.
UMAP	A widely used manifold learning method, we referred to Reference #57
PHATE	A widely used manifold learning method, we referred to Reference #58
DNA	We did not define it since it has been widely known as Deoxyribonucleic acid.
Squidpy	A python package for processing spatial omics data, we referred to Reference #61
HeLa	We did not define it since it has been widely known as a class of cell line.
AUC	Area Under Curve
PCNA	Proliferating Cell Nuclear Antigen
RPS6	Ribosomal Protein S6
ER	Endoplasmic Reticulum

BRAFV600E	A melanoma zebrafish model, we referred to Reference #63
Leiden	An existing clustering algorithm, we referred to Reference #64
Pearson's r	Pearson correlation coefficient
BICCN	BRAIN Initiative Cell Census Network, and we referred to Reference #67
osmFISH	cyclic-ouroboros single molecule fluorescence in situ hybridization, and we referred to Reference #52
Spearman's ρ	Spearman's rank correlation coefficients
seqFISH+	evolution of sequential fluorescence in situ hybridization, and we referred to Reference #17
FOV	Field Of View
SpatialLIBD	A spatial transcriptomics dataset, we referred to Reference 54
DLPFC	DorsoLateral PreFrontal Cortex
k-means	A widely used clustering algorithm
Louvain	A widely used clustering algorithm, we referred to Reference #64
SC3	A widely used clustering algorithm, we referred to Reference #68
Giotto	A spatial omics data processing package, we referred to Reference #69
SEDR	An existing spatial domain identification algorithm, we referred to Reference #40
STAGATE	An existing spatial domain identification algorithm, we referred to Reference #41
SpatialPCA	An existing spatial domain identification algorithm, we referred to Reference #42
ARI	Adjusted Rand Index
MET	MicroEnvironment Trajectory
PAGA	One of the most widely used trajectory inference method, we referred to Reference #70
scMEP	A spatial proteomics technique, we referred to Reference #55
CRC	ColoRectal Carcinoma
FDR	False-Discovery Rate
DMA	Differential Microenvironment Analysis
EH plot	"entropy-of-ME-cluster (EMC)" versus "spatial heterogeneity (SHN)" plot, and we also referred to Methods for detailed explanation.
EMC	entropy-of-ME-cluster, and we also referred to Methods for detailed explanation.
NPC	Non-hepatic Parenchymal Cells
TILs	Tumor-infiltrating lymphocytes
EASI-FISH	3D spatial transcriptomics, we also referred to Reference #53
3D IMC	3D spatial proteomics, we also referred to Reference #91
WB ratio	Within-cluster-Between-cluster ratio
EMD	Earth mover's distance, and we also referred to Reference #96
CGMGD	Connectivity Guided Minimum Graph Distance, and we also referred to "Connectivity guided minimum graph distance (CGMGD)" in Methods section for detail explanation.
PGEV	Pairwise Gene Expression Variation, where we also explained how to compute it.
NCME	Normalized count of MECNs, where we also explained how to compute it.
t-SNE	A famous manifold learning algorithm, and we referred to Reference #100
DPT	One of the most widely used trajectory inference algorithm, and we referred to Reference #104
BCM	Binary Connectivity Matrix
UDM	UMAP Distance Matrix
CGG	Connectivity Guided cluster Graph
ForceAtlas	A manifold learning algorithm, and we referred to Reference #105
Splatter	A single cell simulation tool, and we referred to Reference #106

MELD	An algorithm to compute the relative likelihood of observing samples in different conditions. We referred to Reference #46
BH	Benjamini–Hochberg

Comment 6: As already mentioned, the authors should provide better descriptions of the tasks SOTIP is performing and why and how it compares to other algorithms. For example, there is no explanation of NUCC and IGD, which the authors are benchmarking against. How is SOTIP different from these algorithms?

Author Response: We totally agree with the reviewer that it is important to describe why and how SOTIP is compared on certain datasets. As a method article, the very first and most important thing is to use datasets with ground truth label to validate method's accuracy and to benchmark against existing methods. And the availability of ground truth label is the exact principle how we choose which dataset to be used in which algorithm module. For more details, please see our response in the Common #2 described above.

For the explanation of NUCC and IGD, we added "Other methods include NUCC16,33,35 and IGD (see Methods "NUCC and IGD")," in the manuscript where NUCC and IGD are firstly used.

The description of NUCC and IGD and other terms can also be assessed in Supplementary Table 9.

Furthermore, we have added "NUCC and IGD" in Methods and "Further explanations on SHN" in Supplementary notes.

We hope these changes may improve the readability, and we are sorry again for the inconvenience.

Comment 7: The authors should include a comparison between squidpy and SOTIP as squidpy uses a similar concept of nodes and distances between nodes to find similarities and dissimilarities.

Author Response: We thank the reviewer to point this out. Squidpy¹³ is an excellent framework for spatial omics analysis. Squidpy integrates neighborhood enrichment analysis, co-occurrence analysis, interaction analysis, autocorrelation analysis, and spatially variable gene analysis, as well as image processing tools for histological image analysis. We compared SOTIP with Squidpy in Supplementary Table 10. Specifically, SOTIP's key idea is the construction of MEG, while Squidpy integrated a large number of existing algorithms into a unified framework. SOTIP provides functions such as Spatial heterogeneity quantification (SHN), Spatial domain identification (SDM), and Differential microenvironment analysis (DME). While Squidpy provides functions such as neighborhood enrichment analysis, co-occurrence analysis, interaction analysis, autocorrelation analysis, spatially variable gene analysis, and image processing. We added this comparison into Supplementary Notes.

Comment 8: How does SOTIP compare with RESEPT (<https://www.biorxiv.org/content/10.1101/2021.07.08.451210v1>) for spatial pattern inference?

Author Response: We thank the reviewer to point this out. RESEPT is a supervised deep learning method for spatial domain segmentation. According to RESEPT's manuscript (<https://doi.org/10.1101/2021.07.08.451210>) line 33~38, RESEPT firstly learns a three-dimensional embedding using a graph autoencoder from the spatial transcriptomics data. The embedding is then visualized by mapping as color channels in an RGB image and segmented with a supervised convolutional neural network model. The comparison between RESEPT and SOTIP is in Supplementary Table 10. The common part between RESEPT and SOTIP is the spatial domain identification module. The different part between them is two folds:

- (1) For spatial domain identification (SDM), SOTIP is an unsupervised method which does not require any external datasets for training, while RESEPT is a supervised method, requiring external datasets for training before spatial domain inference. So these two methods are not comparable in performance, we have not seen RESEPT used as control in other unsupervised method papers (BayesSpace¹⁴, SC-MEB¹⁵, SpaGCN¹⁶, CCST¹⁷, STAGATE¹⁸) either.
- (2) SOTIP is a versatile method, which can perform multiple tasks, in which SDM is one of them.

Comment 9: A major benefit of BayesSpace is that it can be run across all samples for SDM such that the same clusters are identified across a large sample set. Can SOTIP be implemented across multiple samples or can it only be run tissue section by tissue section?

Author Response: The reviewer has raised an excellent point. BayesSpace is one of the most widely used methods in spatial omics analysis. In this revision, we revise the code of SOTIP to perform SDM on multiple samples in one run. The following figure shows the joint clustering on 3 samples. The step consists of removing the batch effect using harmony, followed by a multi-sample version of SOTIP-SDM. We provided the step-wise reproducible code in our github “SDM_Visium_cortex_multi.ipynb”.

Comment 10: A major focus in the field has been the integration of single cell and spatial omics data sets. In terms of DME, when relevant single cell data is also available, how does SOTIP outperform algorithms such as Cell2Location (<https://www.nature.com/articles/s41587-021-01139-4>) that are also able to employ microenvironment analyses using non-negative matrix factorization approaches (see Fig. 4).

Author Response: We thank the reviewer to point this out. We carefully read the Cell2Location paper. Cell2Location is an excellent method to infer the cell type composition of spots in low resolution spatially resolved transcriptomics data, e.g., Visium and ST. In Cell2Location manuscript, the control methods are RCTD, Stereoscope, Seurat V3, and SPOTlight (see Fig.4 in Cell2Location manuscript), which are all cell type decomposition methods. However, SOTIP's DME (differential microenvironment analysis) module is used to identify those microenvironments which differentiate between two spatial omics tissue sections. So, we think that they are two completely different methods performing different tasks, thus not comparable. Regarding method principle and functionality, we could contrast SOTIP with Cell2Location in Supplementary Table 10.

Supplementary Table 10. Additional comparison

	SOTIP	Squidpy	RESEPT	BayesSpace	Cell2Location
Method principle	Optimal transport	Integrated computational framework	Deep learning	Probabilistic graphical models	Probabilistic graphical models
Spatial heterogeneity quantification	√				
Spatial domain identification	√		√	√	
Differential microenvironment analysis	√				
neighborhood enrichment analysis		√			
co-occurrence analysis		√			
interaction analysis		√			
autocorrelation analysis		√			
spatially variable gene analysis		√			
Image processing		√			
Enable running on multiple samples	√ (revised)			√	
Resolution enhancement				√	
Cell type decomposition					√

Comment 11: As noted, the manuscript lacks clarity and is difficult to read. In general, the authors could improve the clarity of the manuscript in the following ways:

- a. Shorten the text in all sections and make statements more concise.
- b. Define all acronyms and avoid use of acronyms where possible
- c. Remove jargon and better explain in text technical terms that may be unfamiliar to readers such as specific algorithms, spatial-omics data sets, and computational terms.
- d. There are several spelling/grammar issues that should be corrected throughout the manuscript in all sections.
- e. Provide more details in figure legends and explain all arrows.
- f. Parts of results that are better suited for discussion (for example, lines 433-459, 745-773)

Author Response: We thank the reviewer very much for the valuable suggestions. We found that our texts in the main manuscript exceeds the text limit a lot. Following your suggestions, we moved some less important contents to discussion section or supplementary notes. We have also made some revisions about the difficulties in understanding technical terms, typos, and lack of legends. Our changes include but not limited as follows:

- We moved some of the discussions in “SOTIP stratifies known layers in brain tissues” to Supplementary Notes “Comparison of SOTIP-SDM with other related methods” section.
- Since spatial trajectory analysis is not one of the three main tasks of SOTIP, we moved section “SOTIP delineates microenvironment trajectory of human cerebral cortex” in result section to Supplementary Notes.
- Since the last paragraph of “SOTIP identifies highly specific MECNs associated with prognosis in subtypes of TNBC” in results section mainly focuses on literature supporting for the identified microenvironment differentiating TNBC subtypes. This paragraph is not very related with the method itself, so we moved this paragraph into “Literature validations for MECN 24 found in Figure 7” of Supplementary Notes.
- Since “SOTIP is scalable with 3D spatial omics datasets” in results section focuses the scalability, not the main content, so we moved it to Supplementary Nodes.
- We have created a new table to summarize all technique terms and explanations, please referred to Supplementary 9. Readers could look up from the table when reading the manuscript.
- We have corrected typos and added necessary figure legends and annotations in:
 - Figure 1i and legend.
 - Figure 2c and legend.
 - Figure 2d and legend.
 - Figure 2e arrow and legend
 - Figure 5 arrow definitions and legends.

With above revision, and other minor revision for more concise, we have finally reduced the texts in main manuscript (exclude Methods section) to around 7000 words.

Comment 12: There are several issues with code reproducibility:

a. Provide code for installation instructions

b. Unable to render the code on github. For example, these 3 files don't render:

https://github.com/TencentAILabHealthcare/SOTIP/blob/master/SOTIP_analysis/osmFISH_Cortex/SHN_osmFISH_MEwise.ipynb,

https://github.com/TencentAILabHealthcare/SOTIP/blob/master/SOTIP_analysis/MIBI_TNBC/DMA_TNBC.ipynb,

https://github.com/TencentAILabHealthcare/SOTIP/blob/master/SOTIP_analysis/Visium_Zebrafish/SHN_Visium_zebroFISH_A_diff_res.ipynb

c. Visium DLPFC figure 4c does not match the reproducible example on github

(https://github.com/TencentAILabHealthcare/SOTIP/blob/master/SOTIP_analysis/Visium_Cortex/SDM_Visium_cortex_best.ipynb)

d. Have the authors provided the exact code to reproduce all the figures in the manuscript?

Author Response: We thank the reviewer for pointing this out. We provided “how to install” SOTIP in our github (<https://github.com/TencentAILabHealthcare/SOTIP>), provided tutorial codes for SOTIP (shown in the following figure). We do not recommend to direct render the notebooks online, because there are large contents and images in each notebook file. Note that before running the jupyter notebook, one should clone the repo to his/her local machine to run the provided notebooks, by running (“git clone <https://github.com/TencentAILabHealthcare/SOTIP>”) in his/her terminal.

We have revised Fig. 4c to match the code in “SDM_Visium_cortex.ipynb”.

Tutorial

Please install Jupyter in order to open these notebooks.

For the step-by-step tutorial, please refer to:

https://github.com/TencentAILabHealthcare/SOTIP/tree/master/SOTIP_analysis/tutorial/

For the reproduction of paper's results, please refer to:

https://github.com/TencentAILabHealthcare/SOTIP/tree/master/SOTIP_analysis/

Please download demo datasets from following doi:

10.6084/m9.figshare.18516128.

How to install?

- git clone this repository
- python setup.py install
- conda install pyemd

SOTIP has been tested on

- System: CentOS
- Python: 3.8.0
- Python packages: numpy==1.21.2 pandas==1.3.4 scipy==1.7.1 matplotlib==3.4.3 seaborn==0.11.2 scanpy==1.8.2 squidpy==1.1.2 palettable==3.3.0 scikit-learn==1.0.1 networkx==2.6.3 shapely==1.8.0 pyemd==0.5.1

We have reorganized our github repo so that all original results, and results in this round of review round be reproduced. Please see the reproduce table in the front page of

<https://github.com/TencentAILabHealthcare/SOTIP> (as below):

Figure	Description	Notebook
Fig 2a-b	Simulation data1	link
Fig 2c-d	Simulation data2	link
Fig 2e-f	Simulation data3	link
Fig 3a-f	SOTIP-SNH on 4l data	link
Fig 3g-i	SOTIP-SHN on Zebrafish Visium data	link
Fig 3k,n	SOTIP-SHN on Mouse cortex osmFISH data	link
Fig 3m,o	SOTIP-SHN on Mouse cortex seqFISH+ data	link
Fig 4b-c	SOTIP-SDM on SpatialLIBD Visium data	link
Fig 4d	SOTIP-SDM on SpatialLIBD Visium benchmark	link
Fig 4g-h	SOTIP-SDM on Mouse cortex osmFISH benchmark	link
Fig 5e (right)	SOTIP-SDM on Human CRC scMEP data	link
Fig 5e (left)	SOTIP-SDM on Human CRC scMEP data	link
Fig 5f	SOTIP-SDM on Human CRC scMEP data	link
Fig 5h (top)	SOTIP-SDM on Human TNBC MIBI data	link
Fig 5h (bottom)	SOTIP-SDM on Human TNBC MIBI data	link
Fig 5i (top)	SOTIP-SDM on Human TNBC MIBI data	link1 link2 link3
Fig 5i (bottom)	SOTIP-SDM on Human TNBC MIBI data	link1 link2 link3
Fig 6	SOTIP-DME on SIMS liver data	link
Fig 7a-j	SOTIP-DME on MIBI TNBC data	link
Fig 7k-m	Validation of SOTIP-DME on MIBI TNBC data	link
Fig 7n	Validation of TNBC and survival analysis	link
SuppFig 1a-c	Simulation data 1	link
SuppFig 1d-g	Simulation data 2	link
SuppFig 1h-k	Simulation data 3	link
SuppFig 2a-b	Mouse cortex osmFISH data	link
SuppFig 2c-d	Mouse cortex seqFISH+ data	link
SuppFig 2e-f	Zebrafish Visium data	link
SuppFig 4g	Human TNBC MIBI data	link1 link2 link3
SuppFig 5	SOTIP-DME on SIMS liver data	link
SuppFig 6	SOTIP-DME on MIBI TNBC data	link
SuppFig 7	Validation of TNBC and survival analysis	link
SuppFig 8	Simulation data 4	link
SuppFig 11	Simulation data 5	link
SuppFig 12	Simulation data 6	link

Comment 13: MKT is not defined in the abstract line 46.

Author Response: We thank the reviewer to point this out. We have added the definition of MKT in abstract as “MKT (dominated by 3 cell types, i.e., Macrophage, Keratin+ tumor, and CD8 T cell)”

Comment 14: Provide a clearer and simpler explanation of the term ME (microenvironment).

Author Response: The ME (microenvironment) of a cell is defined using a set of cells within the spatial neighborhood of the cell.

Comment 15: Intro line 68-69 is not clear

Author Response: We are sorry for this. Original sentence “In cancer, researchers also reported that different patterns of immune infiltration were associated with patient survival.” We have replaced this sentence to “In cancer, researchers also reported that different degrees of immune cell infiltration were associated with patient survival”.

Comment 16: What do the blue dots along the x-axis in figure 1e represent?

Author Response: We thank the reviewer for pointing this out. In our original manuscript, we intended to show a schematic representation of SHNs to show that different spatial mixture of cells have different SHNs. But we realized that the information presented in our original diagram caused confusion for readers. To this end, we removed those blue dots and simply used three different spatial cell organizations to represent different SHNs. The revised Fig. 1 is shown as follows.

Figure 1 (added in 1st round revision). Schematic overview.

Overview of SOTIP. SOTIP takes 2D/3D spatial omics data (spatially resolved gene expression, protein or metabolic profiles) as input (a), then represents microenvironments (MEs) by incorporating both spatial (b) and molecular (c) information (second column) to build a MECN graph (d). Finally, multiple tasks can be performed based on the MECN graph, including spatial heterogeneity quantification (e), spatial domain identification (f), differential microenvironment analysis (g), and other downstream tasks (h). (i): The supported spatial data.

Comment 17: Figure 2d legend and figure don't match

Author Response: Thank you for pointing this out. We have corrected this issue.

Comment 18: The result and interpretation of Figure 2f is unclear.

Author Response: This is a great point. We have added the description on how to estimate the relative likelihood of observing each microenvironment in the two tissue samples as:

MELD analysis

MELD46 is an algorithm for estimating the relative likelihood of observing each cell state between different conditions (e.g., different disease state, before/after drug treatment or other experimental perturbations). In original MELD publication, the input of MELD is (1) the single cell graph on gene expression space, and (2) the condition label for each single cell. The output of MELD is the relative likelihood of observing each cell in each condition.

In our application (e.g., Figure 2f), we instead input the microenvironment graph, and the condition label for each microenvironment. In this way, by combining SOTIP's mathematical definition of microenvironment and MELD, we can estimate the relative likelihood of observing each microenvironment between different conditions.

Above content has been added in Methods section. On the other hand, we have added arrows and legends in Fig. 2f to make it clearer to show that our analysis highlighted the unique microenvironments in both sample 1 and sample 2.

Revised legend: The red arrow points to C1-C2 boundary in sample 1, and blue arrow points to C3-C1 boundary in sample 2.

Comment 19: Figure 3, subplot 'L' missing?

Author Response: We intended to skip the "L", since it could be easily messed with "I".

Comment 20: The authors should show spatial trajectory graph of sample 151673 in figure 4i

Author Response: As answered in Comment 11, we did some changes to make the manuscript content more concise. Since the spatial trajectory analysis is not one of the three main tasks of SOTIP, we moved section "SOTIP delineates microenvironment trajectory of human cerebral cortex" in result section to Supplementary Notes. And the figure 4i is moved to Supplementary Figure 3.

Comment 21: Line 492 PAGA has not yet been defined?

Author Response: Thank you for pointing this out! PAGA (partition-based graph abstraction). We added explanations when the word was first used. We have also added “PAGA analysis” section in Methods:

PAGA analysis.

PAGA is an algorithm for trajectory inference through a topology preserving map of single cells. In the original PAGA publication, the input of PAGA is (1) the single cell graph on gene expression space, (2) a clustering assignment for the single cell data. The output of PAGA is graph, in which each node is a cluster, and the edge between two nodes is the strength of connectivity (the connectivity could also be interpreted as similarity) between two clusters. In our application (e.g., Supplementary Figure 3), we instead input the microenvironment graph, and the clustering label. So that we can get the topological map of the targeted tissue.

Comment 22: Figure 5a-c needs more explanation on figure itself and in legend

Author Response: We have added more explanations for Figure 5a-c legend.

Comment 23: Figure 5d,e which is tumor and which is immune, red or black? Bounded regions are not labels. Arrows are not described.

Author Response: We have added this information in the legend. The black is tumor region. In d,e,g,h, all white arrows point to tumor regions.

Comment 24: All figures in figure 5 and corresponding supplementary figure should be shown on same axis orientation (e.g. rotate 90 degrees as needed)

Author Response: Thank you so much for pointing out this detailed issue! We corrected the figures accordingly.

Comment 25: Line 310- what is meant by clipped?

Author Response: This is a good question. By “clip”, we think it’s like to truncate those SHN values larger than the nuclear envelope’s SHN values as the nuclear envelope’s SHN values. It is a way to narrow down the range for display.

Comment 26: Figure 6c is not referred to in text

Author Response: We added the reference in the text.

Comment 27: Fig 7, “L” is missing?

Author Response: Similar with Figure 3. We skipped the “L” on purpose, since it could be easily messed with “I”.

Reviewer #3:

Comment 1: The manuscript described SOTIP, a suite of software tools for analyzing spatial omics data, where the molecular profiles of individual functional units in the tissue are linked to the spatial location of that unit. For technologies such as Visium and Slide-seq, the functional units are spots or beads used to capture the local RNA molecules, thus the RNAseq data for individual spots or beads are accompanied by their x-y coordinates. There is a strong need for computational tools that can analyze such data. SOTIP is designed to accomplish three main tasks: spatial heterogeneity (SHN) quantification, spatial domain (SDM) identification, and differential microenvironment (DME) analysis. The algorithms were described in Methods; while Results were devoted to reporting performance in various tasks, along with some comparisons with previously published software tools.

A brief note of the method. For data with N spots and P features, one can use the P -feature vectors to build spot-spot "distance" measures as one of the starting points to explore spatial heterogeneity. In SOTIP, every spot is decomposed as a mixture of t clusters, where the clusters have been predefined, and sometimes referred to as cell types. Thus a second way to build spot-spot distance measures is to use the t -element composition for each spot. SOTIP uses a third measure. For each spot, its neighboring k spots form a "graph" of k edges. The distance between Spot- i and Spot- j is by comparing graph- i and graph- j , where each graph has M edges, from the index spot to its M neighbors, and each of the $M+1$ nodes is characterized by the K -cluster composition. The rationale is that such a graph represents each spot's microenvironment. This is a good concept and, when used well, can complement each spot's innate properties as measured by the first two options (P features and K -cluster composition). The use of network-based distance measures is part of an original and significant area of research.

Author Response: We thank the reviewer for the positive comments on SOTIP. We think it is necessary to reach a consensus with the reviewer on the premise of the algorithm, and then discuss the follow-up detailed comments. The reviewers wrote a nice summary of the core methodology. To keep with the reviewer on the same page, we also write a summary of SOTIP as follows. In the following, we use "cells" to denote the functional units for ease of explanation, which can be freely replaced to "spots" in technologies like Visium or ST.

For any data with N cells and P features (e.g., gene expression vector with length P), traditional (non-spatial) clustering methods directly construct a graph for the N cells, by computing distances on the P feature vectors. Then Leiden algorithm is performed on this graph to get the clustering result. Such traditional clustering methods only take into account the gene expression profile of each cell, without considering the neighborhood information of each cell.

On the contrary, SOTIP considers both gene expression profiles of cells, and the neighborhood information of cells. The main idea of SOTIP is to represent the neighborhood of each cell (neighborhood indexed by cell- i is called MECN- i) as a histogram of cell types (following figure shows example of 3 MECNs characterized by histograms), then SOTIP uses the earth mover's distance (EMD, a distance metrics for histograms) based on an optimal transport principle to construct a graph (termed MEG) between all MECNs, and finally a clustering procedure is performed on MEG to get the clustering result (i.e., spatial domain identification result). Here, we must re-emphasize that MEG is the graph of all MECNs, each node of MEG is a MECN, and edge between nodes is the distance between two MECNs, which is the EMD distance between the two histograms characterizing the two MECNs. This echoes the definition of MEG when it is

firstly used in original manuscript line 149, and also answers part of the comment #11 of reviewer #3.

The key design of SOTIP is construction of the MECN graph (MEG) using EMD. Using this way, when measuring the distance between two MECNs (e.g., MECN-i and MECN-j, indexed by cell-i and cell-j), SOTIP considers how to design an optimal transportation plan on the gene expression manifold to move cells (characterized by cell type) within MECN-i to match cells within MECN-j, so that the histogram characterizing MECN-i after the transportation is exactly the same as the histogram characterizing MECN-j. Another key design is that when designing the optimal transportation plan, the cells must be moved along the gene expression manifold. The optimal transportation cost is used as the distance between MECN-i and MECN-j. One can imagine that the transportation cost is determined by two factors: (1) the number of cells to be moved, and (2) the cost of moving cells between two positions along the gene expression manifold. The first factor has already been encoded in the histograms associated with MECNs, and the second factor (named ground distance in optimal transport theorem) is the geodesic distance between cell types along the gene expression manifold, the CGMGD in our manuscript. This process is formulated in line 929~939 of the original manuscript.

We hope the above explains the main idea of SOTIP and thank you for reading! Your comments have greatly help us to improve our work. In the following, we will respond them one by one.

Comment 2: A minor confusion: The graph-to-graph connectivity-based distance is defined in line.933. What is confusing is that the same distance seems to also used to calculate edge-distance within each graph (line.954).

Author Response: We thanks the reviewer to point this out. We would like to explain this using the optimal transport theory.

In line 933 of the original manuscript, we described how to define the distance between two MECNs. Each MECN is represented by a *histogram* of cell types (clusters) within the spatial neighborhood of an index cell. The distance between two MECNs could be computed as the distance between the two *distributions*. We chose the earth mover’s distance (EMD, a distance metrics for measuring the distance between two probabilistic distributions, the one-dimensional variant of Wasserstein distance) to measure the distributions between MECNs (as seen in the screenshot image below). The “CGMGD” which we defined in line 1018, which captures the relationships between cell types in the gene expression manifold, is used as the ground distance for earth mover’s distance. In this way, we can compute the pairwise distances between all pairs of MECNs, thus forming the MECN graph (MEG) defined in our manuscript. The MEG could capture the relationships between microenvironments indexed by all cells in the tissue sample, which is then used for clustering and other downstream analysis.

933 distance between two MECNs, e.g. $MECN_p$ and $MECN_q$ is defined as:

$$dist(MECN_p, MECN_q) = \frac{\sum_{i=1}^t \sum_{j=1}^t f_{ij} CGMGD(i, j)}{\sum_{i=1}^t \sum_{j=1}^t f_{ij}}$$

934 Where $CGMGD(i, j)$ is precomputed between all pairs of cell clusters
 935 according to Methods section “Connectivity guided minimum graph distance
 936 (CGMGD)”. And $\sum_{i=1}^t \sum_{j=1}^t f_{ij} CGMGD(i, j)$ is minimized subject to

$$f_{ij} \geq 0, i, j \in [1, t],$$

$$938 \quad \sum_{i=1}^t f_{ij} \leq w_j^{MECN_q}, i, j \in [1, t], q \in [1, n],$$

$$939 \quad \sum_{j=1}^t f_{ij} \leq w_i^{MECN_p}, i, j \in [1, t], p \in [1, n].$$

As to line 954 that you mentioned (as shown in the following screenshot), we guess your confusion is that why CGMGD is also used in computing PGEV, which is further summed up to compute the SHN of each microenvironment. Since SHN measures the spatial heterogeneity of each microenvironment, we mathematically define SHN as the total discrepancy of cell types within the microenvironment, e.g., sum of pairwise CGMGD. The reason why CGMGD could reflect the cell type discrepancy is that it captures the geodesic distance along the gene expression manifold, this is also the reason why CGMGD can be used as the ground distance when computing the EMD between MECNs.

$GROUP(c_i) = T_i, i \in [1, k], T_i \in [1, t]$

949 Since we have also precomputed the pairwise CGMGD among cells, and the
 950 CGMGD between two single cells is a function of their belonging cell groups.
 951 We denote the pairwise gene expression variation (PGEV) as:

$$PGEV(c_i, c_j) = CGMGD(GROUP(c_i), GROUP(c_j)), i, j \in [1, k]$$

952 The SHN of a MECN is computed as the total gene expression variation by
 953 summing over i and j :

$$SHN(MECN) = \sum_{c_i, c_j \in CONSIST(MECN)} PGEV(c_i, c_j)$$

954 When estimating SHN with an isotropic ground distance (IGD), the pairwise
 955 gene expression variation is simply replaced with:

$$PGEV_{IGD}(c_i, c_j) = IDENTITY(GROUP(c_i), GROUP(c_j)), i, j \in [1, k]$$

956 Where IDENTITY is an indicator function defined as:
 --- $IDENTITY(p, q) = \mathbf{1}_{p=q}(p, q)$

In summary, in SOTIP, we used two distances, one is the earth mover's distance (EMD), which is used to compute the distance between two MECNs, the other is the CGMGD, which is used to compute the geodesic distance along the gene expression manifold. The two distances are different in scale, i.e., EMD is the distance between MECNs (MECN scale), and CGMGD is the distance between cells within MECNs (cell scale). The two distances are closely related, i.e., CGMGD is used as the "ground distance" when computing EMD.

Comment 3: The paper has important elements in the development of the algorithms. However, they are not always presented in an easy-to-understand fashion. The exploration of the algorithms may influence the thinking of the field. The claim that the package has better performance is difficult to evaluate, and not likely to have a strong impact.

Author Response: We thank the review for the positive comments on the algorithm, and also thanks for the criticism on the readability issue. We have addressed some of the readability issues in responses to the following detailed comments.

Comment 4: The descriptions of the algorithms in Methods were not easy to follow, especially with regard to the rationale and the innovation when compared to past efforts. It should be considered to explain the methods first.

Author Response: The reviewer has raised an important point. The response to Comment #1 and Comment #2 have explained some important elements of SOTIP, especially SOTIP-SDM. What's more, we added two sections in Supplementary Notes to further explain SHN and DME's, principle and past efforts i.e., "Further explanations on SHN" and "Further explanations on DME".

Further explanations on SHN

SOTIP's SHN (Spatial heterogeneity quantification) module is used to quantify the gene expression variance of cells within a spatial neighborhood. Since each cell has a spatial neighborhood, containing a number of cells, the output of SHN is a scalar value per cell. If a cell's associated spatial neighborhood contains cells with very different gene expression profiles, the SHN value of this cell is high, meaning spatially more heterogeneous. The only control method of SHN is NUCC (IGD mentioned in the original manuscript is not a previous method, but a variant of SOTIP), which simply regards the number of unique cell clusters within each neighborhood as spatial heterogeneity. SOTIP-SHN considers the gene expression variation within neighborhood, and NUCC considers the cell cluster variation within neighborhood. That being said, SOTIP puts the cell relationships into a gene expression space so that distance between cells could be computed by distance metrics between gene expression profiles, but NUCC puts the cell relationships into a one-hot cell type representation space so that the relative distances between cells on the gene expression manifold are missed. So, SOTIP-SHN has higher resolution than NUCC. Because SOTIP-SHN considers gene expression relationships between cells, it is not restricted by the accuracy of cell clustering, but the reliability of NUCC could be easily influenced by the accuracy of cell clustering. SOTIP-SHN is especially useful when continuity pronounced in gene expression space (as demonstrated in Figure 2a,b), or when it is difficult to choose a cell clustering resolution (as demonstrated in Figure 3i and Supplementary Figure 2a-d).

Further explanations on DME

SOTIP's DME (differential microenvironment analysis) module is used to identify those microenvironments which differentiate between two spatial omics tissue sections. Since there is currently no other methods for spatial omics data that can do the same thing as SOTIP-DME, we did not compare SOTIP-DME with other spatial methods. However, there exist non-spatial methods that can identify cell states that can differentiate between two samples, for example, MELD⁴, Milo⁵, and CNA⁶. The main difference between SOTIP-DME and these methods are that SOTIP-DME additionally utilizes the spatial information. Regarding this, SOTIP-DME would perform better than these single cell non-spatial methods when two samples share similar cell state composition but have different cell spatial organization. The TNBC case in Figure 7 has demonstrated this strength.

Comparison of SOTIP-SDM with other related methods.

For clustering procedure, SOTIP followed a hierarchical merging procedure, and the intermediate clustering result could be maintained and evaluated in just one-time running. While other three methods do not enjoy this benefit due to their different principles. Specifically, SpaGCN needs to initialize the cluster labels with predefined number of clusters, then applying the iterative clustering procedure to stochastically optimize parameters and cluster labels. As a probabilistic graphical model, BayesSpace also needs predefined number of clusters to formulize the model. It regarded the labels as hidden variable, then building a fully Bayesian model with Gaussian as conditional distribution and Markov random field as smooth prior. As to stLearn, it adopts either k-means or Louvain to perform clustering after generating the normalized graph adjacency matrix. Because the true number of clusters is not accessible, which is the common situation in real applications,

all these three methods need to constantly re-run entire program to test different number clusters or resolution parameters.

For the extra information usage, as with BayesSpace, SOTIP does not require incorporate histological information (e.g., H&E) to obtain better performance than SpaGCN and stLearn. In the common situation, producing spatial omics data together with paired histological image needs additional efforts on experimental design and finer operations, so that many mainstream spatial omics researches would not actually provide paired or vertically adjacent histological images^{2,7-12}. What's more, the discontinuity of adjacent sections and the uncontrollable noise further complicate the utility of histological information.

We also added discussions on SOTIP and past efforts in Supplementary Notes:

Comparison between SOTIP and Squidpy

Squidpy integrates neighborhood enrichment analysis, co-occurrence analysis, interaction analysis, autocorrelation analysis, and spatially variable gene analysis, as well as image processing tools for histological image analysis. We compared SOTIP with Squidpy in Supplementary Table 10. Specifically, SOTIP's key principle is optimal transport, while Squidpy integrated a large number of existing algorithms into a unified framework. SOTIP provides functions such as Spatial heterogeneity quantification (SHN), Spatial domain identification (SDM), and Differential microenvironment analysis (DME). While Squidpy provides functions such as neighborhood enrichment analysis, co-occurrence analysis, interaction analysis, autocorrelation analysis, spatially variable gene analysis, and image processing.

Comparison between SOTIP and RESEPT

RESEPT is a supervised deep learning method for spatial domain identification. According to RESEPT's manuscript (<https://doi.org/10.1101/2021.07.08.451210>) line 33~38, RESEPT firstly learns a three-dimensional embedding using a graph autoencoder from the spatial transcriptomics data. The embedding is then visualized by mapping as color channels in an RGB image and segmented with a supervised convolutional neural network model. The comparison between RESEPT and SOTIP is in Supplementary Table 10. The common part between RESEPT and SOTIP is the spatial domain identification module. The different part between them is two folds:

- (1) For spatial domain identification (SDM), SOTIP is an unsupervised method which does not require any external datasets for training, while RESEPT requires external datasets for training before spatial domain inference.
- (2) SOTIP is a versatile method, which can perform multiple tasks, in which SDM is one of them.

Comment 5: Spatial heterogeneity quantification (the first task) is based on the total edge distance among the k neighboring cells. The better performance of the connectivity-based distance is shown in a simulation (Figure 2B). I am not convince because the use of the original P-feature vector would also perform well in the situation in Figure 2B.

Author Response: We thank the reviewer for the great question.

Figure 2b is the performance comparison of NUCC (left), IGD (middle) and SOTIP (right) of SHN quantification on simulation data 1.

As shown in the PCA of figure 2a, C1 and C2 are close in molecular expression, far different from C3. It makes the heterogeneity in C1-C2 boundary higher than that in C2-C3 boundary.

The goal of SHN module of SOTIP is not only to identify the two boundaries, i.e., C1-C2 boundary as well as C2-C3 boundary, but to distinguish the heterogeneity difference between them, which may be linked to biological function as we showed in the cases of Fig. 3a-f and Fig. 3g-i.

For the goal to distinguish the level of heterogeneity between boundaries, we can see in Fig. 2b that, only SOTIP achieved it. In the top panel of Fig. 2b, SOTIP showed a weak peak and a strong peak of SHN value, representing the low heterogeneity of C1-C2 boundary and high heterogeneity of C2-C3 boundary. NUCC and IGD failed to tell the difference between C1-C2 and C2-C3 in spatial heterogeneity by giving two peaks with equal SHN values.

Comment 6: The second task, spatial domain identification, has a major conceptual issue. When each of the N spots has been recoded as a graph with k edges, and each of the k nodes has been recoded as a histogram of t components, the "domain" is found by iteratively clustering the N spots by their connectivity-based distance, resulting in a hierarchical clustering tree. As described in lines.958-985, the spatial relationship of the spots are not used, therefore the "neighboring" spots on the tree may or may not be near each other, to form a domain. It is the reader's guess that the number of clusters will come from a tree-cutting method, say have T clusters by cutting the tree at the level of K branches. But they will not form K domains as the spots within a cluster are likely non-contiguous in the tissue. If spatial proximity is used in the algorithm, it is not described. And if it is indeed used, the relative weight of spatial distance and the connectivity - based distance is the key part of the method. Many existing tools have confronted this issue. Besides, clustering of the N spots can be done by using the original P features or the dimension-reduced alternatives, such as the composition of the t clusters. The relative merits of these alternatives need to be better explored, first in simulation, before moving to real data.

Author Response: We thank the reviewer for raising this confusion. The reviewer's confusion lies in (1) whether SOTIP uses the spatial relationship between spots; (2) how SOTIP uses spatial information; and (3) how SOTIP handles the relative weight of spatial distance and gene expression distance. We will try to answer them one by one.

(1) Whether SOTIP uses the spatial relationship between spots?

Yes. When constructing MECN, the spatial relationship is used.

(2) How SOTIP uses spatial information?

On one hand, the spatial information is used when constructing MECNs. For example, MECN-i indexed by spot-i is constructed by finding the k closest spots to spot-i. When finding the k closest spots, the spatial relationship is used. On the other hand, since each MECN is characterized by a histogram of cell types, so the relative spatial relationships within one MECN is not used. Since our choice of k is 10 by default (as described in "Parameter settings" in Methods section), spots within one MECN are typically near each other, which means that spatial contiguity is guaranteed. Even if k is set to a larger value so that the relative spatial relationships within one MECN could not be trivially ignored, SOTIP can still encourage spatial contiguity. This is because SOTIP-SDM is performed on MECNs (each is characterized by a histogram of cell types), and two MECNs near each other would always share similar histograms of cell types (since spots covered by MECN-i are highly overlapped with those spots covered by MECN-j if MECN-i and MECN-j are spatially near with each other).

(3) How SOTIP handles the relative weight of spatial distance and gene expression distance?

SOTIP does not need to handle the relative weight of spatial distance and gene expression distance. And this is a strength for SOTIP compared with other existing tools. As described above, the procedure of SDM identification consists of the following key steps, and all steps does not need to handle the weight.

(a) Clustering on MEG. MEG is defined as MECN graph, each node of the graph is a MECN, and the edge between two nodes is the EMD distance between the histograms that characterize two MECNs).

(b) Constructing MEG. This need definition of the node and the edge of MEG.

(c) The node of MEG is MECN. Constructing MECN does not involve the weight between the two distances.

(d) The edge of MEG is defined by EMD distance between MECNs. Computing EMD distance between MECNs does not involve the weight between the two distances.

Comment 7: The third task, differential microenvironment analysis, requires significant clarification. As described in lines.986-1016, sometimes the comparison seems to be between two different tissue samples, while some other times it seems to be between two different "clusters" within a sample, here the "clusters" seem to be the "domains" found in task-2, not the t spot clusters used in the early steps. The analysis is essentially a visualization in EH plot. If there is a test statistics also proposed, it was not easy to find in the text.

Author Response: We thank the reviewer to pointing this out. We believe the reviewer's confusions would be common for general readers, so it is very important to clarify them. These confusions could be summarized as 2 points: (1) Is the comparison between two different tissue samples, or between two different clusters within a sample? (2) Is there a statistical testing proposed associated with EM plot?

To solve these, we firstly answer the two questions one by one, and then make changes on DME's text in Methods (line 986~1016 in original manuscript) to make it clearer.

(1) Is the comparison between two different tissue samples, or between two different clusters within a sample?

The comparison is between two different tissue samples, to identify those MECN clusters that are more likely to appear in a specific sample. Figure 7 is a good case for DME application. In Figure 7, we used DME to compare between two subtypes (i.e., compartmentalized vs mixed) of triple negative breast cancer (TNBC), and found one class of MECN (consisted with CD8T cell, macrophage, and Keratin positive tumor cell) is specifically presented in the mixed subtype. This different between compartmentalized vs mixed subtypes could not be identified by non-spatial methods, since the cell type component between the two subtypes are similar.

(2) Is there a statistical testing proposed associated with EM plot?

No. The analysis is by visualization in EH plot. A smaller "E" represents more specificity, and a larger "H" represents more spatial heterogeneity. So the MECN clusters lying at the bottom right corner of EH plot have small E (which means have big tendency to be presented in specific condition) and big H (which means the cell types within the MECNs in the cluster are not homogeneous).

To make the description of DME better understood by readers, we added the following texts in "Differential microenvironment (DME) analysis" section in Methods:

To summarize, the input of SOTIP-DME is two tissue samples (spatial omics data), and the output is the EH plot. Using the EH plot, each point is a MECN cluster, one can visually assess those MECN clusters with high sample-specificity and high spatial heterogeneity.

Comment 8: The choice of the distance measures discussed in "Connectivity guided minimum graph distance (CGMGD)" is the foundation of SOTIP. The advantages and disadvantages of the distance measures need to be extensively examined.

Author Response: This is a great point. To compute the geodesic distances between cell clusters on the gene expression manifold, there are 3 potential ways:

1, find geodesic path like in Isomap¹⁹. For example, in the following figure, the manifold distance between cell A and cell B should be the solid red curve, rather than the dashed red line.

We did not use Isomap because:

- (1) There is currently no implementation of Isomap that returns the pairwise geodesic distance between all cells.
- (2) Even if one implementation returns pairwise geodesic distance, the resolution is per cell. SOTIP needs geodesic distances between clusters, rather than cells, which means Isomap implementation would be inefficient.

2, compute Euclidean distances on low-dimensional embedding. This practice is used in another recent paper to compute the manifold distance of single cells²⁰. But the problem is that the low-dimensional embedding typically retains local data structure (especially non-linear methods such as t-sne, umap, and diffusion-map). Global preservation method such as PCA would “bend” the original continuous manifold. These problems have been demonstrated in Supplementary Fig. 8a-f.

3, first set a “root” cell, then use single cell pseudo-time methods, to assign a pseudo-time value to each cell, so that the difference between the estimated pseudo-time of two cells could reflect the manifold distance. This practice has problems, preventing us to exploit it in SOTIP. We use diffusion pseudo-time (DPT), the most widely used pseudo-time inference method in single cells.

- (1) If cells are clustered as N groups, one need to set the “root cell” for N times, for each time, the DPT need to be run. So it is time-consuming.
- (2) Since SOTIP needs cluster-to-cluster distances, so even if the time is not the problem, one need to select the representative cell for each cluster, to assign as root cell, or to compute offset of the pseudo-time. Supplementary Fig. 8g shows the

- distance matrix of clusters computed using this practice, in which the representative cell for each cluster is computed as the center of cluster.
- (3) Pseudo-time computed by different runs of DPT could not be compared, so the generated distance matrix of clusters does not guarantee the three conditions of distance metrics (Supplementary Fig. 8g).

Aforementioned points explain why we need to propose CGMGD as SOTIP's ground distance. There are of course disadvantages of CGMGD.

- 1, Since CGMGD relies on permutation test-based connectivity computation, so each cluster should not be too small.
- 2, Since CGMGD needs to compute the shortest distances on graph, the time cost might be a problem if the graph grows large.

Comment 9: The Results section focused on examples of application. I did not go through them in detail. There are several difficulties. One is the degree of difficulty of knowing the true signal. Some examples are "easy", such as finding different cortical layers have different spatial properties (Figure 3j-o). What constitutes a relevant improvement of performance is not easy to define.

Author Response: We are sorry to cause the reading difficulties because of the complex datasets, true signals, as well as the diverse biological system applications.

To demonstrate SOTIP's applicability and versatility, the amount and diversity of different datasets and biological systems is necessary. For better clarity, we summarized the datasets/modalities information in Supplementary Table 8. In that table, every dataset used in the manuscript is described in terms of 6 aspects, i.e., Module (which algorithm module of SOTIP was applied on this dataset), Design of demonstration (what biological feature was contained in this dataset), Species, Tissue, Protocol, Detected molecule, and Positive control (true signals).

Supplementary Table 8. Datasets and biological systems (This table is added in 1st revision).

Module	Design of demonstration	Species	Tissue	Protocol	Detected molecule	Figure	Positive control
SHN	subcellular resolution	Human	HeLa cell line	4i	protein	Fig. 3a-f	The nuclear envelope should have highest spatial heterogeneity.
	cellular resolution; physiological case	Mouse	Brain cortex	osmFISH	mRNA	Fig. 3j-o	The spatial heterogeneity should have a gradient pattern towards deeper cortical layer.
		Mouse	Brain cortex	seqFISH +	mRNA		
	spot resolution; pathological case	Zebrafish	melanoma	10X Visium	mRNA	Fig. 3g-i	The tumor boundary should have highest spatial heterogeneity.
SDM	Physiological case; Brain	Human	Brain cortex	10X Visium	mRNA	Fig. 4a-d	Labeled region
		Mouse	Brain	osmFISH	mRNA	Fig. 4e-h	Labeled region
	pathological case; CRC	Human	colorectal carcinoma	scMEP	protein	Fig. 5d-e	Labeled region

			(p16)				
		Human	colorectal carcinoma (p23)	scMEP	protein		
	pathological case; TNBC	Human	triple negative breast cancer (compartmentalized, p4)	MIBI	protein	Fig. 5g-h	Labeled region
		Human	triple negative breast cancer (compartmentalized, p9)	MIBI	protein		
DME	Normal tissue vs two subtypes of TNBC	Human	triple negative breast cancer (compartmentalized)	MIBI	protein	Fig. 7	The two subtypes should have different cell type spatial organization, and the difference should be consistent on other 30+ patients.
		Human	triple negative breast cancer (mixed)	MIBI	protein		
	Healthy liver vs Fibrotic liver	Mouse	Healthy liver	TOF-SIMS	metabolite	Fig. 6	There should be microenvironment difference between healthy and fibrotic liver. The H&E could be act as positive control.
		Human	Fibrotic liver	TOF-SIMS	metabolite		

To explain the biological systems. We added three figures, explaining the biological systems in SOTIP's three modules, respectively. Supplementary Figure 13 for SHN module, Supplementary Figure 14 for SDM module, and Supplementary Figure 15 for DME module.

Supplementary Figure 13

Supplementary Figure 13 (added in 1st round revision). Demonstration logic of SHN module

Supplementary Figure 14

Supplementary Figure 15

Supplementary Figure 15 (added in 1st round revision). Demonstration logic of DME module

Besides, in our original manuscript, we described every dataset when they are firstly used in the manuscript, we also provided schematic image for the main datasets we used (Figure 3a, 4a, 4e, 5a, 6a, 7a).

Comment 10: I did not evaluate the quality of the codes but appreciated that the link to GitHub was provided.

The simulations will be more helpful if they are described in greater detail: what type of statistical properties was embedded to probe which type of performance.

Author Response: We thank the reviewer for this comment. We have updated the simulation information in Supplementary Table 2. The last column “Purpose” stated what kind of properties is to be tested.

Supplementary Table 2: Simulation data summary.

Name	Tool	Cell distribution	Cell composition	Spatial distribution	Purpose
Simulation 1	Splatter ²¹	Two continuous groups and one discrete group, Supplementary Figure 1a	Supplementary Figure 1b	Randomly positioned these three clusters of cells as three sequential bands shaped tissue bulk on a two-dimensional plane. Supplementary Figure 1c.	In order to verify the SHN methods could not only identify the existence of SHN peaks, but also distinguish the differences in SHN values.
Simulation 2	Splatter ²¹	5 continuous groups (C1-C5), Supplementary Figure 1d	Supplementary Figure 1e	Regularly mixed these 5 clusters in 5 sequential bands on a two-dimensional plane (Fig. 1d right). In this data, each cell type occupies a major component in a band, and other cell types occupy minor components, and the order of major cell types along these tissue bands (R1 to R5) is agreed with the order of cell types along the gene expression manifold (C1 to C5) (Fig. 1d middle).	In order to verify which graph construction method can be used to restore the structured manifold of microenvironments.

				Supplementary Figure 1f,g	
Simulation 3	Splatter ²¹	3 discrete groups, Supplementary Figure 1h	Supplementary Figure 1i	Positioned the 3 cell clusters with two different order of sequential bands. Supplementary Figure 1j,k	To prove MEG could be used to identify microenvironments which differentiate between samples
Simulation 4	Splatter ²¹	20 continuous groups, Supplementary Figure m	Supplementary Figure 1m	None	Performance of CGMGD

Comment 11: The terminology can be much improved. "Single cell" was used to refer to spots. The difference between MECN and MEG took repeated reading to understand. Differential analysis of microenvironment can happen on multiple scales. As written it is unclear which scale does the comparison take place.

Author Response: We totally agree with the author that the terminology should be clearer.

(1) Single cell and spots.

We have double checked the usage of single cells and spots in the biological cases. In methods where the algorithm is described, we uniformly use spots for the reason of generalization.

(2) MECN and MEG.

We are so sorry to cause the inconvenience. MEG is the graph of all MECNs, each node of MEG is a MECN, and edge between nodes is the distance between two MECNs, which is the EMD distance between the two histograms characterizing the two MECNs. This echoes the definition of MEG when it is firstly used in original manuscript line 149.

(3) DME.

We have updated the DME description part in Methods section.

Besides, given that SOTIP is an interdisciplinary work, the terms used in the manuscript may be over-complexed. We think even if every term or abbreviation has already been explained when it is first used in the manuscript, readers would still have difficulties when reading. To solve this problem, we added a big table in Supplementary Table 9, to summarize all the definitions of terms and abbreviations in the manuscript, so that readers can quickly look up in their reading.

Supplementary Table 9. Terms (This table is added in 1st revision)

Term	Explanation
ISS	In Situ Sequencing
smFISH	single-molecule Fluorescence In Situ Hybridization
MERFISH	Multiplexed Error-Robust Fluorescence In Situ Hybridization
seqFISH	Sequential Fluorescence In Situ Hybridization
ST	Spatial Transcriptomics
HDST	High-Definition Spatial Transcriptomics
4i	iterative indirect immunofluorescence imaging
CODEX	CO-DEtection by indeXing
MIBI-TOF	Multiplexed Ion Beam Imaging by Time-Of-Flight
IMC	Imaging Mass Cytometry
AFADESI-MSI	AirFlow-Assisted Desorption ElectroSpray Ionization Mass Spectrometry Imaging
SEAM	Spatial single nuclEAR Metabolomics
ME	MicroEnvironment
KNN graph	We referred to reference #32
CCNR	Classical Cell Neighborhood Representation, and we referred to Supplementary Table 1 and reference #29~31
NUCC	Number of Unique Cell Clusters, and we referred to Supplementary Table 1 and reference #16,33,35
BayesSpace	It is the name of an existing algorithm, and we referred to Supplementary Table 1 and reference #37
HMMRF	Hidden-Markov Random Field
SpaGCN	It is the name of an existing algorithm, and we referred to Supplementary Table 1 and reference #38
GCN	Graph Convolutional Network

StLearn	It is the name of an existing algorithm, and we referred to Supplementary Table 1 and reference #39
scRNA-seq	single-cell RNA sequencing
SHN	Spatial Heterogeneity
SDM	Spatial Domain
DME	Differential MicroEnvironment
MECN	Molecular-Expression-aware Cellular Neighborhood
MEG	MECN Graph
TNBC	Triple Negative Breast Cancer
MKT	Highly specific MECNs identified by DME
SIMS	Secondary Ion Mass Spectrometry
IGD	An algorithm for quantifying SHN, we referred to Supplementary Table 1, where it is further linked to Methods section for detailed explanation.
UMAP	A widely used manifold learning method, we referred to Reference #57
PHATE	A widely used manifold learning method, we referred to Reference #58
DNA	We did not define it since it has been widely known as Deoxyribonucleic acid.
Squidpy	A python package for processing spatial omics data, we referred to Reference #61
HeLa	We did not define it since it has been widely known as a class of cell line.
AUC	Area Under Curve
PCNA	Proliferating Cell Nuclear Antigen
RPS6	Ribosomal Protein S6
ER	Endoplasmic Reticulum
BRAFV600E	A melanoma zebrafish model, we referred to Reference #63
Leiden	An existing clustering algorithm, we referred to Reference #64
Pearson's r	Pearson correlation coefficient
BICCN	BRAIN Initiative Cell Census Network, and we referred to Reference #67
osmFISH	cyclic-ouroboros single molecule fluorescence in situ hybridization, and we referred to Reference #52
Spearman's ρ	Spearman's rank correlation coefficients
seqFISH+	evolution of sequential fluorescence in situ hybridization, and we referred to Reference #17
FOV	Field Of View
SpatialLIBD	A spatial transcriptomics dataset, we referred to Reference 54
DLPFC	DorsoLateral PreFrontal Cortex
k-means	A widely used clustering algorithm
Louvain	A widely used clustering algorithm, we referred to Reference #64
SC3	A widely used clustering algorithm, we referred to Reference #68
Giotto	A spatial omics data processing package, we referred to Reference #69
SEDR	An existing spatial domain identification algorithm, we referred to Reference #40
STAGATE	An existing spatial domain identification algorithm, we referred to Reference #41
SpatialPCA	An existing spatial domain identification algorithm, we referred to Reference #42
ARI	Adjusted Rand Index
MET	MicroEnvironment Trajectory
PAGA	One of the most widely used trajectory inference method, we referred to Reference #70
scMEP	A spatial proteomics technique, we referred to Reference #55
CRC	ColoRectal Carcinoma
FDR	False-Discovery Rate
DMA	Differential Microenvironment Analysis

EH plot	“entropy-of-ME-cluster (EMC)” versus “spatial heterogeneity (SHN)” plot, and we also referred to Methods for detailed explanation.
EMC	entropy-of-ME-cluster, and we also referred to Methods for detailed explanation.
NPC	Non-hepatic Parenchymal Cells
TILs	Tumor-infiltrating lymphocytes
EASI-FISH	3D spatial transcriptomics, we also referred to Reference #53
3D IMC	3D spatial proteomics, we also referred to Reference #91
WB ratio	Within-cluster-Between-cluster ratio
EMD	Earth mover’s distance, and we also referred to Reference #96
CGMGD	Connectivity Guided Minimum Graph Distance, and we also referred to “Connectivity guided minimum graph distance (CGMGD)” in Methods section for detail explanation.
PGEV	Pairwise Gene Expression Variation, where we also explained how to compute it.
NCME	Normalized count of MECNs, where we also explained how to compute it.
t-SNE	A famous manifold learning algorithm, and we referred to Reference #100
DPT	One of the most widely used trajectory inference algorithm, and we referred to Reference #104
BCM	Binary Connectivity Matrix
UDM	UMAP Distance Matrix
CGG	Connectivity Guided cluster Graph
ForceAtlas	A manifold learning algorithm, and we referred to Reference #105
Splatter	A single cell simulation tool, and we referred to Reference #106
MELD	An algorithm to compute the relative likelihood of observing samples in different conditions. We referred to Reference #46
BH	Benjamini–Hochberg

Reviewer #4:

Comment 1: The authors propose a method called SOTIP to perform three main tasks: 1) spatial heterogeneity quantification, 2) spatial domain identification, and 3) differential microenvironment analysis.

Those tasks are tackled by constructing a MECN graph that uses the gene expression and cell-type information of the neighbourhood. The authors demonstrated their method via both simulation and real data. In general, this is a well-written paper.

Author Response: We thank the reviewer for the positive comments.

Comment 2: This method is convincing, although I have a few comments.

Author Response: We thank the reviewer for the positive comments. The following criticisms are very important to improve our manuscript, and we will respond them one by one.

Comment 3: In simulation, the tissue structures are fairly simple. Please explain/interpret the SHN values in figure 2b. Are they estimated using the same algorithm? If so, how to interpret the large range of SHN estimated by SOTIP? If not, how to compare across methods?

Author Response: We thank the reviewer for pointing this out.

(1) Are they estimated using the same algorithm?

No. As noted in the Fig. 2b and its figure legend, the three sub-figures of Fig. 2b are estimated using three different algorithms, namely NUCC, IGD, and SOTIP, respectively.

(2) How to compare across methods?

Figure 2b is the performance comparison of NUCC (left), IGD (middle) and SOTIP (right) of SHN quantification on simulation data 1.

As shown in the PCA of figure 2a, C1 and C2 are close in gene expression, far different from C3. It makes the heterogeneity in C1-C2 boundary higher than that in C2-C3 boundary. So the true signal in this simulation is that:

- 1) There exist two strings of high spatial heterogeneity, one is at C1-C2 boundary, and the other one is at C2-C3 boundary.
- 2) The spatial heterogeneity of C2-C3 boundary should be higher than that of C1-C2.

The goal of SHN module of SOTIP is not only to identify the two boundaries, i.e., C1-C2 boundary as well as C2-C3 boundary, but to distinguish the heterogeneity difference between them, which may be linked to biological function as we showed in the cases of Fig. 3a-f and Fig. 3g-i.

For the goal to distinguish the level of heterogeneity between boundaries, we can see in Fig. 2b that, only SOTIP achieved this. In the top panel of Fig. 2b, SOTIP showed a weak peak around C1-C2 boundary and a strong peak around C2-C3 boundary, representing the low heterogeneity of C1-C2 boundary and high heterogeneity of C2-C3 boundary. NUCC and IGD failed to tell the difference between C1-C2 and C2-C3 in spatial heterogeneity by giving two peaks with approximately equal SHN values.

Comment 4: Based on simulation 1 (used in Figure 2a), c1 and c2 are indeed two clusters and there does exist spatial heterogeneity, although it is a weak one. NUCC and IGD can clearly identify it. However, the SHN of this boundary estimated from SOTIP is ~200, much smaller than the one for C2 and C3. Is it possible that the statistical power of SOTIP is smaller than another two methods? Similar to Figure 2D, the UMAP of CCNR can confidently split the C1-C5 as five clusters, however, the UMAP of SOTIP cannot, if we don't have prior cluster information. Authors argue this is an advantage that SOTIP preserves the continuous trend in the PC plot, which I'm afraid I have to disagree. In my option, this is a lack of power to cluster confidently.

Author Response: The reviewer raised an excellent point.

(1) Question in Figure 2b:

As discussed in the former comment, the goal of simulation 1 is two folds:

- (I) identify two peaks around C1-C2 and C2-C3.
- (II) The identified C2-C3 peak should be higher than C1-C2.

SOTIP-SHN could successfully achieve the two goals. Although NUCC and IGD could identify two peaks (the first goal), they could not tell any differences between them, since they give two peaks with approximately equal SHN values at C1-C2 and C2-C3. For statistical power, the two peaks of SOTIP (~200 and ~600) are both significantly larger than background (~0).

(2) Question in Figure 2d:

First of all, to make the simulation data 2 clearer, we revise Fig. 2c in the following.

In the revised figure, we added the cell type proportions in each of the 5 tissue bands (right panel). The revised figure together with manuscript text

As you can see from the revised Figure 2c, the five cell types C1-C5 are distributed in specific proportions in the five tissue regions of R1-R5. Also because C1-C5 formed a continuous trajectory in gene expression manifold (Fig. 2c left panel), we reason that the

tissue spatial trajectory of simulation 2 forms a continuous spatial trajectory from R1 to R5.

The goal of simulation 2 in Fig. 2c-d is to identify an ordered trajectory along R1 to R5 (Fig. 2c right panel), rather than obtain discrete clusters. So we used three widely used embedding algorithms (i.e., Diffusion map, umap, and PHATE) to embed the graph constructed by CCNR and SOTIP respectively. To make the comparison clearer, we revised Fig. 2d to add some arrows and annotations in the figure, to better show the identified order of trajectory.

Revised Fig. 2d

This tissue simulation data 2 is also an approximate simulation for the continuous and ordered tissue organization of brain cortex. Ref² reported that the cell type form a trajectory pattern in gene expression manifold, and each cell type also exhibits specific abundance along different cortical layer region, as shown in the following figure from Ref² Figure 3.

Comment 5: For simulation 3 (Figure 2e and 2f), samples 1 and 2 are a clean contrast. C1|C2 in sample 1 is unique compared to sample 2, and C3|C1 does not exist in sample 1. In reality, there could exist C3|C1 in sample 1 as well. For example, there could be a subtle difference in cell spatial distribution in the different stages of disease progression. But this subtle difference could affect the treatment outcome. In this case, the contrast between samples is not as straightforward as in figure 2e, increasing the noise in identifying spatial heterogeneity. What is the performance of SOTIP in such a case?

Author Response: This is an excellent point. In original simulation 3, C1|C2 is strictly unique in sample 1, and C3|C1 is strictly unique in sample 2. However, in real biological case, where C3|C1 exists in sample 1 as well, the output of this analysis would depend on the relative number of C3|C1 microenvironment in the two samples:

- If the number of C3|C1 in sample 1 > the number of C3|C1 in sample 2, which means C3|C1 microenvironment tends to appear more in sample 1. SOTIP will highlight C3|C1 in sample 1.
- If the number of C3|C1 in sample 1 < the number of C3|C1 in sample 2, which means C3|C1 microenvironment tends to appear more in sample 2. SOTIP will highlight C3|C1 in sample 2.
- If the number of C3|C1 in sample 1 = the number of C3|C1 in sample 2, which means C3|C1 do not have preference between sample 1 and sample 2. SOTIP will not highlight C3|C1 in neither sample.

Comment 6: Still, for simulation 3, there are equal distances among C1-C3. What is the effect if two clusters are close in PC space (e.g. C1 and C2 Figure 2a), compared to the third cluster (C3 in Figure 2a), but those clusters have different spatial distributions in different samples (eg the distribution of figure 2e).

Author Response: We thank the reviewer to point this out. Following the reviewer’s suggestions, we have added simulation 5 (Supplementary Figure 11). In simulation 5, C1 and C2 are close in PC space compared to C3 (Supplementary Figure 11a), and the sample 1 and sample 2 are formed by two different spatial distribution of those cells (Supplementary Figure 11b). We firstly performed SOTIP-SHN on sample 1 and sample 2, the results (Supplementary Figure 11c) showed that sample 1 showed high SHN values around C2-C3 boundary, and sample 2 showed high SHN values around both C2-C3 and C3-C1 boundaries. These observations are within expectation since C3 are distal from C1 and C2 in gene expression space (Supplementary Figure 11a).

We further tried to identify sample-specific microenvironments as done in Simulation 3 (Fig. 2f), the results (Supplementary Figure 11d) showed that SOTIP successfully highlighted specific boundaries associated with the two samples, i.e., C1-C2 boundary is highlighted in sample 1, and C3-C1 boundary is highlighted in sample 2. This observation indicated that the differences in gene expression distances do not influence the detection ability of SOTIP.

We provide codes for the additional simulation in “simulation_DMA_different_manifold_distance.ipynb” in our github repo.

Supplementary Figure 11

Supplementary Figure 11 (added in 1st round revision). Simulation 5.

This simulation is similar with Fig. 2e-f, in which the aim is to detect sample specific boundaries. The difference is that, in simulation 3 (Fig. 2e-f), the three cell clusters share the same distance in gene expression manifold, while in simulation 5 (this figure), the C1-C2 are closer, while C3 is further from C1 and C2 (a).

(b): Sample 1 (upper panel) and sample 2 (lower panel) are formed by two different spatial distribution of cells in (a).

(c): The SHN computed by SOTIP in sample 1(upper panel) and sample 2 (lower panel).

(d): Sample-specific microenvironments (left panel for sample 1, right panel for sample 2) are highlighted by MEG constructed with SOTIP. Each panel consists of two parts. Take the left panel as an example, the bottom part shows the same set of cells as in (b), colored according to the relative likelihood of observing each microenvironment in sample 1, and the top part shows the value of relative likelihood of each MECN as a function of horizontal coordinate.

Comment 7: In all simulations, the number of cells per cluster is almost identical. How the size of a cluster affects the detection of SHN?

In my opinion, comprehensive simulation can provide a good sense on the power and type I error rate of the method.

Author Response: We really appreciated for this excellent point. Following the reviewer's suggestions, we have added simulation 6 (Supplementary Figure 12). In original manuscript Fig. 2a-b, the sizes of three clusters are almost identical. To show the performance of SOTIP-SHN with different cluster size, we have tested 6 different combinations of cluster sizes in Supplementary Figure 12, in which each row shows one different cluster size ratio. The SHN plot in Supplementary Figure 12 (column 4) shows that, all SOTIP-SHN detected two peaks around C1-C2 and C2-C3 boundaries, and C2-C3 peak is stronger than C1-C2. This indicates that SOTIP-SHN is not affected by the size of clusters.

We provide codes for the additional simulation in "simulation_heter_boundary_different_cluster_size.ipynb" in our github repo.

Supplementary Figure 12

Supplementary Figure 12 (added in 1st round revision). Simulation 6.

This simulation shows the robustness of SOTIP-SHN module given different cluster size. There are 6 rows in this figure, and each row shows the cluster size ratio (column 1), PCA plot (column 2), spatial distribution of cells (column 3), and spatial plot colored by SHN values computed by SOTIP (column 4).

Comment 8: Figure 1a, please state what are those red squared boxes standing for?

Author Response: We are sorry for causing this confusion. In the original manuscript, the red box should stand for one spot. Thanks to this comment, we realized these red boxes might be mixed with those red boxes in Fig. 1b, so we remove those red boxes in Fig. 1a. In the revised Fig. 1a, the dash lines are directed linked to spots.

Comment 9: Figure 1e, from left to right, is the degree of spatial heterogeneity increasing or decreasing?

Author Response: We thank the reviewer for pointing this out. In our original manuscript, we intended to show a schematic representation of SHNs to show that different spatial mixture of cells have different SHNs. But we realized that the information presented in our original diagram caused confusion for readers. To this end, we removed those blue dots and simply used three different spatial cell organizations to represent different SHNs. The revised Fig. 1 is shown as follows.

Figure 1. Schematic overview.

Overview of SOTIP. SOTIP takes 2D/3D spatial omics data (spatially resolved gene expression, protein or metabolic profiles) as input (a), then represents microenvironments (MEs) by incorporating both spatial (b) and molecular (c) information (second column) to build a MECN graph (d). Finally, multiple tasks can be performed based on the MECN graph, including spatial heterogeneity quantification (e), spatial domain identification (f), differential microenvironment analysis (g), and other downstream tasks (h).

(i): The supported spatial data.

Comment 10: In general, please add a more detailed description to the figure captions.

Author Response: We totally agree that the figure legends should be more detailed. In the revised manuscript, we added more annotations in figures (such as Fig. 1i, Fig. 2c,d,e,f, Fig 3), and also added more descriptions in figure legends, especially meaning of arrows, region annotations, etc. Our revised contents are highlighted in red in original manuscript and figure legends.

Comment 11: Line 989-990: “but from joining MECNs of two samples”: do you perform PCA, neighbors, umap, Leiden within each sample to get MECN and then combine those two MECN? Or do you combine expression data and spatial data of two samples, then perform PCA, neighbors, umap, Leiden to get a MECN for the combined data? If it is the former one, the same cluster (e.g. Cluster 1) in two samples may stand for different cell types. Do you need to perform cell-type annotation beforehand? If it is the latter, how does the batch effect affect the MEG construction? It was not clear to me how you combined two samples.

Author Response: We are sincerely appreciated that the reviewer provides such an in-depth and practical question. We are pleased to answer this.

DME is applied on two biological cases, i.e., healthy liver vs fibrotic liver in Figure 6, and two subtypes of TNBCs in Figure 7. In both datasets, the cell type annotations were provided by original authors (as seen in Fig. 6c and Fig. 7b). To best avoid batch effect, our practice was the former one that you mentioned, that is because we believe manual annotation by markers is generally more reliable than computationally removing the batch effect. We also followed the original paper’s data processing parameters (e.g., generating neighbors, embedding, and clustering parameters) to make the cell type annotation agree well with embedding plot. So, in real applications, if both compared samples are provided with reliable cell type annotation, we would recommend firstly performing PCA, neighbors, umap, leiden within samples, followed by combining two MECNs. Otherwise when cell type annotations were not provided, we would recommend first manually annotate cell types (of course this would cause more human labours). One can alternatively follow the latter practice that you mentioned to firstly run batch effect algorithms followed by playing with integrated datasets, but one should double check the integrated dataset embedding to make sure the batch effect algorithm did disentangle the biological and noise signals.

Comment 12: Figure 6e-i demonstrate the DM in the fibrotic liver, compared to the healthy liver. Is there any DM in the healthy liver? Are they make biological sense?

Author Response: We thank the reviewer to point this out. We did not find those MEs which would tend to occur only in healthy liver. The interpretation behind this observation might be that fibrotic livers also contain areas of healthy liver tissue. Note that this is the DME result on the two specific samples, more commons and differences between healthy livers and fibrotic livers might be find if more comprehensive single cell spatial data are available in the future.

Comment 13: Too many acronyms and some are quite similar. I suggest having a table/ box listing all the acronyms. This will make the manuscript easier to read.

Author Response: This is a great idea. We added such a table in Supplementary Table 9.

Supplementary Table 9. Terms (This table is added in 1st revision)

Term	Explanation
ISS	In Situ Sequencing
smFISH	single-molecule Fluorescence In Situ Hybridization
MERFISH	Multiplexed Error-Robust Fluorescence In Situ Hybridization
seqFISH	Sequential Fluorescence In Situ Hybridization
ST	Spatial Transcriptomics
HDST	High-Definition Spatial Transcriptomics
4i	iterative indirect immunofluorescence imaging
CODEX	CO-DEtection by indeXing
MIBI-TOF	Multiplexed Ion Beam Imaging by Time-Of-Flight
IMC	Imaging Mass Cytometry
AFADESI-MSI	AirFlow-Assisted Desorption ElectroSpray Ionization Mass Spectrometry Imaging
SEAM	Spatial single nuCLEAR Metabolomics
ME	MicroEnvironment
KNN graph	We referred to reference #32
CCNR	Classical Cell Neighborhood Representation, and we referred to Supplementary Table 1 and reference #29~31
NUCC	Number of Unique Cell Clusters, and we referred to Supplementary Table 1 and reference #16,33,35
BayesSpace	It is the name of an existing algorithm, and we referred to Supplementary Table 1 and reference #37
HMMRF	Hidden-Markov Random Field
SpaGCN	It is the name of an existing algorithm, and we referred to Supplementary Table 1 and reference #38
GCN	Graph Convolutional Network
StLearn	It is the name of an existing algorithm, and we referred to Supplementary Table 1 and reference #39
scRNA-seq	single-cell RNA sequencing
SHN	Spatial Heterogeneity
SDM	Spatial DoMain
DME	Differential MicroEnvironment
MECN	Molecular-Expression-aware Cellular Neighborhood
MEG	MECN Graph
TNBC	Triple Negative Breast Cancer
MKT	Highly specific MECNs identified by DME
SIMS	Secondary Ion Mass Spectrometry
IGD	An algorithm for quantifying SHN, we referred to Supplementary Table 1, where it is further linked to Methods section for detailed explanation.
UMAP	A widely used manifold learning method, we referred to Reference #57
PHATE	A widely used manifold learning method, we referred to Reference #58
DNA	We did not define it since it has been widely known as Deoxyribonucleic acid.
Squidpy	A python package for processing spatial omics data, we referred to Reference #61

HeLa	We did not define it since it has been widely known as a class of cell line.
AUC	Area Under Curve
PCNA	Proliferating Cell Nuclear Antigen
RPS6	Ribosomal Protein S6
ER	Endoplasmic Reticulum
BRAFV600E	A melanoma zebrafish model, we referred to Reference #63
Leiden	An existing clustering algorithm, we referred to Reference #64
Pearson's r	Pearson correlation coefficient
BICCN	BRAIN Initiative Cell Census Network, and we referred to Reference #67
osmFISH	cyclic-ouroboros single molecule fluorescence in situ hybridization, and we referred to Reference #52
Spearman's ρ	Spearman's rank correlation coefficients
seqFISH+	evolution of sequential fluorescence in situ hybridization, and we referred to Reference #17
FOV	Field Of View
SpatialLIBD	A spatial transcriptomics dataset, we referred to Reference 54
DLPFC	DorsoLateral PreFrontal Cortex
k-means	A widely used clustering algorithm
Louvain	A widely used clustering algorithm, we referred to Reference #64
SC3	A widely used clustering algorithm, we referred to Reference #68
Giotto	A spatial omics data processing package, we referred to Reference #69
SEDR	An existing spatial domain identification algorithm, we referred to Reference #40
STAGATE	An existing spatial domain identification algorithm, we referred to Reference #41
SpatialPCA	An existing spatial domain identification algorithm, we referred to Reference #42
ARI	Adjusted Rand Index
MET	MicroEnvironment Trajectory
PAGA	One of the most widely used trajectory inference method, we referred to Reference #70
scMEP	A spatial proteomics technique, we referred to Reference #55
CRC	ColoRectal Carcinoma
FDR	False-Discovery Rate
DMA	Differential Microenvironment Analysis
EH plot	"entropy-of-ME-cluster (EMC)" versus "spatial heterogeneity (SHN)" plot, and we also referred to Methods for detailed explanation.
EMC	entropy-of-ME-cluster, and we also referred to Methods for detailed explanation.
NPC	Non-hepatic Parenchymal Cells
TILs	Tumor-infiltrating lymphocytes
EASI-FISH	3D spatial transcriptomics, we also referred to Reference #53
3D IMC	3D spatial proteomics, we also referred to Reference #91
WB ratio	Within-cluster-Between-cluster ratio
EMD	Earth mover's distance, and we also referred to Reference #96
CGMGD	Connectivity Guided Minimum Graph Distance, and we also referred to "Connectivity guided minimum graph distance (CGMGD)" in Methods section for detail explanation.
PGEV	Pairwise Gene Expression Variation, where we also explained how to compute it.
NCME	Normalized count of MECNs, where we also explained how to compute it.
t-SNE	A famous manifold learning algorithm, and we referred to Reference #100
DPT	One of the most widely used trajectory inference algorithm, and we referred to Reference #104

BCM	Binary Connectivity Matrix
UDM	UMAP Distance Matrix
CGG	Connectivity Guided cluster Graph
ForceAtlas	A manifold learning algorithm, and we referred to Reference #105
Splatter	A single cell simulation tool, and we referred to Reference #106
MELD	An algorithm to compute the relative likelihood of observing samples in different conditions. We referred to Reference #46
BH	Benjamini–Hochberg

Comment 14: On Lines 145-174 (figure 1a): the MECN is a vector of cell type frequency within a cellular neighbourhood. Thus, the definition of the microenvironment is the same as before.

Author Response: We thank the reviewer to point this out. The reason why we created MECN as the mathematic name of cellular neighborhood is to avoid the mixture with the “microenvironment” in biologist’s view. The innovation behind MECN is the histogram representation and using EMD distance between MECNs.

Comment 15: Figure 5: what is the meaning of white errors?

Author Response: In Figure 5d,e,g,h, the white arrows all point to tumor regions. We have added this in the revised legend.

Comment 16: In Figure 7, the choice of interesting EMCN clusters is “bottom-right” of the EH plot. Is there a way to quantify this ‘bottom-right’?

Author Response: We thank the reviewer to point this out. MECN clusters with low “E” (high sample specificity) and high “H” (high spatial heterogeneity) tend to be interesting, because low “E” means high sample-specificity, which means the targeted microenvironments tend to occur in one sample rather than another, high “H” means high spatial heterogeneity, which means the targeted microenvironments are not formed by one single cell type (we are interested in DMs which are characterized by interplays of multiple cell types). Low “E” means small value in the EMC (vertical) axis of EH plot, and high “H” means big value in the SHN (horizontal) axis of EH plot, so we noted low “E” and high “H” as “bottom-right” of EH plot in the original manuscript. In terms of how to quantify this “bottom-right”, in current version of SOTIP, the DME analysis is via visualization of the EH plot, so we did not provide rigid value to define the threshold. However, we could provide a recommendation for that, e.g., $EMC \leq 0.25$ and $SHN > 0$.

Comment 17: Line 352-353: how many cells within a radius of 100 um? Does “the same manner as NUCC” mean that NUCC uses cells within 100um to define a ME, but SOTIP uses 10 nearest neighbouring cells?

Author Response: Thank you for pointing this out. “The same manner as NUCC” means ref² uses the number of cell clusters present in neighborhood, just as NUCC. In practice, NUCC and SOTIP both uses 10-NN cells in the comparison. To make the text more clear, we have changed “They defined the neighborhood complexity of a cell as the number of different cell clusters present within a radius of 100 μm from the given cell (the same manner as NUCC)” to “They counted the number of different cell clusters within cell neighborhood to define the spatial heterogeneity (like NUCC does)”.

Comment 18: Typo on line 426: the mean of SpaGCN is 0.433, instead of 4.33

Author Response: Corrected, thanks!

Comment 19: Supplementary figure 4: a-d, the dots in the scatter plots have different sizes. Please add a legend to explain the meaning of it.

Author Response: Thanks for point this out. Different dot sizes mean different cell sizes. The cell size information is provided by original data. We have added the information in the figure legends.

Comment 20: Supplementary figure 9: please explain ‘WB’.

Author Response: Explanation added, thanks! **WB ratio (the Within-cluster-Between-cluster ratio)**.

Comment 21: Provide a guide for the time, number of CPUs, number of threads etc for each of the analyses. Thus, users can easily set up the required computational resource in their analyses.

Author Response: This is a great suggestion, we have added a “Computational resource” section in Supplementary Notes.

Across the manuscript, all the results are run on a machine with following computational resources.

Item	Information
Operation System	CentOS Linux release 7.9
Number of CPUs	16
Number of GPUs	0
Memory Size	31.1G

This means that the required computational resources are recommended to successfully run our analyses. Machines with lower computational resources may also be enough depending on analysis. We listed the expected time of different analyses as follows:

Analysis	Number of spots/cells	Recommend multiprocessing	Expected time
SHN	10^2	No	<1min
SHN	10^3	No	<1min
SHN	10^4	No	1min~5min
SDM	10^2	Yes	<1min
SDM	10^3	Yes	1min~5min
SDM	10^4	Yes	5min~10min
DME	10^2	Yes	<1min
DME	10^3	Yes	1min~5min
DME	10^4	Yes	5min~10min

Comment 22: Line 712: Keratin+ tumor means keratinocyte and tumour?

Author Response: Keratin+ tumor means tumor cells with positive expression of Keratin. We have added the description when this term is firstly presented.

Comment 23: Lines 1101, please give sufficient detail about MELD, such that I don't have read the original paper. The same to PAGA.

Author Response: This is a great point. We have added the description of MELD in "MELD analysis" in Methods section, as follows.

MELD analysis

MELD⁴ is an algorithm for estimating the relative likelihood of observing each cell state between different conditions (e.g., different disease state, before/after drug treatment or other experimental perturbations). In original MELD publication, the input of MELD is (1) the single cell graph on gene expression space, and (2) the condition label for each single cell. The output of MELD is the relative likelihood of observing each cell in each condition.

In our application (e.g., Figure 2f), we instead input the microenvironment graph, and the condition label for each microenvironment. In this way, by combining SOTIP's mathematical definition of microenvironment and MELD, we can estimate the relative likelihood of observing each microenvironment between different conditions.

We have also added the description of PAGA in "PAGA analysis" in Methods section, as follows.

PAGA analysis.

PAGA is an algorithm for trajectory inference through a topology preserving map of single cells. In the original PAGA publication, the input of PAGA is (1) the single cell graph on gene expression space, (2) a clustering assignment for the single cell data. The output of PAGA is graph, in which each node is a cluster, and the edge between two nodes is the strength of connectivity (the connectivity could also be interpreted as similarity) between two clusters. In our application (e.g., Supplementary Figure 3), we instead input the microenvironment graph, and the clustering label. So that we can get the topological map of the targeted tissue.

Comment 24: Lines 1122-1123: I'm afraid I have to disagree that using exactly the same parameter for different models is a fair comparison.

Author Response: This is a good point. We explain how we compared methods as follows.

In SHN performance comparison, NUCC, IGD and SOTIP have the same set of parameters, i.e., KNN when defining microenvironments, and clustering resolution in the initial step, the only difference between them is how to quantify the heterogeneity after the definition of MECN. This is the reason why we used the same set of parameters for NUCC, IGD and SOTIP.

In SDM performance comparison, SOTIP and other control methods (e.g., stLearn, BayesSpace, SpaGCN ,etc.) have completely different parameters. So, we reported their performances (Figure 4d) as they claimed in their original paper, this can represent their performance with the optimal set of parameters tuned by original authors, and thus reduce potential bias.

Comment 25: Line 903-910 is not clear to me how to define a MECN. I understand that a MECN is measured per cell, the cell type frequency estimated based on its KNN (10). Or in MEG, each node is a cell, the feature of this node is MECN. The MECN is the cell type frequency estimated based on the cell's KNN.

Author Response: We think the reviewer pointed out an excellent point. To make things clearer, we explain the key ideas as follows.

For a data with N cells and P features (e.g., gene expression vector with length P), traditional (non-spatial) clustering methods will directly construct a graph for the N cells, by computing distances on the P feature vectors. Then Leiden algorithm is performed on this graph to get the clustering result. Such traditional clustering methods only take into account the gene expression profile of each cell, without considering the neighborhood information of each cell.

On the contrary, SOTIP considers both gene expression profiles of cells, and the neighborhood information of cells. The main idea of SOTIP is to represent the neighborhood of each cell (neighborhood indexed by cell- i is called MECN- i) as a histogram of cell type frequency, then SOTIP uses the earth mover's distance (EMD, a distance metrics for histograms) based on optimal transport to construct a graph (termed MEG) between all MECNs, and finally a clustering procedure is performed on MEG to get the clustering result (i.e., spatial domain identification result). **Here, we must re-emphasis that MEG is the graph of all MECNs, each node of MEG is a MECN, and edge between nodes is the distance between two MECNs, which is the EMD distance between the two histograms characterizing the two MECNs. This echoes the definition of MEG when it is firstly used in original manuscript line 149.**

The key design of SOTIP is the construction of the MECN graph (MEG) using EMD. Using this way, when measuring the distance between two MECNs (e.g., MECN- i and MECN- j , indexed by cell- i and cell- j), SOTIP considers how to design an optimal transportation plan on the gene expression manifold to move cells (characterized by cell type) within MECN- i to match cells within MECN- j , so that the histogram characterizing MECN- i after the transportation is exactly the same as the histogram characterizing MECN- j . Another key design is that when designing the optimal transportation plan, the cells must be moved along the gene expression manifold. The optimal transportation cost is used as the distance between MECN- i and MECN- j . One can imagine that the transportation cost is determined by two factors: (1) the number of cells to be moved, and (2) the cost of moving cells between two positions along the gene expression manifold. The first factor has already been encoded in the histograms associated with MECNs, and the second factor (named ground distance in optimal transport theorem) is the geodesic distance between cell types along the gene expression manifold, the CGMGD in our manuscript. This process is formulated in line 929~939 of original manuscript.

Comment 26: Table Supp 4, 5 where is a comparison with RESEPT, FICT, ect?

Author Response: We thank the reviewer to point this out. We discuss the comparison of SOTIP, RESEPT and FICT in 3 aspects:

(1) Function.

RESEPT and FICT are both SDM methods, which is one part of module of SOTIP. While SOTIP is a versatile method with multiple functions, including SHN, SDM, and DME.

(2) Method principle.

SOTIP and FICT are both unsupervised methods, which do not need external labeled data. While RESEPT is a supervised deep learning method for spatial domain identification. According to RESEPT's manuscript (<https://doi.org/10.1101/2021.07.08.451210>) line 33~38, RESEPT firstly learns a three-dimensional embedding using a graph autoencoder from the spatial transcriptomics data. The embedding is then visualized by mapping as color channels in an RGB image and segmented with a supervised convolutional neural network model.

(3) Performance

As discussed in (2), RESEPT is a supervised SDM method, and could not be compared with unsupervised SDM methods as we compared in Fig. 4d. The reason why FICT is not compared in Fig. 4d is that FICT is a method for multiplexed FISH data (as seen in FICT's abstract). These also explained why recent SDM methods^{15,16,18,22} did not compared with either FICT or RESEPT.

Comment 27: The program on GitHub can be run successfully, and demo data can be downloaded too.

Author Response: Thanks for trying our program.

References

- 1 Chen, H. *et al.* Dissecting Mammalian Spermatogenesis Using Spatial Transcriptomics. *bioRxiv*, doi:10.1101/2020.10.17.343335 (2020).
- 2 Zhang, M. *et al.* Spatially resolved cell atlas of the mouse primary motor cortex by MERFISH. *Nature* **598**, 137-143, doi:10.1038/s41586-021-03705-x (2021).
- 3 Moffitt, J. R. *et al.* Molecular, spatial, and functional single-cell profiling of the hypothalamic preoptic region. *Science* **362**, doi:10.1126/science.aau5324 (2018).
- 4 Burkhardt, D. B. *et al.* Quantifying the effect of experimental perturbations at single-cell resolution. *Nat Biotechnol*, doi:10.1038/s41587-020-00803-5 (2021).
- 5 Dann, E., Henderson, N. C., Teichmann, S. A., Morgan, M. D. & Marioni, J. C. Differential abundance testing on single-cell data using k-nearest neighbor graphs. *Nat Biotechnol*, doi:10.1038/s41587-021-01033-z (2021).
- 6 Reshef, Y. A. *et al.* Co-varying neighborhood analysis identifies cell populations associated with phenotypes of interest from single-cell transcriptomics. *Nat Biotechnol*, doi:10.1038/s41587-021-01066-4 (2021).
- 7 Eng, C. L. *et al.* Transcriptome-scale super-resolved imaging in tissues by RNA seqFISH. *Nature* **568**, 235-239, doi:10.1038/s41586-019-1049-y (2019).
- 8 Stickels, R. R. *et al.* Highly sensitive spatial transcriptomics at near-cellular resolution with Slide-seqV2. *Nat Biotechnol*, doi:10.1038/s41587-020-0739-1 (2020).
- 9 Rodriques, S. G. *et al.* Slide-seq: A scalable technology for measuring genome-wide expression at high spatial resolution. *Science* **363**, 1463+, doi:10.1126/science.aaw1219 (2019).
- 10 Keren, L. *et al.* A Structured Tumor-Immune Microenvironment in Triple Negative Breast Cancer Revealed by Multiplexed Ion Beam Imaging. *Cell* **174**, 1373+, doi:10.1016/j.cell.2018.08.039 (2018).
- 11 Hartmann, F. J. *et al.* Single-cell metabolic profiling of human cytotoxic T cells. *Nat Biotechnol*, doi:10.1038/s41587-020-0651-8 (2020).
- 12 Goltsev, Y. *et al.* Deep Profiling of Mouse Splenic Architecture with CODEX Multiplexed Imaging. *Cell* **174**, 968-981 e915, doi:10.1016/j.cell.2018.07.010 (2018).
- 13 Palla, G. *et al.* Squidpy: a scalable framework for spatial omics analysis. *Nat. Methods*, doi:10.1038/s41592-021-01358-2 (2022).
- 14 Zhao, E. *et al.* Spatial transcriptomics at subspot resolution with BayesSpace. *Nat Biotechnol*, doi:10.1038/s41587-021-00935-2 (2021).
- 15 Yang, Y. *et al.* SC-MEB: spatial clustering with hidden Markov random field using empirical Bayes. *Brief Bioinform* **23**, doi:10.1093/bib/bbab466 (2022).
- 16 Hu, J. *et al.* SpaGCN: Integrating gene expression, spatial location and histology to identify spatial domains and spatially variable genes by graph convolutional network. *Nat Methods* **18**, 1342-1351, doi:10.1038/s41592-021-01255-8 (2021).
- 17 Li, J., Chen, S., Pan, X., Yuan, Y. & Shen, H.-b. CCST: Cell clustering for spatial transcriptomics data with graph neural network. doi:10.21203/rs.3.rs-990495/v1 (2021).
- 18 Dong, K. & Zhang, S. Deciphering spatial domains from spatially resolved transcriptomics with an adaptive graph attention auto-encoder. *Nat Commun* **13**, 1739, doi:10.1038/s41467-022-29439-6 (2022).
- 19 Tenenbaum, J. B., De Silva, V. & Langford, J. C. A global geometric framework for nonlinear dimensionality reduction. *Science* **290**, 2319-2323 (2000).

- 20 Chen, W. S. *et al.* Uncovering axes of variation among single-cell cancer specimens. *Nat. Methods*, doi:10.1038/s41592-019-0689-z (2020).
- 21 Zappia, L., Phipson, B. & Oshlack, A. Splatter: simulation of single-cell RNA sequencing data. *Genome Biol* **18**, 174, doi:10.1186/s13059-017-1305-0 (2017).
- 22 Li, J., Chen, S., Pan, X., Yuan, Y. & Shen, H.-B. Cell clustering for spatial transcriptomics data with graph neural networks. *Nature Computational Science* **2**, 399-408, doi:10.1038/s43588-022-00266-5 (2022).

Reviewer comments:**Reviewer #2 (Remarks to the Author: Overall significance):**

We thank the authors for the additional supplementary figures, tables, and added clarifications to the manuscript. The authors should also be commended for providing their reproducible code. The authors have addressed most of our scientific concerns and the manuscript is significantly strengthened. However, the majority of the text in the main manuscript remains unchanged from the first submission and is highly technical. As requested, the authors have equipped the reader to be able to interrupt the results with the extra tools (terms tables, additional supplementary notes, etc). However, this puts a large onus on the reader to digest a significant amount of specialized information in tables (for example over 80 terms) that perhaps could have been more distilled in the main manuscript. While the work is suitable for publication, I have concerns that the accessibility of the manuscript will be challenging for the readership of the targeted journals as it remains written for a very specialized audience. I was hoping to see some of the simplifications found in the new supplementary figures incorporated into the main text to improve the readability of the work.

Reviewer #3 (Remarks to the Author: Overall significance):

I am satisfied with the revision. Most of my questions are addressed. Thank you!

Reviewer #3 (Remarks to the Author: Strength of the claims):

A few minor comments:

With the response to my question #6 I understand that the spatial domain analysis is in fact a clustering analysis of the cells' graph representations, without using any distance information between graph_i and graph_j. This is helpful clarification. Defined in this way, maybe it is better to call it spatial domain type analysis rather than spatial domain analysis, because the output is simply the K types of MECNs, and these K types may either form contiguous domains or scatter into a large number of fragmented domains. I can see in Figure 4c and 4h that the same domain types may indeed be scattered.

It is also helpful to see the clarification on the use of "CGMGD". This complex acronym can be simply explained as the distance in the transcriptome state space. The "geodesic distance along the gene expression manifold" is not wrong, but may lose some readers.

In Figure 5e-f, it is a bit puzzling that the performance of SpaGCN was not too bad in panel e, but appeared very poor in panel f. Please make sure SpaGCN has a proper tuning to provide a fair comparison.

Reviewer #4 (Remarks to the Author: Overall significance):

Thanks to the authors who have extensively addressed my and other reviewers' comments.

I have one question: for the simulation used in Figure 2 and Figure S11, the SHN max likelihood varied a lot, for example, ~600 in figure 2b and only 0.00032 in figure 2f. Could the authors help me to clarify the reasons? For figure 2f, could authors draw a threshold line for the SHN likelihood to indicate the boundary between significant and non-significant?

Reviewer #5 - This referee co-reviewed with Reviewer #2 and submitted the same report (Remarks to the Author: Overall significance):

We thank the authors for the additional supplementary figures, tables, and added clarifications to the

manuscript. The authors should also be commended for providing their reproducible code. The authors have addressed most of our scientific concerns and the manuscript is significantly strengthened. However, the majority of the text in the main manuscript remains unchanged from the first submission and is highly technical. As requested, the authors have equipped the reader to be able to interrupt the results with the extra tools (terms tables, additional supplementary notes, etc). However, this puts a large onus on the reader to digest a significant amount of specialized information in tables (for example over 80 terms) that perhaps could have been more distilled in the main manuscript. While the work is suitable for publication, I have concerns that the accessibility of the manuscript will be challenging for the readership of the targeted journals as it remains written for a very specialized audience. I was hoping to see some of the simplifications found in the new supplementary figures incorporated into the main text to improve the readability of the work.

Reviewer #6 - Expert in breast cancer genomics, tumour microenvironment and immune microenvironment, and spatial omics (Remarks to the Author: Overall significance):

Review for Yuan et al.

In this manuscript Yuan et al. present SOTIP, a novel computational framework for analyzing microenvironmental information in spatial OMICS datasets. The novelty of the approach is that it considers the low-dimensional manifold of cells when it uses earth movers' distance to define distances between microenvironments. Following the instructions of the editor, my review is limited to the method (figures 1&2) and the analysis of pathological samples (figures 5&7).

Major comments:

1. I found it very difficult to follow the description of the algorithm. The main text and figures only provide an extremely high-level overview of the approach, with all the details in the methods. Frankly, I was only able to understand what was going on when I looked at the rebuttal file in the response to reviewer #3 (pages 30-31). I believe that the paper would be extremely aided by:
 - a. Adding a section in the results describing the model rationale. This should be at a level that provides more details than what is currently in the introduction, but less detail than what is in the methods. I suggest something along the lines of the response to reviewer #3.
 - b. Figure 1 is not comprehensible at all. From the figure I thought that the single-cell manifold is used directly as input to the ME graph, but after reading the rebuttal, I understood that this is the input into cell type classification, and later the histograms of cell types are actually the input to the ME graph (I hope that I am correct). In any case, a better, more detailed illustration of the approach is crucial in order for this paper to have impact in the field.
2. In the section "SOTIP characterizes spatial and molecular tumor-immune organization" it is also unclear what was done. The authors merely state that they used SOTIP. I am missing some details. Was the SDM module used? If so, did the authors indicate that they expect to find two domains? Did the authors use the cell types as defined in the original papers? How does the EMD graph between cell types look like for these proteomic datasets? This last point is crucial because these proteomic datasets have much less genes profiled (on the order of dozens). As such, it is unclear how well the EMD graph will look like, as often a single protein can define a cell subset. It is very difficult to evaluate the approach when all these details are not disclosed.
3. I don't understand the plots in figure 5F and I. The legend states: "The power plots show the proportion of true positives (y axis) detected by different methods at a range of FDRs (x axis)". But I don't understand the underlying analysis. Is it for all proteins? Or only one of the proteins shown in the figure? If so, which one? How do the authors define true positives? Please clarify what exactly was done.

Minor points:

1. I could not find a place in the manuscript that referenced supplementary figure 6B

Reviewer #6 (Remarks to the Author: Impact):

I think that the method will impact thinking in the field

Responses to the reviewers' comments

Response to all reviewers:

We are very grateful to the reviewers for their positive and constructive comments on our manuscript. In this 2nd response letter, original comments of reviewers are highlighted in yellow, author's responses are in black, referred contents from original manuscript are in blue, changes in revised manuscript are in red.

Reviewer comments:

Reviewer #2 (Remarks to the Author: Overall significance):

We thank the authors for the additional supplementary figures, tables, and added clarifications to the manuscript. The authors should also be commended for providing their reproducible code. The authors have addressed most of our scientific concerns and the manuscript is significantly strengthened. However, the majority of the text in the main manuscript remains unchanged from the first submission and is highly technical. As requested, the authors have equipped the reader to be able to interrupt the results with the extra tools (terms tables, additional supplementary notes, etc). However, this puts a large onus on the reader to digest a significant amount of specialized information in tables (for example over 80 terms) that perhaps could have been more distilled in the main manuscript. While the work is suitable for publication, I have concerns that the accessibility of the manuscript will be challenging for the readership of the targeted journals as it remains written for a very specialized audience. I was hoping to see some of the simplifications found in the new supplementary figures incorporated into the main text to improve the readability of the work.

Author's response:

We are grateful for the reviewer's positive feedbacks. To improve the accessibility of the manuscript, we additionally revised in following aspects:

- 1, We updated Figure 1a-h so that more detailed information is shown (including an updated element "clustering", changes in the text "histogram of cell clusters", and "optimal transport").
- 2, Following the reviewer's suggestion, we incorporated a simplified version of Supplementary Figure 13~15 into Figure 1i, so that the reader can quickly get the main structure of this manuscript.
- 3, We added a new section "Model rational" in Results, to strengthen the readers' understanding on the model motivation and rational.

Reviewer #3 (Remarks to the Author: Overall significance):

I am satisfied with the revision. Most of my questions are addressed. Thank you!

Author's response:

We would like to thank you since our revision was mostly inspired by the reviewers' constructive suggestions.

Reviewer #3 (Remarks to the Author: Strength of the claims):

A few minor comments:

With the response to my question #6 I understand that the spatial domain analysis is in fact a clustering analysis of the cells' graph representations, without using any distance information between graph i and graph j . This is helpful clarification. Defined in this way, maybe it is better to call it spatial domain type analysis rather than spatial domain analysis, because the output is simply the K types of MECNs, and these K types may either form contiguous domains or scatter into a large number of fragmented domains. I can see in Figure 4c and 4h that the same domain types may indeed be scattered.

Author's response:

Thank you for pointing this out! We understand that "spatial domain type analysis" is more strict than "spatial domain analysis" since in some cases the same "domain" might be scattered. This nomenclature is also analogous with "cell type analysis" in scRNA-seq, since both their targets are to assign a categorical "type" to some objects. Our reason to name it with "spatial domain analysis" is to keep consensus with other related methods (for example SpaGCN¹ and STAGATE²), so that readers would not be confused when reading and comparing all these papers.

It is also helpful to see the clarification on the use of "CGMGD". This complex acronym can be simply explained as the distance in the transcriptome state space. The "geodesic distance along the gene expression manifold" is not wrong, but may lose some readers.

Author's response:

This is a very nice advice! We admit that "geodesic distance along the gene expression manifold" might be too technical for more generous readers, and we have reworded the term in the main text to "distance in the transcriptome state space" (Line 883, 1028, 1042).

In Figure 5e-f, it is a bit puzzling that the performance of SpaGCN was not too bad in panel e, but appeared very poor in panel f. Please make sure SpaGCN has a proper tuning to provide a fair comparison.

Author's response:

Good point! We have to clarify that it is expected that the ability to identify polarized proteins (Fig. 5f) might be poor based on a not-so-bad SDM result (Fig. 5e left).

The reason is that the way we identify polarized proteins is (and is currently a standard practice in other papers^{3,4}) to first define two populations of cells (cells close to the tissue boundary and cells far from the tissue boundary) based on physical distance, and then use a two-sample test to compare the two groups of cells to identify proteins with significant differences in abundance (as described in Methods section, Line 1203). This approach indicates that the determination of tissue boundaries is very sensitive to the identification of polarized proteins. Although the results of SpaGCN do not look so bad (Fig. 5e left), it mistakenly produced some "holes" within tumor tissues or immune tissues, which led to the wrong identification of tissue boundaries, and ultimately led to the poor identification of polarized proteins.

Reviewer #4 (Remarks to the Author: Overall significance):

Thanks to the authors who have extensively addressed my and other reviewers' comments.

Author's response:

Thanks the reviewer for his/her very professional comments, especially in helping us to improve the comprehensiveness of the simulation study and clarification on multiple terms (e.g., MECN definition).

I have one question: for the simulation used in Figure 2 and Figure S11, the SHN max likelihood varied a lot, for example, ~600 in figure 2b and only 0.00032 in figure 2f. Could the authors help me to clarify the reasons? For figure 2f, could authors draw a threshold line for the SHN likelihood to indicate the boundary between significant and non-significant?

Author's response:

The reviewer has an excellent point. Figure 2b and Figure 2f are two different tasks, and the quantifications in the plots are also different. In Figure 2b, the plot shows the SHN values (see the following figure), while in figure 2f the plot shows the relative likelihood of microenvironments between samples. The SHN values (Figure 2b) was generated by SOTIP's SHN module, and the relative likelihood values (Figure 2f) was generated by MELD analysis (as described in the 1st section of Results and in Methods Line 1090). So it is not surprising at the difference in their ranges. As to determine the "significant" boundaries, since the quantification of SHN is not based on statistical testing, so it might be not possible to generate a p-value for each MECN to determine the significance in current form of SOTIP. We will explore this possibility in the future to explore statistical solutions in spatial omics.

Reviewer #5 - This referee co-reviewed with Reviewer #2 and submitted the same report (Remarks to the Author: Overall significance):

We thank the authors for the additional supplementary figures, tables, and added clarifications to the manuscript. The authors should also be commended for providing their reproducible code. The authors have addressed most of our scientific concerns and the manuscript is significantly strengthened. However, the majority of the text in the main manuscript remains unchanged from the first submission and is highly technical. As requested, the authors have equipped the reader to be able to interrupt the results with the extra tools (terms tables, additional supplementary notes, etc). However, this puts a large onus on the reader to digest a significant amount of specialized information in tables (for example over 80 terms) that perhaps could have been more distilled in the main manuscript. While the work is suitable for publication, I have concerns that the accessibility of the manuscript will be challenging for the readership of the targeted journals as it remains written for a very specialized audience. I was hoping to see some of the simplifications found in the new supplementary figures incorporated into the main text to improve the readability of the work.

Author's response:

We are grateful for the reviewer's positive feedbacks. To improve the accessibility of the manuscript, we additionally revised in following aspects:

- 1, We updated Figure 1a-h so that more detailed information is shown (including an updated element "clustering", changes in the text "histogram of cell clusters", and "optimal transport").
- 2, Following the reviewer's suggestion, we incorporated a simplified version of Supplementary Figure 13~15 into Figure 1i, so that the reader can quickly get the main structure of this manuscript.
- 3, We added a new section "Model rational" in Results, to strengthen reader's understanding on the model's motivation and rational.

Reviewer #6 - Expert in breast cancer genomics, tumour microenvironment and immune microenvironment, and spatial omics (Remarks to the Author: Overall significance):

Review for Yuan et al.

In this manuscript Yuan et al. present SOTIP, a novel computational framework for analyzing microenvironmental information in spatial OMICS datasets. The novelty of the approach is that it considers the low-dimensional manifold of cells when it uses earth movers' distance to define distances between microenvironments. Following the instructions of the editor, my review is limited to the method (figures 1&2) and the analysis of pathological samples (figures 5&7).

Major comments:

1. I found it very difficult to follow the description of the algorithm. The main text and figures only provide an extremely high-level overview of the approach, with all the details in the methods. Frankly, I was only able to understand what was going on when I looked at the rebuttal file in the response to reviewer #3 (pages 30-31). I believe that the paper would be extremely aided by:

a. Adding a section in the results describing the model rational. This should be at a level that provides more details than what is currently in the introduction, but less detail than what is in the methods. I suggest something along the lines of the response to reviewer #3.

b. Figure 1 is not comprehensible at all. From the figure I thought that the single-cell manifold is used directly as input to the ME graph, but after reading the rebuttal, I understood that this is the input into cell type classification, and later the histograms of cell types are actually the input to the ME graph (I hope that I am correct). In any case, a better, more detailed illustration of the approach is crucial in order for this paper to have impact in the field.

Author's response:

We gratefully thank the reviewer to point out the readability issues of figures and texts. We tried to address these issues in the following aspects:

1, Add a new section in Results to describe the model rational.

Model rational

To smooth the reading, we describe SOTIP's rational in this section (see Methods for more technical description). In the following, we use "cells" to denote the measurement units in spatial omics data for ease of explanation, which can be freely replaced to "spots" in technologies like Visium or ST.

For any data with N cells and P features (e.g., gene expression vector with length P), traditional (non-spatial) clustering methods directly construct a graph for the N cells, by computing distances on the P feature vectors. Then Leiden algorithm is performed on this graph to get the clustering result. Such traditional clustering methods only take into account the gene expression profile of each cell, without considering the neighborhood information of each cell.

On the contrary, SOTIP considers both gene expression profiles of cells, and the neighborhood information of cells. The main idea of SOTIP is to represent the neighborhood of each cell (neighborhood indexed by cell- i is called MECN- i) as a histogram of cell types (following figure shows example of 3 MECNs characterized by histograms), then SOTIP uses the earth mover's distance (EMD, a distance metrics for histograms) based on an optimal transport principle to construct a graph (termed MEG) between all MECNs, and finally a clustering procedure is performed on MEG to get the clustering result (i.e., spatial domain identification result). Here, it is worth re-emphasizing that MEG is the graph of all MECNs, each node of MEG is a MECN, and edge between nodes is the distance between two MECNs, which is the EMD distance between the two histograms characterizing the two MECNs.

The key design of SOTIP is construction of the MECN graph (MEG) using EMD. In this way, when measuring the distance between two MECNs (e.g., MECN- i and MECN- j , indexed by cell- i and cell- j), SOTIP considers how to design an optimal transportation plan on the gene expression manifold to move cells (characterized by cell type) within MECN- i to match cells within MECN- j , so that the histogram characterizing MECN- i after the transportation is exactly the same as the histogram characterizing MECN- j . Another key design is that when designing the optimal transportation plan, the cells must be moved along the gene expression manifold. The optimal transportation cost is used as the distance between MECN- i and MECN- j . One can imagine that the transportation cost is determined by two factors: (1) the number of cells to be moved, and (2) the cost of moving cells between two positions along the gene expression manifold. The first factor has already been encoded in the histograms associated with MECNs, and the second factor (named ground distance in optimal transport theorem) is the geodesic distance between cell types along the gene

expression manifold.

2, Revise Figure 1 to be more comprehensible.

We are sorry for the confusion in Figure 1. On one hand, we have to clarify that the ME graph is indeed constructed with the cell type histogram (Figure 1b) and the manifold distance (Figure 1c) as the direct inputs. In the ME graph construction, the cell type histogram is the representation of nodes in the graph, and the edge between two nodes is defined by the earth mover's distance (EMD) (the manifold distance is used here to act as the ground distance of EMD) between the two histograms. So from the authors' view, the cell type histogram (Figure 1b) and manifold distance (Figure 1c) are two relatively independent inputs (in a parallel manner) to ME graph construction, rather than in a tandem manner.

On the other hand, before the histogram (Figure 1b) and the manifold distance (Figure 2c), there is actually a cell clustering step, which solely takes the expression matrix as input. The cell type histogram (Figure 1b) is obtained by combining the spatial coordinates and the clustering result, and the manifold distance (Figure 1c) is obtained by combining the expression matrix and the clustering result. We added these elements in the new version of Figure 1. Furthermore, to make the figure more clear, we updated some other texts in figure 1. Such as "Histogram of cell clusters", and "Optimal transport". And the figure legends were also updated accordingly.

2. In the section “SOTIP characterizes spatial and molecular tumor-immune organization” it is also unclear what was done. The authors merely state that they used SOTIP. I am missing some details. Was the SDM module used? If so, did the authors indicate that they expect to find two domains? Did the authors use the cell types as defined in the original papers? How does the EMD graph between cell types look like for these proteomic datasets? This last point is crucial because these proteomic datasets have much less genes profiled (on the order of dozens). As such, it is unclear how well the EMD graph will look like, as often a single protein can define a cell subset. It is very difficult to evaluate the approach when all these details are not disclosed.

Author’s response:

This is a great point. In the section “SOTIP characterizes spatial and molecular tumor-immune organization”, we used the SDM module of SOTIP to identify tumor and immune domains. The original papers provided the expert-annotated tissue annotations (2 domains, Figure 5d,g) so that they could be nice data to demonstrate SOTIP-SDM’s performance. We used the cell type annotations provided by the original paper.

It is the authors’ guess that the “EMD graph” mentioned by the review might be either the single cell UMAP plot or the ME graph. To make this clear, we provided both in the following (MIBI data of TNBC patient 4 as an example). The first figure is the single cell UMAP plot, and the second figure is the ME graph during the hierarchical merging process described in Methods “Spatial domain (SDM) identification”. With the iteration, the ME graph finally reaches to a graph containing 2 nodes, which is the number of domains that we set initially. Corresponding to the ME graphs during the hierarchical process, we also provided the animation showing the intermediate spatial domain results in the “Example of SDM identification on MIBI-TOF” section of <https://github.com/TencentAILabHealthcare/SOTIP>, and the gif file can be accessed in https://github.com/TencentAILabHealthcare/SOTIP/blob/master/SOTIP_analysis/images/merge_MIBI.gif.

Single cell UMAP plot

ME graphs of hierarchical merging iterations

3. I don't understand the plots in figure 5F and I. The legend states: "The power plots show the proportion of true positives (y axis) detected by different methods at a range of FDRs (x axis)". But I don't understand the underlying analysis. Is it for all proteins? Or only one of the proteins shown in the figure? If so, which one? How do the authors define true positives? Please clarify what exactly was done.

Author's response:

We thank the reviewer to point this out. First, we would like to answer the questions of the reviewer, then we explain the steps to show what we have done in the analysis in Figure 5f,i.

1, Is it for all proteins?

Yes. We analyzed all the proteins and identified those proteins that exhibited spatially polarized patterns towards the tissue boundaries.

2, How do the authors define true positives?

The original papers identified the polarized proteins based on the expert-annotated tissue boundaries, and also used established knowledge to validate these patterns. We used these list of polarized proteins as positive controls.

3, What exactly was done?

We identified the polarized proteins following the standard procedures of previous studies^{3,4}, with the following steps:

- Define spatial domains using either SpaGCN or SOTIP's SDM module.
- Classify immune cells into two groups. Group1: Proximal (those immune cells whose smallest distance from any tumor cells are smaller than a radius), and Group2: Distal (those immune cells whose smallest distance from any tumor cells are larger than a radius).
- For each protein, we compared the protein abundance between Group1 and Group2, and assessed the significance using two-sample rank-sum test, which is further corrected by Benjamini–Hochberg (BH) procedure for multiple hypotheses testing.
- We set different FDR cutoffs (the x-axis in Figure 5f,i) to filter those statistical significant polarized proteins and obtained the identification power by comparing with the true positives.

Above power analysis was also applied in previous studies^{5,6}. Please refer to "Molecular polarization with spatial proteomics data" in Methods section for specific settings.

Minor points:

1. I could not find a place in the manuscript that referenced supplementary figure 6B

Author's response:

Supplementary figure 6B was referenced in Line 714.

Reviewer #6 (Remarks to the Author: Impact):

I think that the method will impact thinking in the field

References

- 1 Hu, J. *et al.* SpaGCN: Integrating gene expression, spatial location and histology to identify spatial domains and spatially variable genes by graph convolutional network. *Nat Methods* **18**, 1342–1351, doi:10.1038/s41592-021-01255-8 (2021).
- 2 Dong, K. & Zhang, S. Deciphering spatial domains from spatially resolved transcriptomics with an adaptive graph attention auto-encoder. *Nat Commun* **13**, 1739, doi:10.1038/s41467-022-29439-6 (2022).
- 3 Keren, L. *et al.* A Structured Tumor-Immune Microenvironment in Triple Negative Breast Cancer Revealed by Multiplexed Ion Beam Imaging. *Cell* **174**, 1373–+, doi:10.1016/j.cell.2018.08.039 (2018).
- 4 Hartmann, F. J. *et al.* Single-cell metabolic profiling of human cytotoxic T cells. *Nat Biotechnol*, doi:10.1038/s41587-020-0651-8 (2020).
- 5 Zhu, J., Sun, S. & Zhou, X. SPARK-X: non-parametric modeling enables scalable and robust detection of spatial expression patterns for large spatial transcriptomic studies. *Genome Biol* **22**, 184, doi:10.1186/s13059-021-02404-0 (2021).
- 6 Sun, S., Zhu, J. & Zhou, X. Statistical analysis of spatial expression patterns for spatially resolved transcriptomic studies. *Nat Methods* **17**, 193–200, doi:10.1038/s41592-019-0701-7 (2020).

REVIEWERS' COMMENTS:

Reviewer #3 (Remarks to the Author: Overall significance):

I am satisfied with the response. I will leave it to the authors' discretion to share some of the points as part of the Discussion. Examples are "spatial domain type analysis" and the importance of feature selection in explaining the differences between Fig 5e and 5f.

Reviewer #4 (Remarks to the Author: Overall significance):

EDITORIAL NOTE: Please see the attached file for comments from this Reviewer.

Reviewer #6 (Remarks to the Author: Overall significance):

The reviewers have adequately addressed my comments.

Regarding my questions in comments 3 and 4, they have provided answers in the rebuttle, but I could find similar information in the manuscript. The authors should update the results and methods sections to include this information.

Responses to the reviewers' comments

Response to all reviewers:

We are very grateful to the reviewers for their positive and constructive comments on our manuscript. In this 3rd response letter, original comments of reviewers are highlighted in yellow, author's responses are in black.

Reviewer comments:

Reviewer #3 (Remarks to the Author: Overall significance):

I am satisfied with the response. I will leave it to the authors' discretion to share some of the points as part of the Discussion. Examples are "spatial domain type analysis" and the importance of feature selection in explaining the differences between Fig 5e and 5f.

Response: Thanks! We have updated the Discussion accordingly (highlighted in red)..

Reviewer #4 (Remarks to the Author: Overall significance):

Authors have addressed my questions. I don't have any further questions about this manuscript. Thanks!

Response: Thanks!

Reviewer #6 (Remarks to the Author: Overall significance):

The reviewers have adequately addressed my comments. Regarding my questions in comments 3 and 4, they have provided answers in the rebuttle, but I could find similar information in the manuscript. The authors should update the results and methods sections to include this information.

Response: Thanks for the nice comment! We have updated these information in Methods (highlighted in red).